# Optimization and Generalization Guarantees for Weight Normalization

**Pedro Cisneros-Velarde**[*]                                        *pacisne@gmail.com*
*VMware Research*

**Zhijie Chen**                                                     *lucmon@illinois.edu*
*University of Illinois Urbana-Champaign*

**Sanmi Koyejo**                                                *sanmi@cs.stanford.edu*
*Stanford University*

**Arindam Banerjee**                                            *arindamb@illinois.edu*
*University of Illinois Urbana-Champaign*

**Reviewed on OpenReview:** *https://openreview.net/forum?id=gpHOtQQPJG*

## Abstract

Weight normalization (WeightNorm) is widely used in practice for the training of deep neural networks and modern deep learning libraries have built-in implementations of it. In this paper, we provide the first theoretical characterizations of both optimization and generalization of deep WeightNorm models with smooth activation functions. For optimization, from the form of the *Hessian of the loss*, we note that a small *Hessian of the predictor* leads to a tractable analysis. Thus, we bound the spectral norm of the Hessian of WeightNorm networks and show its dependence on the network width and weight normalization terms–the latter being unique to networks without WeightNorm. Then, we use this bound to establish training convergence guarantees under suitable assumptions for gradient decent. For generalization, we use WeightNorm to get a uniform convergence based generalization bound, which is independent from the width and depends sublinearly on the depth. Finally, we present experimental results which illustrate how the normalization terms and other quantities of theoretical interest relate to the training of WeightNorm networks.

## 1 Introduction

Weight normalization (WeightNorm) was first introduced by Salimans and Kingma (2016). The idea is to normalize each weight vector—by dividing it by its Euclidean norm—in order to decouple its length from its direction. Salimans and Kingma (2016) reported that WeightNorm is able to speed-up the training of neural networks, as an alternative to the then recently introduced and still widely-used batch normalization (BatchNorm) (Ioffe and Szegedy, 2015). WeightNorm is different from BatchNorm because it is independent from the statistics of the batch sample being used at the gradient step. It is also different from another class of normalization, called layer normalization (LayerNorm) (Ba et al., 2016) in that it does not entail the normalization of any activation function of the neural network. Thus WeightNorm also has less computational overhead because it does not require the computation and storage of additional mean and standard deviation statistics as the other methods do. However, its empirical testing performance (or accuracy) has often been found to not be as good as other normalization methods by itself, and thus various versions have been formulated to improve its performance (Salimans and Kingma, 2016; Huang et al., 2017; Qiao et al., 2020). Nevertheless, WeightNorm has been one of the most popular normalization methods since its introduction, and as a result current machine learning libraries include built-in implementations of it, e.g., PyTorch 2.0.

---

[*]Corresponding author.

Parallel to the empirical development of normalization methods, recent years have seen advances in theoretically understanding convergence of gradient descent (GD) and variants for deep learning models (Du et al., 2019; Allen-Zhu et al., 2019; Zou et al., 2020; Nguyen, 2021; Liu et al., 2022; Banerjee et al., 2023). However, to the best of our knowledge, no prior work has focused on providing formal optimization and generalization guarantees for deep models with WeightNorm. In this paper, our contribution is to study the effect of WeightNorm on both training and generalization from a theoretical perspective. In particular, we focus on smooth networks—deep models with smooth activations—a setting which has become increasingly popular in recent years (Du et al., 2019; Huang and Yau, 2020; Liu et al., 2022; Banerjee et al., 2023).

Training a neural network is a non-convex optimization problem, and establishing optimization guarantees using GD requires analyzing the empirical loss used for training. Particularly, our analysis is based on the second-order Taylor expansion of the empirical loss, for which we require bounds on its Hessian and gradient. We observe that it is possible to bound the empirical loss Hessian by using a bound on the Hessian of the network or *predictor*. Thus, our first technical contribution (Section 4) is a characterization of the spectral norm of the Hessian of a smooth WeightNorm neural network. Similar to networks without WeightNorm (Liu et al., 2020; Banerjee et al., 2023), our bound decreases with width, but unlike networks without WeightNorm, our bound also decreases with the minimum weight vector norm across all hidden layers. A similar dependence appears when bounding the norm of the empirical loss gradient, which follows from bounding the norm of the predictor gradient (with respect to the network weights). Now, towards establishing the generalization of WeightNorm networks, we obtain a bound on the output of the predictor which does not have any dependence on the minimum weight vector norm. Similarly, the Lipschitz constant of the network does not have such a dependency either. These results are used to subsequently establish a uniform convergence bound based on the Rademacher complexity of WeightNorm networks (Section 6). Remarkably, all of our bounds used for establishing both optimization and generalization guarantees have at most a polynomial dependence on the depth of the network.

Our second technical contribution (Section 5) is to provide optimization guarantees for the training of smooth deep networks with WeightNorm by GD under suitable assumptions. Using our previously derived bounds, we use the restricted strong convexity (RSC) approach recently introduced by Banerjee et al. (2023) to establish sufficient conditions that ensure a decrease on the value of the loss function with respect to its minimum value. In particular, one such condition relies on an inequality (lower bound) on the quantity $\|\nabla\mathcal{L}\|_2^2/\mathcal{L}$, where $\mathcal{L}$ is the empirical loss and $\nabla\mathcal{L}$ its gradient. This inequality benefits from a larger width as well as a larger minimum weight vector norm–as opposed to another condition for networks without WeightNorm which only benefits from a larger width (Banerjee et al., 2023). We relate our condition to the so-called restricted Polyak-Lojasiewicz (PL) condition, showing that ours is milder.

Our third technical contribution (Section 6) is to provide generalization guarantees to smooth networks with WeightNorm. We iteratively use WeightNorm properties and our network scaling to obtain a generalization gap which is independent of the network width—our bound is simply $\mathcal{O}(\sqrt{L}/\sqrt{n})$, where $L$ is the depth and $n$ the sample size. Thus, WeightNorm allows for a natural non-exponential dependence on the depth compared to networks without WeightNorm (Bartlett et al., 2017; Neyshabur et al., 2018; Golowich et al., 2018). Our generalization guarantee uses a uniform convergence approach based on the Rademacher complexity of WeightNorm networks.

Fourth, we present experiments on the training of WeightNorm networks (Section 7). We present numerical evidence that the convergence rate improves for larger values of (i) the ratio $\|\nabla\mathcal{L}\|_2^2/\mathcal{L}$, and (ii) the minimum weight vector norm. According to our theoretical results, larger values of these two quantities strengthen the RSC property of the empirical loss, which then improves the optimization convergence rate.

## 2 Related work

**WeightNorm and Other Normalizations**  WeightNorm was first proposed by Salimans and Kingma (2016), after a year from the introduction of BatchNorm by Ioffe and Szegedy (2015). LayerNorm was proposed in the same year by Ba et al. (2016), and has also become very popular (Xu et al., 2019). Extensions and variations of WeightNorm have been introduced since then, for example, centered weight normalization (Huang et al., 2017), orthogonal weight normalization (Huang et al., 2018), and weight standardization (Qiao et al.,

2020). Theoretical analysis of normalization techniques in general has caught an increasing recent interest. We focus mostly on WeightNorm, but mention that BatchNorm has some theoretical work, e.g. (Santurkar et al., 2018; Lian and Liu, 2019). The work (Arpit et al., 2019) uses a theoretical approach to better come up with initialization schemes tailored for WeightNorm networks with ReLU activations. In contrast, our work focuses in understanding the optimization (training) and generalization of WeightNorm networks with smooth activations. Wu et al. (2020) study the effect of WeightNorm on the convergence of overparameterized least squares problem (which translates to a normalization of the parameters to estimate). We, in contrast, study the non-convex problem of optimizing WeighNorm networks. Dukler et al. (2020) present exponential convergence guarantees on shallow ReLU networks with WeightNorm, whereas we study smooth deep networks and present generalization too. We conclude by remarking that the field of normalization is growing in proposing new normalization schemes and improving their understanding and application—we refer the reader to the recent survey (Huang et al., 2023).

**Training Guarantees of Neural Networks**  Given the increasingly vast literature, we refer the readers to the surveys (Fan et al., 2021; Bartlett et al., 2021) for an overview of the field of gradient descent training of neural networks. Deep ReLU networks were analyzed by Zou and Gu (2019); Zou et al. (2020); Allen-Zhu et al. (2019); Nguyen (2021); Nguyen et al. (2021), whereas deep networks with smooth activations—our case—were analyzed by Du et al. (2019); Nguyen and Mondelli (2020); Liu et al. (2022). Regarding the convergence analysis of gradient descent, Du et al. (2019); Allen-Zhu et al. (2019); Zou and Gu (2019); Zou et al. (2020); Liu et al. (2022) used the Neural Tangent Kernel (NTK) approach (Jacot et al., 2018); whereas Banerjee et al. (2023) used the RSC approach. All training guarantees hold over a neighborhood around the initialization point of the network's weights.

**Generalization of Neural Networks**  To the best of our knowledge, the first work in studying the Rademacher complexity for neural networks was (Neyshabur et al., 2015). Then, the seminal work (Bartlett et al., 2017) provided generalization bounds for neural networks considering Lipschitz activation functions. The bound, obtained using a covering approach, depends exponentially on the depth of the network. The generalization bound was tightened for ReLU activations in (Neyshabur et al., 2018) using a PAC-Bayes approach, but the same exponential dependency persisted. Then, the work (Golowich et al., 2018) avoided some exponential dependence on the depth and established conditions on the norm of the network's weight matrices in order to obtain generalization bounds independent of depth. Unlike our work, all the cited works do not study WeightNorm networks—indeed, we show, through a suitable adaptation of such existing results, that WeightNorm allows the generalization bound to depend on $\sqrt{L}$, where $L$ is the depth. Finally, we remark that our generalization bound estimates how well finding a solution to the empirical loss approximates finding a solution to the loss under the real data distribution. A different problem in the literature is inductive bias, i.e., how a specific optimization method, while training the network, may converge to a solution that generalizes well. This was studied for WeightNorm under GD in (Morwani and Ramaswamy, 2022) and under stochastic GD in (Li et al., 2022).

## 3 Problem setup: deep learning with WeightNorm

Consider a training set $\{(\mathbf{x}_i, y_i)\}_{i=1}^n, \mathbf{x}_i \in \mathcal{X} \subseteq \mathbb{R}^d, y_i \in \mathcal{Y} \subseteq \mathbb{R}$. For a suitable loss function $\ell$, the goal is to minimize the empirical loss: $\mathcal{L}(\theta) = \frac{1}{n} \sum_{i=1}^n \ell(y_i, \hat{y}_i) = \frac{1}{n} \sum_{i=1}^n \ell(y_i, f(\theta; \mathbf{x}_i))$, where the prediction $\hat{y}_i := f(\theta; \mathbf{x}_i)$ is from a deep neural network with parameter vector $\theta \in \mathbb{R}^p$ for some positive integer $p$. In our setting $f$ is a feed-forward multi-layer (fully-connected) neural network with weight normalization (WeightNorm), having depth $L$ and hidden layers with widths $m_l, l \in [L] := \{1, \ldots, L\}$; and it is given by

$$\alpha^{(0)}(\mathbf{x}) = \mathbf{x} \ , \ \alpha_i^{(l)}(\mathbf{x}) = \phi\left(\frac{1}{\sqrt{m_l}} \frac{(W_i^{(l)})^\top}{\left\|W_i^{(l)}\right\|_2} \alpha^{(l-1)}(\mathbf{x})\right) , \ i = 1, \ldots, m_l, \ l = 1, \ldots, L \ ,$$

$$f(\theta; \mathbf{x}) = \mathbf{v}^\top \alpha^{(L)}(\mathbf{x}) \ , \tag{1}$$

where $\alpha_i^{(l)}(\mathbf{x}) \in \mathbb{R}$ is the $i$-th entry of the activation function $\alpha^{(l)}(\mathbf{x}) \in \mathbb{R}^{m_l}$, $W_i^{(l)} \in \mathbb{R}^{m_{l-1}}$ is the weight vector corresponding to the transpose of the $i$-th row of the layer-wise weight matrix $W^{(l)} \in \mathbb{R}^{m_l \times m_{l-1}}, l \in [L]$, $\mathbf{v} \in \mathbb{R}^{m_L}$ is the last layer vector, $\phi(\cdot)$ is the smooth (pointwise) activation function. The total set of parameters is represented by the parameter vector

$$\theta := (\text{vec}(W^{(1)})^\top, \ldots, \text{vec}(W^{(L)})^\top, \mathbf{v}^\top)^\top \in \mathbb{R}^{\sum_{k=1}^{L} m_k m_{k-1} + m_L} , \tag{2}$$

with $m_0 = d$. The pre-activation function $\tilde{\alpha}^{(l)} \in \mathbb{R}^{m_l}$ has its $i$-th entry defined by $\alpha_i^{(l)}(\mathbf{x}) = \phi(\tilde{\alpha}_i^{(l)}(\mathbf{x}))$, $l \in [L]$. For simplicity, we assume that the width of all hidden layers is the same, i.e., $m_l = m, l \in [L]$, and so $\theta \in \mathbb{R}^{dm + (L-1)m^2 + m}$. We remark that $f$ is also called the *predictor* since it is the predictive output of the neural network.

Define the pointwise loss $\ell_i := \ell(y_i, \cdot) : \mathbb{R} \to \mathbb{R}_+$ and denote its first- and second-derivative as $\ell_i' := \frac{d\ell(y_i, \hat{y}_i)}{d\hat{y}_i}$ and $\ell_i'' := \frac{d^2\ell(y_i, \hat{y}_i)}{d\hat{y}_i^2}$. The particular case of square loss is $\ell(y_i, \hat{y}_i) = (y_i - \hat{y}_i)^2$. We denote the gradient and Hessian of $f(\cdot; \mathbf{x}_i) : \mathbb{R}^p \to \mathbb{R}$ by $\nabla_i f := \frac{\partial f(\theta; \mathbf{x}_i)}{\partial \theta}$ and $\nabla_i^2 f := \frac{\partial^2 f(\theta; \mathbf{x}_i)}{\partial \theta^2}$ respectively. The gradient and Hessian of the empirical loss w.r.t. $\theta$ are given by

$$\frac{\partial \mathcal{L}(\theta)}{\partial \theta} = \frac{1}{n} \sum_{i=1}^{n} \ell_i' \nabla_i f , \quad \frac{\partial^2 \mathcal{L}(\theta)}{\partial \theta^2} = \frac{1}{n} \sum_{i=1}^{n} \left[ \ell_i'' \nabla_i f \nabla_i f^\top + \ell_i' \nabla_i^2 f \right] . \tag{3}$$

We denote the gradient with respect to the input data $\mathbf{x}_i$, $i \in [n]$, by $\nabla_{\mathbf{x}} f(\theta; \mathbf{x}_i) := \left. \frac{\partial f(\theta; \mathbf{x})}{\partial \mathbf{x}} \right|_{\mathbf{x} = \mathbf{x}_i}$ for any $\theta \in \mathbb{R}^p$. Let $\| \cdot \|_2$ denote the spectral norm for matrices and $L_2$-norm for vectors. Finally, we introduce the following convenient notation:

**Definition 3.1 (Minimum weight vector norm).** $\|\bar{W}\|_2 = \min_{\substack{i \in [m] \\ l \in [L]}} \|W_i^{(l)}\|_2$. As an example of the notation, let $\{W_i^{(l)}\}_{i=1, l=1}^{m, L}$ and $\{W'_i^{(l)}\}_{i=1, l=1}^{m, L}$ be two sets of hidden weights and define $f(W_i^{(l)}, W'_i^{(l)}) := \alpha W_i^{(l)} + \beta W'_i^{(l)}$ for some $\alpha, \beta \in \mathbb{R}$, then $\|\bar{f}(W, W')\|_2 = \min_{\substack{i \in [m] \\ l \in [L]}} \|\alpha W_i^{(l)} + \beta W'_i^{(l)}\|_2$.

We state the following two assumptions:

**Assumption 1 (Activation function).** *The activation $\phi$ is 1-Lipschitz, i.e., $|\phi'| \leq 1$, and $\beta_\phi$-smooth, i.e., $|\phi_l''| \leq \beta_\phi$.*

**Assumption 2 (Ouput layer and input data).** *We initialize the vector of the output layer $\mathbf{v}_0 \in \mathbb{R}^m$ as a unit vector, i.e., $\|\mathbf{v}_0\|_2 = 1$. Further, we assume the input data satisfies: $\|\mathbf{x}_i\|_2 = 1$, $i \in [n]$.*

The assumption $\|\mathbf{x}\|_2 = 1$ is for convenient scaling. Normalization assumptions are common in the literature (Allen-Zhu et al., 2019; Oymak and Soltanolkotabi, 2020; Nguyen et al., 2021). The unit value for the last layer's weight norm at initialization is also for convenience, since our results hold under appropriate scaling for any other constant in $\mathcal{O}(1)$.

We introduce some additional notation. For two vectors $\pi, \bar{\pi} \in \mathbb{R}^q$, $\cos(\pi, \bar{\pi})$ denotes the cosine of the angle between $\pi$ and $\bar{\pi}$. We define the Euclidean ball around $\pi$ with radius $\rho > 0$ by $B_\rho^{\text{Euc}}(\pi) := \{\hat{\pi} \in \mathbb{R}^q \mid \|\hat{\pi} - \pi\|_2 \leq \rho\}$.

Finally, we point out that the original WeightNorm paper by Salimans and Kingma (2016) also has a scalar multiplying the normalized weight vectors $W_i^{(l)} / \|W_i^{(l)}\|_2$, $l \in [L]$. Their work considers optimizing over such scalars whereas we consider a fixed scaling for our analysis.

## 4 Characterizing WeightNorm bounds on the loss and predictor

To characterize both the optimization with gradient descent (GD) and generalization of WeightNorm networks, we need bounds on the empirical loss and predictor, as well as on their gradients, and the Hessian of the

latter. For example, the optimization guarantees in Section 5 make use of the second-order Taylor expansion of the empirical loss $\mathcal{L}$ for which we need to bound its Hessian and gradient. As seen in equation (3), the loss Hessian contains terms related to the gradient and the Hessian of the predictor $f$, which we then need to bound. Now, regarding our generalization guarantees in Section 6, we need to obtain a bound on the values of the predictor in order to bound the generalization gap. We obtain this bound by using intermediate results derived from both the proofs of the Hessian and empirical loss $\mathcal{L}$ bounds.

All of these bounds are presented in this section, which we believe could also be of independent interest. The proofs are found in Section A and Section B of the appendix.

**Theorem 4.1** (**Hessian bound for WeightNorm**). *Under Assumptions 1 and 2, for any $\theta \in \mathbb{R}^p$ with $\mathbf{v} \in B_{\rho_1}^{\mathrm{Euc}}(\mathbf{v}_0)$, and any $\mathbf{x}_i, i \in [n]$, we have*

$$\left\| \nabla_\theta^2 f(\theta; \mathbf{x}_i) \right\|_2 \leq \mathcal{O}\left( \frac{(1+\rho_1)L^3(\frac{1}{\sqrt{m}} + L^2 \max\{|\phi(0)|, |\phi(0)|^2\})}{\min\left\{ \|\bar{W}\|_2, \|\bar{W}\|_2^2 \right\}} \right). \tag{4}$$

**Corollary 4.1** (**Hessian bound for WeightNorm under $\phi(0)=0$**). *Under Assumptions 1 and 2, and assuming that the activation function satisfies $\phi(0) = 0$, we have for any $\theta \in \mathbb{R}^p$ with $\mathbf{v} \in B_{\rho_1}^{\mathrm{Euc}}(\mathbf{v}_0)$, and any $\mathbf{x}_i, i \in [n]$,*

$$\left\| \nabla_\theta^2 f(\theta; \mathbf{x}_i) \right\|_2 \leq \mathcal{O}\left( \frac{(1+\rho_1)L^3}{\sqrt{m} \min\left\{ \|\bar{W}\|_2, \|\bar{W}\|_2^2 \right\}} \right). \tag{5}$$

**Remark 4.1** (**Comparison to networks without WeightNorm**). The recent works (Liu et al., 2020; Banerjee et al., 2023) analyzed the Hessian spectral norm bound for feedforward networks without WeightNorm. (Liu et al., 2020) presented an exponential dependence on the network's depth $L$, whereas (Banerjee et al., 2023) avoided exponential dependence upon the choice of the initialization variance for the hidden layers' weights. We show that WeightNorm eliminates such exponential dependence for at most $L^5$ independent from any initialization of the weights, which improves to $L^3$ when $\phi(0) = 0$. Another important difference is that our bound also depends on the inverse of the minimum weight vector norm—large enough values on the weight vectors decrease the upper bound. Finally, our bound is defined over any value on the weights, and not just over a neighborhood around the initialization point. $\qquad\square$

**Remark 4.2** (**Dependence on the activation function**). Corollary 4.1 shows that the Hessian bound becomes tighter when $\phi(0) = 0$ because of an additional $\frac{1}{\sqrt{m}}$ scaling. In contrast, the Hessian bounds by Liu et al. (2020); Banerjee et al. (2023) have the scaling $\frac{1}{\sqrt{m}}$ *independent* from the value of $\phi(0)$. The reason for this difference is the use of different scale factors: while those two other works introduce an extra scale factor $\frac{1}{\sqrt{m}}$ on the linear output layer, we do not. In our paper, we follow the neural network scaling as done in the seminal work on the Neural Tangent Kernel by Jacot et al. (2018). $\qquad\square$

**Lemma 4.1** (**Predictor gradient bounds**). *Under Assumptions 1 and 2, for any $\theta \in \mathbb{R}^p$ with $\mathbf{v} \in B_{\rho_1}^{\mathrm{Euc}}(\mathbf{v}_0)$, and any $\mathbf{x}_i, i \in [n]$, we have*

$$\|\nabla_\theta f(\theta; \mathbf{x}_i)\|_2 \leq \varrho_\theta \leq \mathcal{O}\left( (1 + L|\phi(0)|\sqrt{m}) \left( 1 + \frac{\sqrt{L}(1+\rho_1)}{\|\bar{W}\|_2} \right) \right) \tag{6}$$

*where $\varrho_\theta^2 := (1 + L|\phi(0)|\sqrt{m})^2 + 4(1+\rho_1)^2 \sum_{l=1}^{L} \frac{1}{\|\bar{W}\|_2^2} \left( \frac{1}{\sqrt{m}} + (l-1)|\phi(0)| \right)^2$, and*

$$\|\nabla_\mathbf{x} f(\theta; \mathbf{x}_i)\|_2 \leq 1 + \rho_1. \tag{7}$$

**Corollary 4.2** (**Predictor gradient bounds under $\phi(0) = 0$**). *Under Assumptions 1 and 2, and assuming that the activation function satisfies $\phi(0) = 0$, we have for any $\theta \in \mathbb{R}^p$ with $\mathbf{v} \in B_{\rho_1}^{\mathrm{Euc}}(\mathbf{v}_0)$, and any $\mathbf{x}_i, i \in [n]$,*

$$\|\nabla_\theta f(\theta; \mathbf{x}_i)\|_2 \leq \varrho_\theta \leq \mathcal{O}\left( 1 + \frac{\sqrt{L}(1+\rho_1)}{\sqrt{m}\|\bar{W}\|_2} \right) \tag{8}$$

*where $\varrho_\theta^2 := 1 + \frac{4L(1+\rho_1)^2}{m\|\bar{W}\|_2^2}$.*

**Remark 4.3 (The Lipschitz constant is the same for networks of any depth).** Surprisingly, the Lipschitz constant of the network (7)—unlike networks without WeightNorm—does not explicitly depend on the depth $L$ of the network nor its width $m$. It depends on the radius $\rho_1$ which can be set to be a constant $\mathcal{O}(1)$. This is a direct effect of WeightNorm: when computing $\nabla_{\mathbf{x}} f(\theta; \mathbf{x}_i)$, we use a chain-rule argument that depends on the product of Jacobians of the output $\alpha^{(l)}$ at layer $l$ with respect to the previous output $\alpha^{(l-1)}$, and we show each Jacobian has at most unit norm due to WeightNorm. Thus, no matter how deep the network is, the final effect this product of Jacobians has on the Lipschitz constant of the network is just a constant one. □

**Proposition 4.1 (Empirical loss and empirical loss gradient bounds).** *Consider the square loss. Under Assumptions 1 and 2, the following inequality holds for any $\theta \in \mathbb{R}^p$ with $\mathbf{v} \in B_{\rho_1}^{\mathrm{Euc}}(\mathbf{v}_0)$,*

$$\mathcal{L}(\theta) \leq \varphi \leq \mathcal{O}((1 + \rho_1)^2 (1 + L^2 |\phi(0)|^2 m)), \tag{9}$$

*where $\varphi := \frac{2}{n} \sum_{i=1}^n y_i^2 + 2(1 + \rho_1)^2 (1 + L|\phi(0)|\sqrt{m})^2$. Moreover,*

$$\|\nabla_\theta \mathcal{L}(\theta)\|_2 \leq 2\sqrt{\mathcal{L}(\theta)}\varrho_\theta \leq 2\varrho_\theta \sqrt{\varphi}, \tag{10}$$

*with $\varrho_\theta$ as in Lemma 4.1.*

**Corollary 4.3 (Empirical loss and empirical loss gradient bounds under $\phi(0) = 0$).** *Consider the square loss. Under Assumptions 1 and 2, and assuming that the activation function satisfies $\phi(0) = 0$, the following inequality holds for any $\theta \in \mathbb{R}^p$ with $\mathbf{v} \in B_{\rho_1}^{\mathrm{Euc}}(\mathbf{v}_0)$,*

$$\mathcal{L}(\theta) \leq \varphi \leq \mathcal{O}((1 + \rho_1)^2), \tag{11}$$

*where $\varphi := \frac{2}{n} \sum_{i=1}^n y_i^2 + 2(1 + \rho_1)^2$. Moreover, the bound in equation (10) holds with $\varrho_\theta$ as in Corollary 4.2.*

**Remark 4.4 (Comparison to networks without WeightNorm).** As in Remark 4.1, our bounds are polynomial on $L$ irrespective of weight values, whereas previous results were exponential on $L$. □

# 5 Optimization guarantees for WeightNorm

In this section we prove our training guarantees for WeightNorm networks under the square loss. We first introduce two technical lemmas that will be used for our convergence analysis, defining the restricted strong convexity (RSC) property and a smoothness-like property respectively. These two properties use the bounds derived in Section 4. All proofs are found in Section C of the appendix.

**Definition 5.1 ($\mathbf{Q}_\kappa^\theta$ sets).** *Given $\theta \in \mathbb{R}^p$ and $\kappa \in (0, 1]$, define $Q_\kappa^\theta := \{\hat{\theta} \in \mathbb{R}^p \mid |\cos(\hat{\theta} - \theta, \nabla_\theta \mathcal{L}(\theta))| \geq \kappa\}$.*

**Lemma 5.1 (RSC for WeightNorm under Square Loss).** *For square loss, under Assumptions 1 and 2, for every $\theta' \in Q_\kappa^\theta \cap B_{\rho_2}^{\mathrm{Euc}}(\theta)$ with $\theta \in \mathbb{R}^p$ and $\mathbf{v}, \mathbf{v}' \in B_{\rho_1}^{\mathrm{Euc}}(\mathbf{v}_0)$,*

$$\mathcal{L}(\theta') \geq \mathcal{L}(\theta) + \langle \theta' - \theta, \nabla_\theta \mathcal{L}(\theta) \rangle + \frac{\alpha_{\theta,\theta'}}{2} \|\theta' - \theta\|_2^2, \tag{12}$$

*with*

$$\alpha_{\theta,\theta'} = \frac{\kappa^2}{2} \frac{\|\nabla_\theta \mathcal{L}(\theta)\|_2^2}{\mathcal{L}(\theta)}$$
$$- \mathcal{O}\left( \left(1 + \frac{\sqrt{L}}{\|\bar{W}\|_2}\right) \frac{(1 + \rho_2)(1 + \rho_1)^3 L^3 A(\frac{1}{\sqrt{m}}, L^2, |\phi(0)|) B(1, L, |\phi(0)|\sqrt{m})}{\min\left\{\|\bar{f}(W', W)_\xi\|_2, \|\bar{f}(W', W)_\xi\|_2^2\right\}} \right), \tag{13}$$

*where $f(W', W)_\xi := \xi W' + (1 - \xi)W$ for some $\xi \in [0, 1]$, $A(\frac{1}{\sqrt{m}}, L^2, |\phi(0)|) := \frac{1}{\sqrt{m}} + L^2 \max\{|\phi(0)|, |\phi(0)|^2\}$ and $B(1, L, |\phi(0)|\sqrt{m}) := 1 + L|\phi(0)|\sqrt{m}$. We say that the empirical loss $\mathcal{L}$ satisfies the RSC property w.r.t. $(Q_\kappa^\theta \cap B_{\rho_2}^{\mathrm{Euc}}(\theta), \theta)$ whenever $\alpha_{\theta,\theta'} > 0$.*

**Corollary 5.1** (**RSC for WeightNorm under Square Loss and** $\phi(0) = 0$). *Consider the square loss. Under Assumptions 1 and 2 and assuming that the activation function satisfies $\phi(0) = 0$, for every $\theta' \in Q_\kappa^\theta \cap B_{\rho_2}^{\mathrm{Euc}}(\theta)$ with $\theta \in \mathbb{R}^p$ and $\mathbf{v}, \mathbf{v}' \in B_{\rho_1}^{\mathrm{Euc}}(\mathbf{v}_0)$, the inequality in (13) becomes*

$$
\begin{aligned}
\alpha_{\theta,\theta'} &= \frac{\kappa^2}{2} \frac{\|\nabla_\theta \mathcal{L}(\theta)\|_2^2}{\mathcal{L}(\theta)} \\
&- \mathcal{O}\left( \left(1 + \frac{\sqrt{L}}{\sqrt{m}\|\bar{W}\|_2}\right) \frac{(1+\rho_2)(1+\rho_1)^3 L^3}{\sqrt{m} \min\left\{\left\|\bar{f}(W',W)_\xi\right\|_2, \left\|\bar{f}(W',W)_\xi\right\|_2^2\right\}} \right)
\end{aligned}
\tag{14}
$$

*using the same notation as in Lemma 5.1.*

**Remark 5.1** (**The RSC parameter $\alpha_\theta$ and prior work**). Our RSC characterization in the left-hand side of (13) depends on $\frac{\|\nabla_\theta \mathcal{L}(\theta)\|_2^2}{\mathcal{L}(\theta)}$, whereas (Banerjee et al., 2023)—the work that introduced the RSC-based optimization analysis—depends on $\|\frac{1}{n}\sum_{i=1}^n \nabla_\theta f(\theta, x_i)\|_2^2$. Moreover, the minimum weight vector norm appears in the right-hand side since we used the bounds on Section 4. $\square$

**Remark 5.2** (**Connection with the restricted Polyak-Lojasiewicz (PL) inequality**). The RSC condition has the form $\frac{\|\nabla_\theta \mathcal{L}(\theta)\|_2^2}{\mathcal{L}(\theta)} \geq 2\mu_{\theta,\theta'}$ where $\mu_{\theta,\theta'}$ depends on the parameters $\theta$ and $\theta'$, which are in turn restricted to a specific set. This is a milder condition than the so-called *restricted PL condition* (Oymak and Soltanolkotabi, 2019; Banerjee et al., 2023), which is satisfied whenever $\mu_{\theta,\theta'}$ is a constant independent from both $\theta$ and $\theta'$. Our RSC condition becomes even milder when $\phi(0) = 0$ (see Corollary 5.1), since now the parameter $\mu_{\theta,\theta'}$ has the term $1/\sqrt{m}$ which decreases with the width. $\square$

**Remark 5.3** (**The right-hand side of the RSC parameter $\alpha_\theta$**). When $\phi(0) = 0$, Corollary 5.1 shows that the right-hand side of (13) will have the scaling parameter $\frac{1}{\sqrt{m}}$ due to the Hessian bound (Corollary 4.1). $\square$

**Lemma 5.2** (**Smoothness-like property for WeightNorm under Square Loss**). *For square loss, under Assumptions 1 and 2, for every $\theta, \theta' \in \mathbb{R}^p$ with $\mathbf{v}, \mathbf{v}' \in B_{\rho_1}^{\mathrm{Euc}}(\mathbf{v}_0)$,*

$$
\mathcal{L}(\theta') \leq \mathcal{L}(\theta) + \langle \theta' - \theta, \nabla_\theta \mathcal{L}(\theta) \rangle + \frac{\beta_{\theta,\theta'}}{2}\|\theta' - \theta\|_2^2 ,
\tag{15}
$$

*with*

$$
\beta_{\theta,\theta'} \leq \mathcal{O}\left( L^4 A(1, L^2, |\phi(0)|)(1+\rho_1)^2 \left(1 + \frac{1}{\min\{\|\bar{f}(W',W)_\xi\|_2, \|\bar{f}(W',W)_\xi\|_2^2\}}\right) \right) ,
\tag{16}
$$

*where $f(W',W)_\xi := \xi W' + (1-\xi)W$ for some $\xi \in [0,1]$, $A(1, L^2, |\phi(0)|) = (1 + L^2 \max\{|\phi(0)|, |\phi(0)|^2\}m)^2$.*

**Corollary 5.2** (**Smoothness-like property for WeightNorm under Square Loss and $\phi(0) = 0$**). *Consider the square loss. Under Assumptions 1 and 2 and assuming that the activation function satisfies $\phi(0) = 0$, for every $\theta, \theta' \in \mathbb{R}^p$ with $\mathbf{v}, \mathbf{v}' \in B_{\rho_1}^{\mathrm{Euc}}(\mathbf{v}_0)$, the inequality in (16) becomes*

$$
\beta_{\theta,\theta'} \leq \mathcal{O}\left( 1 + \frac{L^3(1+\rho_1)^2}{\sqrt{m}\min\{\|\bar{f}(W',W)_\xi\|_2, \|\bar{f}(W',W)_\xi\|_2^2\}}) \right)
\tag{17}
$$

*using the same notation as in Lemma 5.2.*

**Remark 5.4** (**A stronger notion of convexity and smoothness**). If the expressions in equations (12) and (15) hold for any $\theta, \theta' \in \mathbb{R}^p$ so that $\alpha_{\theta,\theta'} \equiv \alpha > 0$ and $\beta_{\theta,\theta'} \equiv \beta > 0$, then we have the definitions of strong convexity and smoothness, respectively—moreover, it immediately follows that $\frac{\alpha}{\beta} < 1$. $\square$

The following assumptions are useful for establishing our convergence analysis—indeed, as we shortly show, a decrease on the loss value is guaranteed as long as these assumptions are satisfied.

**Assumption 3** (**Iterates' conditions**). *Consider the iterate $\theta_t \in \mathbb{R}^p$ with $\mathbf{v}_t \in B_{\rho_1}^{\mathrm{Euc}}(\mathbf{v}_0)$ and that:* **(A3.1)** *gradient descent (GD) update $\theta_{t+1} = \theta_t - \eta_t \nabla \mathcal{L}(\theta_t)$ with learning rate $\eta_t > 0$ chosen appropriately so that $\mathbf{v}_{t+1} \in B_{\rho_1}^{\mathrm{Euc}}(\mathbf{v}_0)$;* **(A3.2)** *$\rho_2$ is chosen so that $\theta_{t+1} \in B_{\rho_2}^{\mathrm{Euc}}(\theta_t)$.*

The first statement in Assumption 3 ensures that gradient descent has an appropriate learning rate. The second statement defines a ball around the current iterate that should cover the next iterate—in principle, its radius could be defined after defining the learning rate of GD of the first assumption; more details in Remark 5.7.

Finally, we present our main optimization result: a reduction on the empirical loss towards its minimum value (with the last layer in the Euclidean set of radius $\rho_1$) using gradient descent. This is proved by using Lemmas 5.1 and 5.2 and an adaptation of (Banerjee et al., 2023, Theorem 5.3); see Section C.2 of the appendix.

**Theorem 5.1** (**Global Empirical Loss Reduction for WeightNorm under Square Loss, (Banerjee et al., 2023, Theorem 5.3)**). *Let $B_t := Q_\kappa^{\theta_t} \cap B_{\rho_2}^{\text{Euc}}(\theta_t) \cap \{\theta \in \mathbb{R}^p \,|\, \mathbf{v} \in B_{\rho_1}^{\text{Euc}}(\mathbf{v}_0)\}$, $\theta^* \in$ $\text{arginf}_{\theta \in \{\theta \in \mathbb{R}^p \,|\, \mathbf{v} \in B_{\rho_1}^{\text{Euc}}(\mathbf{v}_0)\}} \mathcal{L}(\theta)$, $\overline{\theta}_t \in \text{arginf}_{\theta \in B_t} \mathcal{L}(\theta)$, and $\gamma_t := \frac{\mathcal{L}(\overline{\theta}_t) - \mathcal{L}(\theta^*)}{\mathcal{L}(\theta_t) - \mathcal{L}(\theta^*)}$. Let $\alpha_{\theta_t, \overline{\theta}_t}, \beta_{\theta_t}$ be as in Lemmas 5.1 and 5.2 respectively with $\beta_{\theta_t} \equiv \beta_{\theta_t, \theta_{t+1}} = \beta_{\theta_t, \theta_t - \eta_t \nabla_\theta \mathcal{L}(\theta_t)}$. Consider Assumptions 1, 2, and 3, and the RSC property $\alpha_{\theta_t, \overline{\theta}_t} > 0$. Then, we have $\gamma_t \in [0, 1)$; and if $\mathcal{L}(\theta_t) \neq \mathcal{L}(\theta^*)$, then for gradient descent with step size $\eta_t = \frac{1}{\beta_{\theta_t}}$, we have $\frac{\alpha_{\theta_t, \overline{\theta}_t}}{\beta_{\theta_t}} \in (0, 1]$ and*

$$\mathcal{L}(\theta_{t+1}) - \mathcal{L}(\theta^*) \leq \left(1 - \frac{\alpha_{\theta_t, \overline{\theta}_t}}{\beta_{\theta_t}}(1 - \gamma_t)\right)(\mathcal{L}(\theta_t) - \mathcal{L}(\theta^*)) . \tag{18}$$

The result in (18) implies training convergence as the number of iterations increase. Since the rate at which the loss decreases is time- and weight-dependent in (18), it is difficult to establish the number of steps needed for convergence to a specific loss value; however, this shows that our framework may cover cases where the convergence rate is heterogeneous across iterations. Finally, we note that the rate is always guaranteed to be less than one.

**Remark 5.5** (**Important differences with respect to networks without WeightNorm**). Theorem 5.1 differentiates from (Banerjee et al., 2023, Theorem 5.3) in that (i) it is *deterministic*, i.e., it is not stated as holding with high probability; (ii) does not have all the weights of the network defined over a particular neighborhood (with the exception of the output linear layer). Indeed, these differences also hold for all the previously derived results in Section 4 and Section 5. We summarize the comparison between our results and those for networks without WeightNorm by Banerjee et al. (2023) in Table 1 from Section D of the appendix. □

**Remark 5.6** (**About the radius $\rho_2$ in Assumption 3**). We prove in Proposition C.2 of the appendix that, for the case of $\phi(0) = 0$, $\|\theta_{t+1} - \theta_t\|_2 \leq \eta_t \cdot \mathcal{O}\left((1 + \rho_1)^2 \left(1 + \frac{\sqrt{L}}{\sqrt{m}\|\bar{W}_t\|_2}\right)\right)$. Note that $\rho_2$ as in Assumption 3 could in principle depend on $t$, and thus we could set $\rho_2$ to be equal to the upper bound just shown for $\|\theta_{t+1} - \theta_t\|_2$. Remarkably, $\eta_t$ as chosen in Theorem 5.1 can be upper bounded by one[1] and if $\|\bar{W}_t\|_2$ has a uniform lower bound across $t$ (note that our experiments in Section 7 provide evidence for $\|\bar{W}_t\|_2 \geq \|\bar{W}_0\|_2$), then the upper bound of $\|\theta_{t+1} - \theta_t\|_2$ becomes a constant independent of $t$, and so does $\rho_2$ to satisfy **(A3.2)** in Assumption 3. □

**Remark 5.7** (**About gradient descent in Assumption 3**). Regarding **(A3.1)** in Assumption 3, we remark that we are *only* considering that the weight vector of the output layer is inside a neighborhood of its initialization value across iterations—no constraint exist on the weights from the hidden layers. Moreover, the radius $\rho_1$ of this neighborhood region can be set to be any arbitrary constant of order $\Theta(1)$ or even a polynomial of $L$; thus, $\rho_1$ is not assumed to be arbitrarily close to zero and the neighborhood is not restricted to be arbitrarily small. These arguments imply that our optimization guarantees are not restricted to be in the lazy training regime (Liu et al., 2020). We also point out that **(A3.1)** is a necessary assumption so that the linear output of the neural network, which has no normalization, remains bounded. Finally, we remark that **(A3.1)** is not stronger than other existing works where assumptions of closeness around initialization points are needed even on the hidden weights; e.g., see (Liu et al., 2020; 2022; Banerjee et al., 2023). □

---

[1]We can set $\beta_{\theta_t}$ so that $1 \leq \beta_{\theta_t}$ from (17) and so $\eta_t = \frac{1}{\beta_{\theta_t}} \leq 1$.

## 6  Generalization guarantees for WeightNorm

We establish generalization bounds for WeightNorm networks. Our bound is based on a uniform convergence argument by bounding the Rademacher complexity of functions of the form (1) as long as the last layer vector $\mathbf{v}$ stays within a ball of radius $\rho_1$—the same condition we used in our optimization analysis. The core of the analysis relies on the use of a contraction-like property across the network layers similar to (Golowich et al., 2018) and uses WeightNorm and the network's scaling to avoid exponential dependence on the depth and avoid any dependence on the width. All proofs are found in Section E of the appendix.

Consider the training set $S = \{(x_i, y_i) \sim \mathcal{D}, i \in [n]\}$, where $\mathcal{D}$ denotes the true but unknwon distribution. The training and population losses are respectively defined as

$$\mathcal{L}_S(\theta) := \frac{1}{n}\sum_{i=1}^{n} \ell(y_i, f(\theta; \mathbf{x}_i)) \quad \text{and} \quad \mathcal{L}_{\mathcal{D}}(\theta) := \mathbb{E}_{(X,Y)\sim\mathcal{D}}[\ell(Y, f(\theta; X))] .$$

We added the subscript $S$ to the empirical loss notation to explicitly denote its dependence on the training set $S$.

**Theorem 6.1** (**Generalization Bound for WeightNorm under Square Loss**). *Consider the square loss and the training set $S = \{(x_i, y_i) \overset{i.i.d.}{\sim} \mathcal{D}, i \in [n]\}$ and $|y| \le 1$ for any $y \sim \mathcal{D}_y$ with probability one. Under Assumptions 1 and 2, assuming the activation function satisfies $\phi(0) = 0$, with probability at least $(1 - \delta)$ over the choice of the training data $\mathbf{x}_i \sim \mathcal{D}_{\mathbf{x}}, i \in [n]$, for any WeightNorm network $f(\theta; \cdot)$ of the form (1) with any fixed $\theta \in \mathbb{R}^p$ and $\mathbf{v} \in B_{\rho_1}^{Euc}(\mathbf{v}_0)$, we have*

$$\mathcal{L}_D(\theta) - \mathcal{L}_S(\theta) \le 4(2 + \rho_1)(1 + \rho_1)\frac{\sqrt{2\log(2)L} + 1}{\sqrt{n}} + 2(1 + (1 + \rho_1)^2)\frac{\sqrt{2\log(2/\delta)}}{\sqrt{n}}. \tag{19}$$

**Remark 6.1** (**About the activation function**). Theorem 6.1, unlike our previous results in the paper, requires the smooth activation function to satisfy $\phi(0) = 0$. As mentioned earlier in Remarks 4.2 and 5.3, the case when $\phi(0) = 0$ improves our Hessian bounds and thus improves our optimization conditions by introducing a better dependence on the width $m$. We point out that commonly used activation functions such as the hyperbolic tangent function tanh and Gaussian Error Linear Unit (GELU) satisfy both $\phi(0) = 0$ and Assumption 1. □

**Remark 6.2** (**The independence from the width $m$ and the dependence on depth $L$**). The generalizaton bound (19) does not explicitly depend on the network's width, but explicitly depends on its depth. Setting $\rho_1 = \Theta(1)$, a choice which does not negatively affect our optimization results, results in a generalization bound of $\mathcal{O}(\sqrt{\frac{L}{n}})$. Effectively, for deeper networks, we would need to increase the number of samples at least linearly on the depth to improve our generalization bound—in practice, the number of samples is usually much larger than the network's depth. This is a benign dependence that WeightNorm allows compared to an exponential one found in networks without WeightNorm (Bartlett et al., 2017; Neyshabur et al., 2018; Golowich et al., 2018). □

## 7  Experimental results

We show empirical results to support our theory. In our experiments, the WeightNorm networks are fully connected with two hidden layers of equal width $m$, tanh activation function, and linear output layer. We do empirical evaluations on CIFAR-10 (Krizhevsky, 2009) and MNIST (Deng, 2012). Because of computational efficiency, we apply mini-batch stochastic gradient descent (SGD) with batch size 512 to optimize the WeightNorm networks under mean squared loss. In our experiments we are only concerned about the training of WeightNorm networks from an optimization perspective—the fact that the training error minimizes across epochs is enough for our purposes. Therefore, using a squared loss with a linear output in our experiments suffices—as opposed to cross-entropy loss with a softmax output layer. Our experiments were conducted on a computing cluster with AMD EPYC 7713 64-Core Processor and NVIDIA A100 Tensor Core GPU.

In our theory, the convergence rate as in Theorem 5.1 is directly proportional to the RSC parameter $\alpha_{\theta_t, \bar{\theta}_t}$, and it benefits from a larger positive $\alpha_{\theta_t, \bar{\theta}_t}$. According to equation (13), $\alpha_{\theta_t, \bar{\theta}_t}$ becomes larger under two

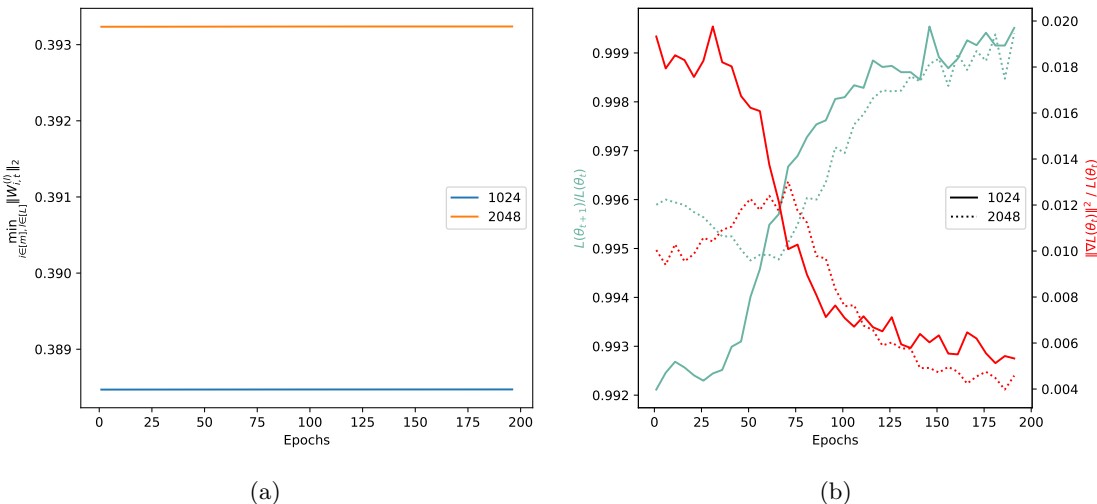

(a)                                                                 (b)

Figure 1: Training neural networks for two different widths $m \in \{1024, 2048\}$ on the CIFAR-10 dataset, with two hidden layers $L = 2$ of same width, with learning rate 0.001, and weights initialized independently from a uniform distribution $[-\frac{0.5}{\sqrt{m}}, \frac{0.5}{\sqrt{m}}]$. Each subfigure plots: (a): $\min_{\substack{i \in [m] \\ l \in [L]}} \left\| W_{i,t}^{(l)} \right\|_2$; (b): $\left\| \nabla \mathcal{L}(\theta_t) \right\|_2^2 / \mathcal{L}(\theta_t)$; where $t$ represents the number of iterations.

conditions: as the ratio $\|\nabla \mathcal{L}(\theta_t)\|_2^2 / \mathcal{L}(\theta_t)$ increases and/or the minimum weight vector norm $\min_{\substack{i \in [m] \\ l \in [L]}} \left\| W_{i,t}^{(l)} \right\|_2$ increases. Then, the relevant question is: *Can we provide empirical evidence of these two conditions benefiting convergence during training?*

We provide an affirmative answer in our simulations. We present our experiments on CIFAR-10 in Figure 1 and on MNIST in Figure 2.

We first focus on the effect of the ratio $\|\nabla \mathcal{L}(\theta_t)\|_2^2 / \mathcal{L}(\theta_t)$ on the RSC parameter. In Figure 1(a), we show that the values for the minimum weight vector stay nearly constant across epochs for both networks (in reality, they increase in value very slowly), i.e., the second term in the RSC parameter $\alpha_{\theta_t}$ (see equation (13)) basically remains stable. Therefore, according to our theory, the convergence speed should mostly depend on the ratio $\|\nabla \mathcal{L}(\theta_t)\|_2^2 / \mathcal{L}(\theta_t)$, i.e., the first term in the RSC parameter $\alpha_{\theta_t}$ (see equation (13)). Indeed, this is expressed in Figure 1(b): the curve of the ratio $\|\nabla \mathcal{L}(\theta_t)\|_2^2 / \mathcal{L}(\theta_t)$ has a decreasing trend towards the end of training for both networks (starting at around epoch 75) while the curve of the loss ratio between two consecutive epochs $\mathcal{L}(\theta_{t+1})/\mathcal{L}(\theta_t)$ has an increasing trend. Therefore, since an increase in the ratio $\mathcal{L}(\theta_{t+1})/\mathcal{L}(\theta_t)$ corresponds to a decrease on convergence speed, both networks have a slower convergence than they had at the initial epochs.

Next, we focus on the effect of the minimum weight vector norm on the RSC parameter. According to our theory, if the two networks in Figure 1 have the same value for the ratio $\|\nabla \mathcal{L}(\theta_t)\|_2^2 / \mathcal{L}(\theta_t)$, the network with the larger minimum weight vector norm will have a larger RSC parameter (see equation (13)), and so we would expect such network to have a faster convergence. Indeed, we observe this in our experiment: when the red solid curve crosses the dashed red curve (around epochs 70 and 115) in Figure 1(b), the two networks have the same value for $\|\nabla \mathcal{L}(\theta_t)\|_2^2 / \mathcal{L}(\theta_t)$. At this crossing, Figure 1(b) shows that the network with larger width—and so with larger minimum weight vector norm—converges faster afterwards (i.e., the loss ratio $\mathcal{L}(\theta_{t+1})/\mathcal{L}(\theta_t)$ for the green dashed curve has smaller values than the green solid curve).

We present similar results on the MNIST dataset in Figure 2. For example, similar to Figure 1, the minimum weight vector norms for both networks in Figure 2 are practically stable, which results in the convergence speed changing according to the changes in the ratio $\|\nabla \mathcal{L}(\theta_t)\|_2^2 / \mathcal{L}(\theta_t)$. Interestingly, the two curves of the ratio $\|\nabla \mathcal{L}(\theta_t)\|_2^2 / \mathcal{L}(\theta_t)$ do not cross, but they become very close on value around epoch 25. After this, we

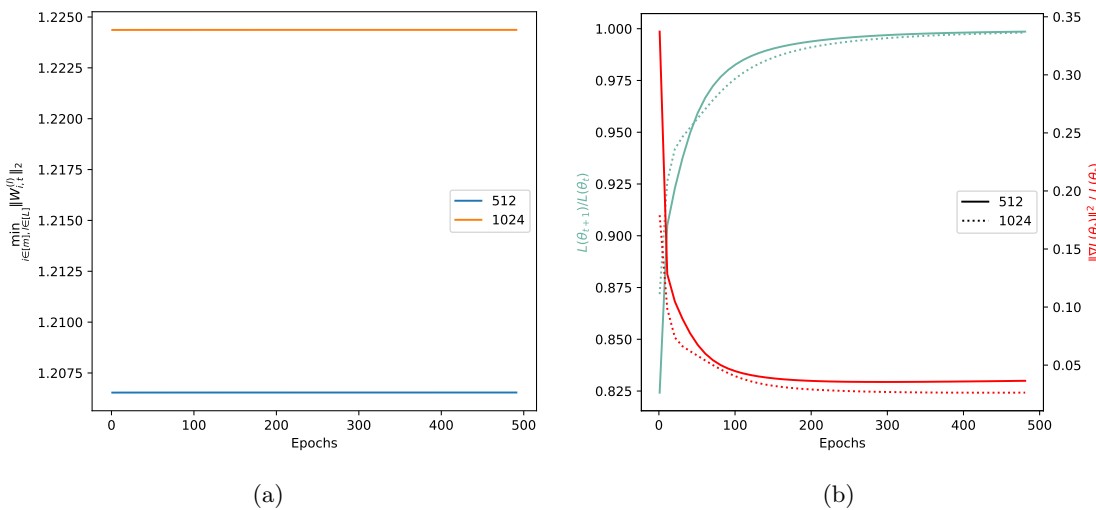

(a)                                                      (b)

Figure 2: Training neural networks for two different widths $m \in \{512, 1024\}$ on the MNIST dataset, with two hidden layers $L = 2$ of same width, with learning rate 0.001 and the weights are initialized with a uniform distribution $[-\frac{5}{\sqrt{m}}, \frac{5}{\sqrt{m}}]$. Each subfigure plots: (a): $\min_{\substack{i \in [m] \\ l \in [L]}} \left\| W_{i,t}^{(l)} \right\|_2$; (b): $\left\| \nabla \mathcal{L}(\theta_t) \right\|_2^2 / \mathcal{L}(\theta_t)$; where $t$ represents the number of iterations.

observe that the wider network, which is the network with the larger minimum weight vector norm, keeps achieving faster convergence for the rest of epochs despite having lower values of the ratio $\|\nabla \mathcal{L}(\theta_t)\|_2^2 / \mathcal{L}(\theta_t)$.

# 8 Conclusion

We formally analyzed the training and generalization of deep neural networks with WeightNorm under smooth activations. For this we needed to characterize a series of bounds which highlighted the dependence of our results on the depth, width, and minimum weight vector norm. Our simulations showed the possibility that our training guarantees hold in practice. We hope our paper will elicit further work on the theoretical foundation for other types of normalization whose connection to training and generalization are not well explored.

### Acknowledgments

Part of the work was done when Pedro Cisneros-Velarde was affiliated with the University of Illinois Urbana-Champaign and was concluded during his current affiliation with VMware Research. The work was supported in part by the National Science Foundation (NSF) through awards IIS 21-31335, OAC 21-30835, DBI 20-21898, as well as a C3.ai research award. Sanmi Koyejo acknowledges support by NSF 2046795 and 2205329, NIFA award 2020-67021-32799, the Alfred P. Sloan Foundation, and Google Inc.

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

## A  Spectral norm of the Hessian for WeightNorm

We establish the main theorem from Section 4.

**Theorem 4.1 (Hessian bound for WeightNorm).** *Under Assumptions 1 and 2, for any $\theta \in \mathbb{R}^p$ with $\mathbf{v} \in B_{\rho_1}^{\mathrm{Euc}}(\mathbf{v}_0)$, and any $\mathbf{x}_i, i \in [n]$, we have*

$$\left\| \nabla_\theta^2 f(\theta; \mathbf{x}_i) \right\|_2 \leq \mathcal{O}\left( \frac{(1+\rho_1)L^3(\frac{1}{\sqrt{m}} + L^2 \max\{|\phi(0)|, |\phi(0)|^2\})}{\min\left\{ \|\bar{W}\|_2, \|\bar{W}\|_2^2 \right\}} \right). \tag{4}$$

### A.1  Starting the Proof

For an order-3 tensor $T \in \mathbb{R}^{d_1 \times d_2 \times d_3}$ we define the operator $\|\cdot\|_{2,2,1}$ as follows,

$$\|T\|_{2,2,1} := \sup_{\|\mathbf{x}\|_2 = \|\mathbf{z}\|_2 = 1} \sum_{k=1}^{d_3} \left| \sum_{i=1}^{d_1} \sum_{j=1}^{d_2} T_{ijk} x_i z_j \right|, \quad \mathbf{x} \in \mathbb{R}^{d_1}, \mathbf{z} \in \mathbb{R}^{d_2}. \tag{20}$$

Let us consider any $\mathbf{x} \in \mathbb{R}^d$ such that $\|\mathbf{x}\|_2 = \sqrt{d}$. Our analysis follows the general structure developed in (Liu et al., 2020, Theorem 3.1). We first observe that

$$\left\| \frac{\partial^2 f(\theta; \mathbf{x})}{(\partial \theta)^2} \right\|_2 \leq \sum_{l_1, l_2 = 1}^{L} \left\| \frac{\partial^2 f(\theta; \mathbf{x})}{\partial W^{(l_1)} \partial W^{(l_2)}} \right\|_2 + 2 \sum_{l_1 = 1}^{L} \left\| \frac{\partial^2 f(\theta; \mathbf{x})}{\partial W^{(l_1)} \partial W^{(L+1)}} \right\|_2 + \left\| \frac{\partial^2 f(\theta; \mathbf{x})}{(\partial W^{(L+1)})^2} \right\|_2$$

$$= \sum_{l_1, l_2 = 1}^{L} \left\| \frac{\partial^2 f(\theta; \mathbf{x})}{\partial W^{(l_1)} \partial W^{(l_2)}} \right\|_2 + 2 \sum_{l_1 = 1}^{L} \left\| \frac{\partial^2 f(\theta; \mathbf{x})}{\partial W^{(l_1)} \partial W^{(L+1)}} \right\|_2, \tag{21}$$

where the equality follows by noticing that, since $W^{(L+1)} = \mathbf{v}$, $\left( \frac{\partial^2 f(\theta; \mathbf{x})}{(\partial W^{(L+1)})^2} \right)_{i,j} = 0$ for $(i,j) \in [m] \times [m]$, and so $\left\| \frac{\partial^2 f(\theta; \mathbf{x})}{(\partial W^{(L+1)})^2} \right\|_2 = 0$.

From now on, for simplicity of notation, we will obviate the dependency on $\mathbf{x}$ when necessary.

Now, again from the proof of (Liu et al., 2020, Theorem 3.1), for $1 \leq l_1 \leq l_2 \leq L$,

$$\left\| \frac{\partial^2 f(\theta)}{\partial W^{(l_1)} \partial W^{(l_2)}} \right\|_2 = \left\| \frac{\partial^2 \alpha^{(l_1)}}{(\partial W^{(l_1)})^2} \right\|_{2,2,1} \left\| \frac{\partial f}{\partial \alpha^{(l_1)}} \right\|_\infty \mathbb{1}_{[l_1 = l_2]}$$

$$+ \mathbb{1}_{[l_2 > l_1]} \left\| \frac{\partial \alpha^{(1_1)}}{\partial W^{(l_1)}} \right\|_2 \prod_{l = l_1 + 1}^{l_2 - 1} \left\| \frac{\partial \alpha^{(l)}}{\partial \alpha^{(l-1)}} \right\|_2 \left\| \frac{\partial^2 \alpha^{(l_2)}}{\partial \alpha^{(l_2 - 1)} \partial W^{(l_2)}} \right\|_{2,2,1} \left\| \frac{\partial f}{\partial \alpha^{(l_2)}} \right\|_\infty$$

$$+ \mathbb{1}_{[l_2 > l_1]} \left\| \frac{\partial \alpha^{(l_1)}}{\partial W^{(l_1)}} \right\|_2 \left\| \frac{\partial \alpha^{(l_2)}}{\partial W^{(l_2)}} \right\|_2 \sum_{l = l_2 + 1}^{L} \prod_{l' = l_1 + 1}^{l} \left\| \frac{\partial \alpha^{(l')}}{\partial \alpha^{(l'-1)}} \right\|_2 \left\| \frac{\partial^2 \alpha^{(l)}}{(\partial \alpha^{(l-1)})^2} \right\|_{2,2,1}$$

$$\times \prod_{\hat{l} = l_2 + 1}^{l} \left\| \frac{\partial \alpha^{(\hat{l})}}{\partial \alpha^{(\hat{l}-1)}} \right\|_2 \left\| \frac{\partial f}{\partial \alpha^{(l)}} \right\|_\infty \tag{22}$$

We use the following conventional notation: if a product has the form $\prod_{i=b}^{a} z_i$ where $a < b$ given some real scalars $\{z_i\}_{i=a}^{n}$ where $n \geq b$, then $\prod_{i=b}^{a} z_i = 0$.

Now we proceed to obtain upper bounds for all the required terms in the previous expression.

## A.2 Upper bounds of the $L_2$-norms of $\alpha^{(l)}$

**Lemma A.1.** *Consider Assumption 1. For any $l \in [L]$, any $\mathbf{x} \in \mathbb{R}^d$ such that $\|\mathbf{x}\|_2 = 1$, and any $\theta \in \mathbb{R}^p$, we have*

$$\|\alpha^{(l)}(\mathbf{x})\|_2 \leq 1 + l|\phi(0)|\sqrt{m} \ .$$

*Proof.* Following (Allen-Zhu et al., 2019; Liu et al., 2020), we prove the result by induction. First, since $\|\mathbf{x}\|_2 = 1$ by assumption, we have $\|\alpha^{(0)}(\mathbf{x})\|_2 = 1$. Since $\phi$ is 1-Lipschitz,

$$\left\|\phi\left(\left[\frac{1}{\sqrt{m}}\frac{(W_i^{(1)})^\top}{\left\|W_i^{(1)}\right\|_2}\alpha^{(0)}\right]_i\right)\right\|_2 - \|\phi(\mathbf{0})\|_2 \leq \left\|\phi\left(\frac{1}{\sqrt{m}}\left[\frac{(W_i^{(1)})^\top}{\left\|W_i^{(1)}\right\|_2}\alpha^{(0)}\right]_i\right) - \phi(\mathbf{0})\right\|_2$$

$$\leq \left\|\left[\frac{1}{\sqrt{m}}\frac{(W_i^{(1)})^\top}{\left\|W_i^{(1)}\right\|_2}\alpha^{(0)}\right]_i\right\|_2 \ ,$$

so that

$$\|\alpha^{(1)}\|_2 = \left\|\phi\left(\left[\frac{1}{\sqrt{m}}\frac{(W_i^{(1)})^\top}{\left\|W_i^{(1)}\right\|_2}\alpha^{(0)}\right]_i\right)\right\|_2 \leq \left\|\left[\frac{1}{\sqrt{m}}\frac{(W_i^{(1)})^\top}{\left\|W_i^{(1)}\right\|_2}\alpha^{(0)}\right]_i\right\|_2 + \|\phi(\mathbf{0})\|_2$$

$$= \frac{1}{\sqrt{m}}\sqrt{\sum_{i=1}^{m}\left(\frac{(W_i^{(1)})^\top}{\left\|W_i^{(1)}\right\|_2}\alpha^{(0)}\right)^2} + |\phi(0)|\sqrt{m}$$

$$\overset{(a)}{\leq} \frac{1}{\sqrt{m}}\sqrt{\sum_{i=1}^{m}\left\|\alpha^{(0)}\right\|_2^2} + |\phi(0)|\sqrt{m}$$

$$= 1 + |\phi(0)|\sqrt{m} \ ,$$

where (a) follows from Chauchy-Schwarz. For the inductive step, we assume that for layer $l-1$, we have

$$\|\alpha^{(l-1)}\|_2 \leq (1 + (l-1)|\phi(0)|\sqrt{m}).$$

Since $\phi$ is 1-Lipschitz, for layer $l$, we can similarly conclude that

$$\left\|\phi\left(\left[\frac{1}{\sqrt{m}}\frac{(W_i^{(l)})^\top}{\left\|W_i^{(l)}\right\|_2}\alpha^{(l-1)}\right]_i\right)\right\|_2 - \|\phi(\mathbf{0})\|_2 \leq \left\|\left[\frac{1}{\sqrt{m}}\frac{(W_i^{(l)})^\top}{\left\|W_i^{(l)}\right\|_2}\alpha^{(l-1)}\right]_i\right\|_2 \ ,$$

so that

$$
\begin{aligned}
\|\alpha^{(l)}\|_2 = \left\| \phi\left( \left[ \frac{1}{\sqrt{m}} \frac{(W_i^{(l)})^\top}{\left\|W_i^{(l)}\right\|_2} \alpha^{(l-1)} \right]_i \right) \right\|_2 &\leq \left\| \left[ \frac{1}{\sqrt{m}} \frac{(W_i^{(l)})^\top}{\left\|W_i^{(l)}\right\|_2} \alpha^{(l-1)} \right]_i \right\|_2 + \|\phi(\mathbf{0})\|_2 \\
&\leq \frac{1}{\sqrt{m}} \sqrt{\sum_{i=1}^m \left\|\alpha^{(l-1)}\right\|_2^2} + |\phi(0)|\sqrt{m} \\
&\overset{(a)}{\leq} (1 + (l-1)|\phi(0)|\sqrt{m}) + |\phi(0)|\sqrt{m} \\
&= 1 + l|\phi(0)|\sqrt{m},
\end{aligned}
$$

where (a) follows from the inductive step. This completes the proof. $\qquad\square$

### A.3 Upper bounds on the spectral norms of $\frac{\partial \alpha^{(l)}}{\partial W^{(l)}}$ and $\frac{\partial \alpha^{(l)}}{\partial \alpha^{(l-1)}}$

**Lemma A.2.** *Consider Assumption 1. For any $l \in \{2, \ldots, L\}$, any $\mathbf{x} \in \mathbb{R}^d$ such that $\|\mathbf{x}\|_2 = \sqrt{d}$, and any $\theta \in \mathbb{R}^p$, we have*

$$
\left\| \frac{\partial \alpha^{(l)}(\mathbf{x})}{\partial \alpha^{(l-1)}(\mathbf{x})} \right\|_2 \leq 1 , \tag{23}
$$

*and*

$$
\left\| \frac{\partial \alpha^{(1)}(\mathbf{x})}{\partial \alpha^{(0)}(\mathbf{x})} \right\|_2 = \left\| \frac{\partial \alpha^{(1)}(\mathbf{x})}{\partial \mathbf{x}} \right\|_2 \leq 1 . \tag{24}
$$

*Additionally, under Assumption 2 with $\mathbf{v} \in B_{\rho_1}^{\mathrm{Euc}}(\mathbf{v}_0)$,*

$$
\left\| \frac{\partial f(\mathbf{x})}{\partial \alpha^{(L)}(\mathbf{x})} \right\|_2 \leq 1 + \rho_1. \tag{25}
$$

*Proof.* By definition, we have

$$
\left[ \frac{\partial \alpha^{(l)}}{\partial \alpha^{(l-1)}} \right]_{i,j} = \frac{1}{\sqrt{m}} \phi'(\tilde{\alpha}_i^{(l)}) \frac{W_{ij}^{(l)}}{\left\|W_i^{(l)}\right\|_2} . \tag{26}
$$

Then, we have that for $2 \leq l \leq L$,

$$
\begin{aligned}
\left\| \frac{\partial \alpha^{(l)}}{\partial \alpha^{(l-1)}} \right\|_2^2 &= \sup_{\|\mathbf{v}\|_2=1} \frac{1}{m} \sum_{i=1}^m \left( \phi'(\tilde{\alpha}_i^{(l)}) \sum_{j=1}^m \frac{W_{ij}^{(l)}}{\left\|W_i^{(l)}\right\|_2} \mathbf{v}_j \right)^2 \\
&\overset{(a)}{\leq} \frac{1}{m} \sup_{\|\mathbf{v}\|_2=1} \sum_{i=1}^m \left( \frac{(W_i^{(l)})^\top \mathbf{v}}{\left\|W_i^{(l)}\right\|_2} \right)^2 \\
&\leq \frac{1}{m} \sup_{\|\mathbf{v}\|_2=1} \sum_{i=1}^m \frac{\left\|W_i^{(l)}\right\|_2^2 \|\mathbf{v}\|_2^2}{\left\|W_i^{(l)}\right\|_2^2} \\
&= \frac{1}{m} \cdot m \\
&= 1 ,
\end{aligned}
$$

where (a) follows from $\phi$ being 1-Lipschitz by Assumption 1. Finally, it is easy to follow the same proof and show that $\left\| \frac{\partial \alpha^{(1)}}{\partial \alpha^{(0)}} \right\|_2^2 \leq 1$. This completes the proof for equation (24).

Finally, for proving equation (25), note that $\left\| \frac{\partial f}{\partial \alpha^{(L)}} \right\|_2 = \|\mathbf{v}\|_2 \leq \|\mathbf{v}_0\|_2 + \|\mathbf{v} - \mathbf{v}_0\|_2 \leq (1 + \rho_1)$. $\qquad \square$

**Lemma A.3.** *Consider Assumption 1. For any $l \in [L]$, any $\mathbf{x} \in \mathbb{R}^d$ such that $\|\mathbf{x}\|_2 = 1$, and any $\theta \in \mathbb{R}^p$, we have*

$$\left\| \frac{\partial \alpha^{(l)}(\mathbf{x})}{\partial W^{(l)}} \right\|_2 \leq \frac{2}{\min_{i \in [m]} \left\| W_i^{(l)} \right\|_2} \left( \frac{1}{\sqrt{m}} + (l-1)|\phi(0)| \right). \tag{27}$$

*Proof.* Note that the parameter vector $\mathbf{w}^{(l)} = \text{vec}(W^{(l)})$ and can be indexed with $j \in [m]$ and $j' \in [d]$ when $l = 1$ and $j' \in [m]$ when $l \geq 2$. We will do the analysis for $l \geq 2$ since a similar analysis holds for the case $l = 1$. Thus, we have

$$
\begin{aligned}
\left[ \frac{\partial \alpha^{(l)}}{\partial W^{(l)}} \right]_{i,jj'} &= \left[ \frac{\partial \alpha^{(l)}}{\partial \mathbf{w}^{(l)}} \right]_{i,jj'} \\
&= \phi'(\tilde{\alpha}_i^{(l)}) \frac{\partial}{\partial W_{j,j'}} \left( \frac{1}{\sqrt{m}} \frac{(W_i^{(l)})^\top}{\left\| W_i^{(l)} \right\|_2} \alpha^{(l-1)} \right) \mathbb{1}_{[j=i]} \\
&= \phi'(\tilde{\alpha}_i^{(l)}) \frac{1}{\sqrt{m}} \left( \alpha_{j'}^{(l-1)} - \frac{((W_i^{(l)})^\top \alpha^{(l-1)}) W_{ij'}^{(l)}}{\left\| W_i^{(l)} \right\|_2^2} \right) \frac{1}{\left\| W_i^{(l)} \right\|_2} \mathbb{1}_{[j=i]} \\
&= \underbrace{\frac{\phi'(\tilde{\alpha}_i^{(l)})}{\sqrt{m}} \frac{\alpha_{j'}^{(l-1)}}{\left\| W_i^{(l)} \right\|_2} \mathbb{1}_{[j=i]}}_{(I)} - \underbrace{\frac{\phi'(\tilde{\alpha}_i^{(l)})}{\sqrt{m}} \frac{((W_i^{(l)})^\top \alpha^{(l-1)}) W_{ij'}^{(l)}}{\left\| W_i^{(l)} \right\|_2^3} \mathbb{1}_{[j=i]}}_{(II)} .
\end{aligned}
\tag{28}
$$

where we used the fact that

$$\frac{\partial}{\partial W_{jj'}} \left( \frac{1}{\left\| W_i^{(l)} \right\|_2} \right) = -\frac{W_{ij'}}{\left\| W_i^{(l)} \right\|_2^3} \mathbb{1}_{[j=i]}.$$

Defining the matrices $A, B \in \mathbb{R}^{m \times m^2}$ by $[A]_{i,jj'} := (I)$ and $[B]_{i,jj'} := (II)$, we observe that $\frac{\partial \alpha^{(l)}}{\partial W^{(l)}} = A - B$.

Then, noting that $\frac{\partial \alpha^{(l)}}{\partial W^{(l)}} \in \mathbb{R}^{m \times m^2}$ and $\|\mathbf{V}\|_F = \|\text{vec}(\mathbf{V})\|_2$ for any matrix $\mathbf{V}$, we have

$$
\begin{aligned}
\|A\|_2^2 &= \sup_{\|\mathbf{V}\|_F=1} \sum_{i=1}^{m} \left( \sum_{j,j'=1}^{m} \frac{\phi'(\tilde{\alpha}_i^{(l)})}{\sqrt{m}} \frac{\alpha_{j'}^{(l-1)}}{\left\|W_i^{(l)}\right\|_2} \mathbb{1}_{[j=i]} \mathbf{V}_{jj'} \right)^2 \\
&= \sup_{\|\mathbf{V}\|_F=1} \sum_{i=1}^{m} \left( \sum_{j'=1}^{m} \frac{\phi'(\tilde{\alpha}_i^{(l)})}{\sqrt{m}} \frac{\alpha_{j'}^{(l-1)}}{\left\|W_i^{(l)}\right\|_2} \mathbf{V}_{ij'} \right)^2 \\
&\leq \frac{1}{m} \sup_{\|\mathbf{V}\|_F=1} \sum_{i=1}^{m} \left( \sum_{j'=1}^{m} \frac{\alpha_{j'}^{(l-1)}}{\left\|W_i^{(l)}\right\|_2} \mathbf{V}_{ij'} \right)^2 \\
&\leq \frac{1}{m} \sup_{\|\mathbf{V}\|_F=1} \sum_{i=1}^{m} \frac{1}{\left\|W_i^{(l)}\right\|_2^2} \left\|\alpha^{(l-1)}\right\|_2^2 \|\mathbf{V}_{i*}\|_2^2 \\
&\leq \frac{\left\|\alpha^{(l-1)}\right\|_2^2}{m \cdot \min_{i \in [m]} \left\|W_i^{(l)}\right\|_2^2} \sup_{\|\mathbf{V}\|_F=1} \sum_{i=1}^{m} \|\mathbf{V}_{i*}\|_2^2 \\
&= \frac{\left\|\alpha^{(l-1)}\right\|_2^2}{m \cdot \min_{i \in [m]} \left\|W_i^{(l)}\right\|_2^2} \\
&\overset{(a)}{\leq} \frac{1}{\min_{i \in [m]} \left\|W_i^{(l)}\right\|_2^2} \left( \frac{1}{\sqrt{m}} + (l-1)|\phi(0)| \right)^2
\end{aligned}
$$

where (a) follows from Lemma A.1.

Similarly,

$$
\begin{aligned}
\|B\|_2^2 &= \sup_{\|\mathbf{V}\|_F=1} \sum_{i=1}^m \left( \sum_{j,j'=1}^m \frac{\phi'(\tilde{\alpha}_i^{(l)})}{\sqrt{m}} \frac{((W_i^{(l)})^\top \alpha^{(l-1)}) W_{ij'}^{(l)}}{\left\| W_i^{(l)} \right\|_2^3} \mathbb{1}_{[j=i]} \mathbf{V}_{jj'} \right)^2 \\
&= \sup_{\|\mathbf{V}\|_F=1} \sum_{i=1}^m \left( \frac{\phi'(\tilde{\alpha}_i^{(l)})}{\sqrt{m}} \frac{((W_i^{(l)})^\top \alpha^{(l-1)})}{\left\| W_i^{(l)} \right\|_2^3} \right)^2 \left( \sum_{j'=1}^m W_{ij'}^{(l)} \mathbf{V}_{ij'} \right)^2 \\
&\le \frac{1}{m} \sup_{\|X\|_F=1} \sum_{i=1}^m \frac{((W_i^{(l)})^\top \alpha^{(l-1)})^2}{\left\| W_i^{(l)} \right\|_2^6} \left( (W_i^{(l)})^\top X_{i*} \right)^2 \\
&\le \frac{1}{m} \sup_{\|\mathbf{V}\|_F=1} \sum_{i=1}^m \frac{((W_i^{(l)})^\top \alpha^{(l-1)})^2}{\left\| W_i^{(l)} \right\|_2^6} \left\| W_i^{(l)} \right\|_2^2 \|\mathbf{V}_{i*}\|_2^2 \\
&\le \frac{1}{m} \sup_{\|\mathbf{V}\|_F=1} \sum_{i=1}^m \frac{\left\| W_i^{(l)} \right\|_2^2}{\left\| W_i^{(l)} \right\|_2^6} \left\| \alpha^{(l-1)} \right\|_2^2 \left\| W_i^{(l)} \right\|_2^2 \|\mathbf{V}_{i*}\|_2^2 \\
&= \frac{1}{m} \sup_{\|\mathbf{V}\|_F=1} \sum_{i=1}^m \frac{\left\| \alpha^{(l-1)} \right\|_2^2}{\left\| W_i^{(l)} \right\|_2^2} \|\mathbf{V}_{i,*}\|_2^2 \\
&\le \frac{\left\| \alpha^{(l-1)} \right\|_2^2}{m \cdot \min_{i \in [m]} \left\| W_i^{(l)} \right\|_2^2} \\
&\overset{(a)}{\le} \frac{1}{\min_{i \in [m]} \left\| W_i^{(l)} \right\|_2^2} \left( \frac{1}{\sqrt{m}} + (l-1)|\phi(0)| \right)^2
\end{aligned}
$$

where (a) follows from Lemma A.1.

Using the recent derived results, we have

$$
\left\| \frac{\partial \alpha^{(l)}}{\partial W^{(l)}} \right\|_2 \le \|A\|_2 + \|B\|_2 \le \frac{2}{\min_{i \in [m]} \left\| W_i^{(l)} \right\|_2} \left( \frac{1}{\sqrt{m}} + (l-1)|\phi(0)| \right).
$$

That completes the proof. $\qquad\square$

## A.4   Bound on $\|\cdot\|_{2,2,1}$

**Lemma A.4.** *Consider Assumption 1. For any $\mathbf{x} \in \mathbb{R}^d$ such that $\|\mathbf{x}\|_2 = 1$ and any $\theta \in \mathbb{R}^p$, each of the following inequalities hold,*

$$
\left\| \frac{\partial^2 \alpha^{(l)}(\mathbf{x})}{(\partial \alpha^{(l-1)}(\mathbf{x}))^2} \right\|_{2,2,1} \le \beta_\phi, \tag{29}
$$

$$
\left\| \frac{\partial^2 \alpha^{(l)}(\mathbf{x})}{\partial \alpha^{(l-1)}(\mathbf{x}) \partial W^{(l)}} \right\|_{2,2,1} \le \frac{2(\beta_\phi(\frac{1}{\sqrt{m}} + (l-1)|\phi(0)|) + 1)}{\min_{i \in [m]} \left\| W_i^{(l)} \right\|_2} \tag{30}
$$

*for $l = 2, \ldots, L$; and*

$$
\left\| \frac{\partial^2 \alpha^{(l)}(\mathbf{x})}{(\partial W^{(l)})^2} \right\|_{2,2,1} \le \frac{2(2\beta_\phi(\frac{1}{\sqrt{m}} + (l-1)|\phi(0)|) + 3)(\frac{1}{\sqrt{m}} + (l-1)|\phi(0)|)}{\min_{i \in [m]} \left\| W_i^{(l)} \right\|_2^2}, \tag{31}
$$

*for $l \in [L]$.*

*Proof.* For the inequality (29), note that from (26) we obtain

$$\left(\frac{\partial^2 \alpha^{(l)}}{(\partial \alpha^{(l-1)})^2}\right)_{i,j,k} = \frac{1}{m}\phi''(\tilde{\alpha}_i^{(l)})\frac{W_{ik}^{(l)}W_{ij}^{(l)}}{\left\|W_i^{(l)}\right\|_2^2},$$

and so

$$\begin{aligned}
\left\|\frac{\partial^2 \alpha^{(l)}}{(\partial \alpha^{(l-1)})^2}\right\|_{2,2,1} &= \sup_{\|\mathbf{v_1}\|_2=\|\mathbf{v_2}\|_2=1}\frac{1}{m}\sum_{i=1}^{m}\left|\phi''(\tilde{\alpha}_i^{(l)})\frac{((W_i^{(l)})^\top \mathbf{v_1})((W_i^{(l)})^\top \mathbf{v_2})}{\left\|W_i^{(l)}\right\|_2^2}\right| \\
&\leq \frac{\beta_\phi}{m}\sup_{\|\mathbf{v_1}\|_2=\|\mathbf{v_2}\|_2=1}\sum_{i=1}^{m}\left|\frac{((W_i^{(l)})^\top \mathbf{v_1})((W_i^{(l)})^\top \mathbf{v_2})}{\left\|W_i^{(l)}\right\|_2^2}\right| \\
&\leq \frac{\beta_\phi}{m}\sup_{\|\mathbf{v_1}\|_2=\|\mathbf{v_2}\|_2=1}\sum_{i=1}^{m}\left|\frac{\left\|W_i^{(l)}\right\|_2\|\mathbf{v_1}\|_2\cdot\left\|W_i^{(l)}\right\|\|\mathbf{v_2}\|}{\left\|W_i^{(l)}\right\|_2^2}\right| \\
&= \frac{\beta_\phi}{m}\cdot m.
\end{aligned}$$

$$(32)$$

For the inequality (30), carefully following the chain rule in (26) we obtain

$$\begin{aligned}
\left(\frac{\partial^2 \alpha^{(l)}}{\partial \alpha^{(l-1)}\partial W^{(l)}}\right)_{i,j,kk'} &= \phi''(\tilde{\alpha}_i^{(l)})\left(\frac{W_{ij}^{(l)}}{\sqrt{m}\left\|W_i^{(l)}\right\|_2}\right)\left(\frac{\alpha_{k'}^{(l-1)}}{\sqrt{m}\left\|W_i^{(l)}\right\|_2}\mathbb{1}_{[k=i]}\right. \\
&\qquad\left. -\frac{((W_i^{(l)})^\top \alpha^{(l-1)})W_{ik'}^{(l)}}{\sqrt{m}\left\|W_i^{(l)}\right\|_2^3}\mathbb{1}_{[k=i]}\right) \\
&\qquad + \phi'(\tilde{\alpha}_i^{(l)})\frac{1}{\sqrt{m}}\mathbb{1}_{[k=i]}\left(-\frac{W_{ik'}^{(l)}W_{ij}^{(l)}}{\sqrt{m}\left\|W_i^{(l)}\right\|_2^3} + \frac{1}{\left\|W_i^{(l)}\right\|_2}\mathbb{1}_{[k'=j]}\right).
\end{aligned}$$

Then, we have

$$
\left\|\frac{\partial^2 \alpha^{(l)}}{\partial \alpha^{(l-1)} \partial W^{(l)}}\right\|_{2,2,1}
$$

$$
= \sup_{\|\mathbf{v}_1\|_2 = \|\mathbf{V}_2\|_F = 1} \sum_{i=1}^m \left| \sum_{j=1}^m \sum_{k'=1}^m \phi''(\tilde{\alpha}_i^{(l)}) \left( \left( \frac{W_{ij}^{(l)}}{\sqrt{m} \left\| W_i^{(l)} \right\|_2} \right) \left( \frac{\alpha_{k'}^{(l-1)}}{\sqrt{m} \left\| W_i^{(l)} \right\|_2} \right) \right. \right.
$$

$$
- \frac{((W_i^{(l)})^\top \alpha^{(l-1)}) W_{ik'}^{(l)}}{\sqrt{m} \left\| W_i^{(l)} \right\|_2^3}\Bigg)
$$

$$
\left. \left. + \phi'(\tilde{\alpha}_i^{(l)}) \frac{1}{\sqrt{m}} \left( -\frac{W_{ik'}^{(l)} W_{ij}^{(l)}}{\sqrt{m} \left\| W_i^{(l)} \right\|_2^3} + \frac{1}{\left\| W_i^{(l)} \right\|_2} \mathbb{1}_{[k'=j]} \right) \right) \mathbf{v}_{1,j} \mathbf{V}_{2,ik'} \right|
$$

$$
= \sup_{\|\mathbf{v}_1\|_2 = \|\mathbf{V}_2\|_F = 1} \sum_{i=1}^m \left| \sum_{j=1}^m \sum_{k'=1}^m \frac{1}{m} \phi''(\tilde{\alpha}_i^{(l)}) \frac{W_{ij}^{(l)}}{\left\| W_i^{(l)} \right\|_2^2} \alpha_{k'}^{(l-1)} \mathbf{v}_{1,j} \mathbf{V}_{2,ik'} \right.
$$

$$
- \sum_{j=1}^m \sum_{k'=1}^m \frac{1}{m} \phi''(\tilde{\alpha}_i^{(l)}) \frac{W_{ij}^{(l)}}{\left\| W_i^{(l)} \right\|_2} \frac{((W_i^{(l)})^\top \alpha^{(l-1)})}{\left\| W_i^{(l)} \right\|_2^3} W_{ik'}^{(l)} \mathbf{v}_{1,j} \mathbf{V}_{2,ik'} \tag{33}
$$

$$
\left. - \sum_{j=1}^m \sum_{k'=1}^m \frac{1}{\sqrt{m}} \phi'(\tilde{\alpha}_i^{(l)}) \frac{W_{ik'}^{(l)}}{\left\| W_i^{(l)} \right\|_2^3} W_{ij}^{(l)} \mathbf{v}_{1,j} \mathbf{V}_{2,ik'} + \sum_{j=1}^m \frac{1}{\sqrt{m}} \phi'(\tilde{\alpha}_i^{(l)}) \frac{1}{\left\| W_i^{(l)} \right\|_2} \mathbf{v}_{1,j} \mathbf{V}_{2,ij} \right|
$$

$$
\leq \sup_{\|\mathbf{v}_1\|_2 = \|\mathbf{V}_2\|_F = 1} \sum_{i=1}^m \left| \sum_{j=1}^m \sum_{k'=1}^m \frac{1}{m} \phi''(\tilde{\alpha}_i^{(l)}) \frac{W_{ij}^{(l)}}{\left\| W_i^{(l)} \right\|_2^2} \alpha_{k'}^{(l-1)} \mathbf{v}_{1,j} \mathbf{V}_{2,ik'} \right|
$$

$$
+ \sup_{\|\mathbf{v}_1\|_2 = \|\mathbf{V}_2\|_F = 1} \sum_{i=1}^m \left| \sum_{j=1}^m \sum_{k'=1}^m \frac{1}{m} \phi''(\tilde{\alpha}_i^{(l)}) \frac{W_{ij}^{(l)}}{\left\| W_i^{(l)} \right\|_2} \frac{((W_i^{(l)})^\top \alpha^{(l-1)})}{\left\| W_i^{(l)} \right\|_2^3} W_{ik'}^{(l)} \mathbf{v}_{1,j} \mathbf{V}_{2,ik'} \right|
$$

$$
+ \sup_{\|\mathbf{v}_1\|_2 = \|\mathbf{V}_2\|_F = 1} \sum_{i=1}^m \left| \sum_{j=1}^m \sum_{k'=1}^m \frac{1}{\sqrt{m}} \phi'(\tilde{\alpha}_i^{(l)}) \frac{W_{ik'}^{(l)}}{\left\| W_i^{(l)} \right\|_2^3} W_{ij}^{(l)} \mathbf{v}_{1,j} \mathbf{V}_{2,ik'} \right|
$$

$$
+ \sup_{\|\mathbf{v}_1\|_2 = \|\mathbf{V}_2\|_F = 1} \sum_{i=1}^m \left| \sum_{j=1}^m \frac{1}{\sqrt{m}} \phi'(\tilde{\alpha}_i^{(l)}) \frac{1}{\left\| W_i^{(l)} \right\|_2} \mathbf{v}_{1,j} \mathbf{V}_{2,ij} \right|
$$

Now we upper bound each of the terms in the last inequality. We analyze the first term,

$$
\sup_{\|\mathbf{v}_1\|_2=\|\mathbf{V}_2\|_F=1} \sum_{i=1}^{m} \left| \sum_{j=1}^{m} \sum_{k'=1}^{m} \frac{1}{m} \phi''(\tilde{\alpha}_i^{(l)}) \frac{W_{ij}^{(l)}}{\left\| W_i^{(l)} \right\|_2^2} \alpha_{k'}^{(l-1)} \mathbf{v}_{1,j} \mathbf{V}_{2,ik'} \right|
$$

$$
\leq \frac{\beta_\phi}{m} \sup_{\|\mathbf{v}_1\|_2=\|\mathbf{V}_2\|_F=1} \sum_{i=1}^{m} \left| \left( \sum_{j=1}^{m} \frac{W_{ij}^{(l)}}{\left\| W_i^{(l)} \right\|_2^2} \mathbf{v}_{1,j} \right) \left( \sum_{k'=1}^{m} \alpha_{k'}^{(l-1)} \mathbf{V}_{2,ik'} \right) \right|
$$

$$
\overset{(a)}{\leq} \frac{\beta_\phi}{m} \sup_{\|\mathbf{v}_1\|_2=\|\mathbf{V}_2\|_F=1} \sqrt{ \sum_{i=1}^{m} \frac{\|\mathbf{v}_1\|_2^2}{\left\| W_i^{(l)} \right\|_2^2} } \left\| \mathbf{V}_2 \alpha^{(l-1)} \right\|_2
$$

$$
\leq \frac{\beta_\phi}{m} \sup_{\|\mathbf{V}_2\|_F=1} \sqrt{ \sum_{i=1}^{m} \frac{1}{\left\| W_i^{(l)} \right\|_2^2} } \|\mathbf{V}_2\|_2 \left\| \alpha^{(l-1)} \right\|_2
$$

$$
\overset{(b)}{\leq} \frac{\beta_\phi}{m} \sqrt{ \sum_{i=1}^{m} \frac{1}{\left\| W_i^{(l)} \right\|_2^2} } \left\| \alpha^{(l-1)} \right\|_2
$$

$$
\overset{(c)}{\leq} \frac{\beta_\phi}{\sqrt{m}} \left( \frac{1}{\sqrt{m}} + (l-1)|\phi(0)| \right) \cdot \frac{1}{\min_{i\in[m]} \left\| W_i^{(l)} \right\|_2} \sqrt{m}
$$

$$
= \frac{\beta_\phi \left( \frac{1}{\sqrt{m}} + (l-1)|\phi(0)| \right)}{\min_{i\in[m]} \left\| W_i^{(l)} \right\|_2}
$$

where (a) follows from applying Cauchy-Schwarz first with respect to the dot product described by the index $j$ and then on the index $i$; (b) follows from $\|\mathbf{V}_2\|_2 \leq \|\mathbf{V}_2\|_F$; and (c) follows from Lemma A.1.

Now we analyze the second term,

$$
\sup_{\|\mathbf{v}_1\|_2 = \|\mathbf{V}_2\|_F = 1} \sum_{i=1}^{m} \left| \sum_{j=1}^{m} \sum_{k'=1}^{m} \frac{1}{m} \phi''(\tilde{\alpha}_i^{(l)}) \frac{W_{ij}^{(l)}}{\left\| W_i^{(l)} \right\|_2} \frac{((W_i^{(l)})^\top \alpha^{(l-1)})}{\left\| W_i^{(l)} \right\|_2^3} W_{ik'}^{(l)} \mathbf{v}_{1,j} \mathbf{V}_{2,ik'} \right|
$$

$$
\leq \frac{\beta_\phi}{m} \sup_{\|\mathbf{v}_1\|_2 = \|\mathbf{V}_2\|_F = 1} \sum_{i=1}^{m} \frac{|(W_i^{(l)})^\top \alpha^{(l-1)}|}{\left\| W_i^{(l)} \right\|_2} \left| \left( \sum_{j=1}^{m} \frac{W_{ij}^{(l)}}{\left\| W_i^{(l)} \right\|_2^2} \mathbf{v}_{1,j} \right) \left( \sum_{k'=1}^{m} \frac{W_{ik'}^{(l)}}{\left\| W_i^{(l)} \right\|_2} \mathbf{V}_{2,ik'} \right) \right|
$$

$$
\leq \frac{\beta_\phi}{m} \left\| \alpha^{(l-1)} \right\|_2 \sup_{\|\mathbf{v}_1\|_2 = \|\mathbf{V}_2\|_F = 1} \sum_{i=1}^{m} \left| \left( \sum_{j=1}^{m} \frac{W_{ij}^{(l)}}{\left\| W_i^{(l)} \right\|_2^2} \mathbf{v}_{1,j} \right) \left( \sum_{k'=1}^{m} \frac{W_{ik'}^{(l)}}{\left\| W_i^{(l)} \right\|_2} \mathbf{V}_{2,ik'} \right) \right|
$$

$$
\overset{(a)}{\leq} \frac{\beta_\phi}{m} \left\| \alpha^{(l-1)} \right\|_2 \sup_{\|\mathbf{v}_1\|_2 = \|\mathbf{V}_2\|_F = 1} \sqrt{\sum_{i=1}^{m} \frac{\|\mathbf{v}_1\|_2^2}{\left\| W_i^{(l)} \right\|_2^2}} \sqrt{\sum_{i=1}^{m} \|\mathbf{V}_{2,i*}\|_2^2}
$$

$$
= \frac{\beta_\phi}{m} \left\| \alpha^{(l-1)} \right\|_2 \sup_{\|\mathbf{v}_1\|_2 = \|\mathbf{V}_2\|_F = 1} \sqrt{\sum_{i=1}^{m} \frac{\|\mathbf{v}_1\|_2^2}{\left\| W_i^{(l)} \right\|_2^2}} \|\mathbf{V}_2\|_F
$$

$$
= \frac{\beta_\phi}{m} \left\| \alpha^{(l-1)} \right\|_2 \sqrt{\sum_{i=1}^{m} \frac{1}{\left\| W_i^{(l)} \right\|_2^2}}
$$

$$
\overset{(b)}{\leq} \frac{\beta_\phi}{\sqrt{m}} \left( \frac{1}{\sqrt{m}} + (l-1)|\phi(0)| \right) \cdot \frac{1}{\min_{i \in [m]} \left\| W_i^{(l)} \right\|_2} \sqrt{m}
$$

$$
= \frac{\beta_\phi \left( \frac{1}{\sqrt{m}} + (l-1)|\phi(0)| \right)}{\min_{i \in [m]} \left\| W_i^{(l)} \right\|_2}
$$

where (a) follows from successive applications of the Cauchy-Schwarz inequality; (b) follows from Lemma A.1.

Now we analyze the third term,

$$
\sup_{\|\mathbf{v}_1\|_2 = \|\mathbf{V}_2\|_F = 1} \sum_{i=1}^{m} \left| \sum_{j=1}^{m} \sum_{k'=1}^{m} \frac{1}{\sqrt{m}} \phi'(\tilde{\alpha}_i^{(l)}) \frac{W_{ik'}^{(l)}}{\left\| W_i^{(l)} \right\|_2^3} W_{ij}^{(l)} \mathbf{v}_{1,j} \mathbf{V}_{2,ik'} \right|
$$

$$
\leq \frac{1}{\sqrt{m}} \sup_{\|\mathbf{v}_1\|_2 = \|\mathbf{V}_2\|_F = 1} \sum_{i=1}^{m} \left| \left( \sum_{j=1}^{m} \frac{W_{ij}^{(l)}}{\left\| W_i^{(l)} \right\|_2^2} \mathbf{v}_{1,j} \right) \left( \sum_{k'=1}^{m} \frac{W_{ik'}^{(l)}}{\left\| W_i^{(l)} \right\|_2} \mathbf{V}_{2,ik'} \right) \right|
$$

$$
\overset{(a)}{\leq} \frac{1}{\sqrt{m}} \sup_{\|\mathbf{v}_1\|_2 = \|\mathbf{V}_2\|_F = 1} \sqrt{\sum_{i=1}^{m} \frac{\|\mathbf{v}_1\|_2^2}{\left\| W_i^{(l)} \right\|_2^2}} \sqrt{\sum_{i=1}^{m} \|\mathbf{V}_{2,i*}\|_2^2}
$$

$$
= \frac{1}{\sqrt{m}} \sup_{\|\mathbf{v}_1\|_2 = \|\mathbf{V}_2\|_F = 1} \sqrt{\sum_{i=1}^{m} \frac{\|\mathbf{v}_1\|_2^2}{\left\| W_i^{(l)} \right\|_2^2}} \|\mathbf{V}_2\|_F
$$

$$
= \frac{1}{\sqrt{m}} \sqrt{\sum_{i=1}^{m} \frac{1}{\left\| W_i^{(l)} \right\|_2^2}}
$$

$$
\leq \frac{1}{\min_{i \in [m]} \left\| W_i^{(l)} \right\|_2}
$$

where (a) follows from successive applications of the Cauchy-Schwarz inequality.

Finally, for the fourth term,

$$
\sup_{\|\mathbf{v}_1\|_2 = \|\mathbf{V}_2\|_F = 1} \sum_{i=1}^{m} \left| \sum_{j=1}^{m} \frac{1}{\sqrt{m}} \phi'(\tilde{\alpha}_i^{(l)}) \frac{1}{\left\| W_i^{(l)} \right\|_2} \mathbf{v}_{1,j} \mathbf{V}_{2,ij} \right|
$$

$$
\leq \frac{1}{\sqrt{m}} \frac{1}{\min_{i \in [m]} \left\| W_i^{(l)} \right\|_2} \sup_{\|\mathbf{v}_1\|_2 = \|\mathbf{V}_2\|_F = 1} \sum_{i=1}^{m} \left| \sum_{j=1}^{m} \mathbf{v}_{1,j} \mathbf{V}_{2,ij} \right|
$$

$$
\leq \frac{1}{\sqrt{m}} \frac{1}{\min_{i \in [m]} \left\| W_i^{(l)} \right\|_2} \sup_{\|\mathbf{v}_1\|_2 = \|\mathbf{V}_2\|_F = 1} \sum_{i=1}^{m} \|\mathbf{V}_{2,i*}\|_2 \|\mathbf{v}_1\|_2
$$

$$
= \frac{1}{\sqrt{m}} \frac{1}{\min_{i \in [m]} \left\| W_i^{(l)} \right\|_2} \sup_{\|\mathbf{V}_2\|_F = 1} \sum_{i=1}^{m} \|\mathbf{V}_{2,i*}\|_2
$$

$$
\leq \frac{1}{\sqrt{m}} \frac{1}{\min_{i \in [m]} \left\| W_i^{(l)} \right\|_2} \sup_{\|\mathbf{V}_2\|_F = 1} \sqrt{m} \sqrt{\sum_{i=1}^{m} \|\mathbf{V}_{2,i*}\|_2^2}
$$

$$
= \frac{1}{\min_{i \in [m]} \left\| W_i^{(l)} \right\|_2}.
$$

Then, taking all the analyzed terms back in (33), we obtain

$$
\left\| \frac{\partial^2 \alpha^{(l)}}{\partial \alpha^{(l-1)} \partial W^{(l)}} \right\|_{2,2,1} \leq \frac{2\beta_\phi (\frac{1}{\sqrt{m}} + (l-1)|\phi(0)|)}{\min_{i\in[m]} \left\| W_i^{(l)} \right\|_2} + \frac{2}{\min_{i\in[m]} \left\| W_i^{(l)} \right\|_2}
$$

$$
= \frac{2(\beta_\phi (\frac{1}{\sqrt{m}} + (l-1)|\phi(0)|) + 1)}{\min_{i\in[m]} \left\| W_i^{(l)} \right\|_2},
$$

which proves the inequality (30).

We now analyze the last inequality (31). Carefully following the chain rule in (28), we obtain

$$
\left( \frac{\partial^2 \alpha^{(l)}}{(\partial W^{(l)})^2} \right)_{i,jj',kk'}
$$

$$
= \phi''(\tilde{\alpha}_i^{(l)}) \frac{1}{\sqrt{m}} \left( \frac{\alpha_{j'}^{(l-1)}}{\left\| W_i^{(l)} \right\|_2} - \frac{((W_i^{(l)})^\top \alpha^{(l-1)}) W_{ij'}^{(l)}}{\left\| W_i^{(l)} \right\|_2^3} \right) \mathbb{1}_{[j=i]}
$$

$$
\times \frac{1}{\sqrt{m}} \left( \frac{\alpha_{k'}^{(l-1)}}{\left\| W_i^{(l)} \right\|_2} - \frac{((W_i^{(l)})^\top \alpha^{(l-1)}) W_{ik'}^{(l)}}{\left\| W_i^{(l)} \right\|_2^3} \right) \mathbb{1}_{[k=i]}
$$

$$
+ \phi'(\tilde{\alpha}_i^{(l)}) \frac{1}{\sqrt{m}} \mathbb{1}_{[j=i]} \left( - \frac{\alpha_{j'}^{(l-1)}}{\left\| W_i^{(l)} \right\|_2^3} W_{ik'}^{(l)} \mathbb{1}_{[k=i]} \right.
$$

$$
\left. - \left( \frac{\alpha_{k'}^{(l-1)} W_{ij'}}{\left\| W_i^{(l)} \right\|_2^3} + \frac{((W_i^{(l)})^\top \alpha^{(l-1)})}{\left\| W_i^{(l)} \right\|_2^3} \mathbb{1}_{[k'=j']} - 3 \frac{((W_i^{(l)})^\top \alpha^{(l-1)}) W_{ij'}^{(l)} W_{ik'}^{(l)}}{\left\| W_i^{(l)} \right\|_2^5} \right) \mathbb{1}_{[k=i]} \right).
$$

Then, we have, after some careful calculation,

$$
\left\|\frac{\partial^2 \alpha^{(l)}}{(\partial W^{(l)})^2}\right\|_{2,2,1}
$$

$$
= \sup_{\|\mathbf{V}_1\|_F=\|\mathbf{V}_2\|_F=1} \sum_{i=1}^{m} \left| \sum_{j'=1}^{m} \sum_{k'=1}^{m} \phi''(\tilde{\alpha}_i^{(l)}) \frac{1}{\sqrt{m}} \left( \frac{\alpha_j^{(l-1)}}{\left\|W_i^{(l)}\right\|_2} - \frac{((W_i^{(l)})^\top \alpha^{(l-1)})W_{ij'}^{(l)}}{\left\|W_i^{(l)}\right\|_2^3} \right) \right.
$$

$$
\times \frac{1}{\sqrt{m}} \left( \frac{\alpha_{k'}^{(l-1)}}{\left\|W_i^{(l)}\right\|_2} - \frac{((W_i^{(l)})^\top \alpha^{(l-1)})W_{ik'}^{(l)}}{\left\|W_i^{(l)}\right\|_2^3} \right) \mathbf{V}_{1,ij'}\mathbf{V}_{2,ik'}
$$

$$
+ \sum_{j'=1}^{m} \sum_{k'=1}^{m} \phi'(\tilde{\alpha}_i^{(l)}) \frac{1}{\sqrt{m}} \left( -\frac{\alpha_{j'}^{(l-1)}}{\left\|W_i^{(l)}\right\|_2^3} W_{ik'}^{(l)} \right.
$$

$$
\left. - \left( \frac{\alpha_{k'}^{(l-1)}W_{ij'}}{\left\|W_i^{(l)}\right\|_2^3} + \frac{((W_i^{(l)})^\top \alpha^{(l-1)})}{\left\|W_i^{(l)}\right\|_2^3} \mathbb{1}_{[k'=j']} - 3\frac{((W_i^{(l)})^\top \alpha^{(l-1)})W_{ij'}^{(l)}W_{ik'}^{(l)}}{\left\|W_i^{(l)}\right\|_2^5} \right) \right)
$$

$$
\left. \times \mathbf{V}_{1,ij'}\mathbf{V}_{2,ik'} \right|
$$

$$
\leq \sup_{\|\mathbf{V}_1\|_2=\|\mathbf{V}_2\|_F=1} \sum_{i=1}^{m} \left| \sum_{j'=1}^{m} \sum_{k'=1}^{m} \frac{1}{m} \phi''(\tilde{\alpha}_i^{(l)}) \frac{\alpha_{j'}^{(l-1)}\alpha_{k'}^{(l-1)}}{\left\|W_i^{(l)}\right\|_2^2} \mathbf{V}_{1,ij'}\mathbf{V}_{2,ik'} \right|
$$

$$
+ \sup_{\|\mathbf{V}_1\|_2=\|\mathbf{V}_2\|_F=1} \sum_{i=1}^{m} \left| \sum_{j'=1}^{m} \sum_{k'=1}^{m} \frac{1}{m} \phi''(\tilde{\alpha}_i^{(l)}) \frac{((W_i^{(l)})^\top \alpha^{(l-1)})}{\left\|W_i^{(l)}\right\|_2^4} W_{ik'}^{(l)}\alpha_{j'}^{(l-1)}\mathbf{V}_{1,ij'}\mathbf{V}_{2,ik'} \right| \tag{34}
$$

$$
+ \sup_{\|\mathbf{V}_1\|_2=\|\mathbf{V}_2\|_F=1} \sum_{i=1}^{m} \left| \sum_{j'=1}^{m} \sum_{k'=1}^{m} \frac{1}{m} \phi''(\tilde{\alpha}_i^{(l)}) \frac{((W_i^{(l)})^\top \alpha^{(l-1)})}{\left\|W_i^{(l)}\right\|_2^4} W_{ij'}^{(l)}\alpha_{k'}^{(l-1)}\mathbf{V}_{1,ij'}\mathbf{V}_{2,ik'} \right|
$$

$$
+ \sup_{\|\mathbf{V}_1\|_2=\|\mathbf{V}_2\|_F=1} \sum_{i=1}^{m} \left| \sum_{j'=1}^{m} \sum_{k'=1}^{m} \frac{1}{m} \phi''(\tilde{\alpha}_i^{(l)}) \frac{((W_i^{(l)})^\top \alpha^{(l-1)})^2}{\left\|W_i^{(l)}\right\|_2^6} W_{ij'}^{(l)}W_{ik'}^{(l)}\mathbf{V}_{1,ij'}\mathbf{V}_{2,ik'} \right|
$$

$$
+ \sup_{\|\mathbf{V}_1\|_2=\|\mathbf{V}_2\|_F=1} \sum_{i=1}^{m} \left| \sum_{j'=1}^{m} \sum_{k'=1}^{m} \frac{1}{\sqrt{m}} \phi'(\tilde{\alpha}_i^{(l)}) \frac{1}{\left\|W_i^{(l)}\right\|_2^3} W_{ik'}^{(l)}\alpha_{j'}^{(l-1)}\mathbf{V}_{1,ij'}\mathbf{V}_{2,ik'} \right|
$$

$$
+ \sup_{\|\mathbf{V}_1\|_2=\|\mathbf{V}_2\|_F=1} \sum_{i=1}^{m} \left| \sum_{j'=1}^{m} \sum_{k'=1}^{m} \frac{1}{\sqrt{m}} \phi'(\tilde{\alpha}_i^{(l)}) \frac{1}{\left\|W_i^{(l)}\right\|_2^3} W_{ij'}^{(l)}\alpha_{k'}^{(l-1)}\mathbf{V}_{1,ij'}\mathbf{V}_{2,ik'} \right|
$$

$$
+ \sup_{\|\mathbf{V}_1\|_2=\|\mathbf{V}_2\|_F=1} \sum_{i=1}^{m} \left| \sum_{j'=1}^{m} \frac{1}{\sqrt{m}} \phi'(\tilde{\alpha}_i^{(l)}) \frac{((W_i^{(l)})^\top \alpha^{(l-1)})}{\left\|W_i^{(l)}\right\|_2^3} \mathbf{V}_{1,ij'}\mathbf{V}_{2,ij'} \right|
$$

$$
+ \sup_{\|\mathbf{V}_1\|_2=\|\mathbf{V}_2\|_F=1} \sum_{i=1}^{m} \left| \sum_{j'=1}^{m} \sum_{k'=1}^{m} \frac{3}{\sqrt{m}} \phi'(\tilde{\alpha}_i^{(l)}) \frac{((W_i^{(l)})^\top \alpha^{(l-1)})}{\left\|W_i^{(l)}\right\|_2^5} W_{ij'}^{(l)}W_{ik'}^{(l)}\mathbf{V}_{1,ij'}\mathbf{V}_{2,ik'} \right|
$$

Now we upper bound each of the terms in the last inequality. We analyze the first term,

$$\sup_{\|\mathbf{V}_1\|_2=\|\mathbf{V}_2\|_F=1} \sum_{i=1}^{m} \left| \sum_{j'=1}^{m} \sum_{k'=1}^{m} \frac{1}{m} \phi''(\tilde{\alpha}_i^{(l)}) \frac{\alpha_{j'}^{(l-1)} \alpha_{k'}^{(l-1)}}{\left\| W_i^{(l)} \right\|_2^2} \mathbf{V}_{1,j'} \mathbf{V}_{2,ik'} \right|$$

$$\leq \frac{\beta_\phi}{m} \sup_{\|\mathbf{V}_1\|_2=\|\mathbf{V}_2\|_F=1} \sum_{i=1}^{m} \left| \frac{(\mathbf{V}_1 \alpha^{(l-1)})_i (\mathbf{V}_2 \alpha^{(l-1)})_i}{\left\| W_i^{(l)} \right\|_2^2} \right|$$

$$\leq \frac{\beta_\phi}{m} \frac{1}{\min_{i\in[m]} \left\| W_i^{(l)} \right\|_2^2} \sup_{\|\mathbf{V}_1\|_2=\|\mathbf{V}_2\|_F=1} \left\| \mathbf{V}_1 \alpha^{(l-1)} \right\|_2 \left\| \mathbf{V}_2 \alpha^{(l-1)} \right\|_2$$

$$\leq \frac{\beta_\phi}{m} \frac{1}{\min_{i\in[m]} \left\| W_i^{(l)} \right\|_2^2} \left\| \alpha^{(l-1)} \right\|_2^2 \sup_{\|\mathbf{V}_1\|_2=\|\mathbf{V}_2\|_F=1} \left\| \mathbf{V}_1 \right\|_2 \left\| \mathbf{V}_2 \right\|_2$$

$$= \frac{\beta_\phi}{m} \frac{1}{\min_{i\in[m]} \left\| W_i^{(l)} \right\|_2^2} \left\| \alpha^{(l-1)} \right\|_2^2$$

$$\leq \frac{\beta_\phi (\frac{1}{\sqrt{m}} + (l-1)|\phi(0)|)^2}{\min_{i\in[m]} \left\| W_i^{(l)} \right\|_2^2}.$$

Now we analyze the second term,

$$\sup_{\|\mathbf{V}_1\|_F=\|\mathbf{V}_2\|_F=1} \sum_{i=1}^{m} \left| \sum_{j'=1}^{m} \sum_{k'=1}^{m} \frac{1}{m} \phi''(\tilde{\alpha}_i^{(l)}) \frac{((W_i^{(l)})^\top \alpha^{(l-1)})}{\left\| W_i^{(l)} \right\|_2^4} W_{ik'}^{(l)} \alpha_{j'}^{(l-1)} \mathbf{V}_{1,ij'} \mathbf{V}_{2,ik'} \right|$$

$$\leq \frac{\beta_\phi}{m} \sup_{\|\mathbf{V}_1\|_F=\|\mathbf{V}_2\|_F=1} \sum_{i=1}^{m} \frac{|(W_i^{(l)})^\top \alpha^{(l-1)}|}{\left\| W_i^{(l)} \right\|_2^3} \left| \left( \sum_{j'=1}^{m} \alpha_{j'}^{(l-1)} \mathbf{V}_{1,ij'} \right) \left( \sum_{k'=1}^{m} \frac{W_{ik'}^{(l)}}{\left\| W_i^{(l)} \right\|_2} \mathbf{V}_{2,ik'} \right) \right|$$

$$\leq \frac{\beta_\phi}{m} \left\| \alpha^{(l-1)} \right\|_2 \frac{1}{\min_{i\in[m]} \left\| W_i^{(l)} \right\|_2^2} \sup_{\|\mathbf{V}_1\|_F=\|\mathbf{V}_2\|_F=1} \left\| \mathbf{V}_1 \alpha^{(l-1)} \right\|_2$$

$$\times \sqrt{\sum_{i=1}^{m} \left( \sum_{k'=1}^{m} \frac{W_{ik'}^{(l)}}{\left\| W_i^{(l)} \right\|_2} \mathbf{V}_{2,ik'} \right)^2}$$

$$\leq \frac{\beta_\phi}{m} \left\| \alpha^{(l-1)} \right\|_2^2 \frac{1}{\min_{i\in[m]} \left\| W_i^{(l)} \right\|_2^2} \sup_{\|\mathbf{V}_2\|_F=1} \sqrt{\sum_{i=1}^{m} \|\mathbf{V}_{2,i*}\|_2^2}$$

$$\leq \frac{\beta_\phi (\frac{1}{\sqrt{m}} + (l-1)|\phi(0)|)^2}{\min_{i\in[m]} \left\| W_i^{(l)} \right\|_2^2}.$$

The third term is analyzed very similarly as the second term and thus have the same upper bound.

Now, we analyze the fourth term,

$$
\sup_{\|\mathbf{V}_1\|_F=\|\mathbf{V}_2\|_F=1} \sum_{i=1}^m \left| \sum_{j'=1}^m \sum_{k'=1}^m \frac{1}{m} \phi''(\tilde{\alpha}_i^{(l)}) \frac{((W_i^{(l)})^\top \alpha^{(l-1)})^2}{\left\| W_i^{(l)} \right\|_2^6} W_{ij'}^{(l)} W_{ik'}^{(l)} \mathbf{V}_{1,ij'} \mathbf{V}_{2,ik'} \right|
$$

$$
\leq \frac{\beta_\phi}{m} \sup_{\|\mathbf{V}_1\|_F=\|\mathbf{V}_2\|_F=1} \sum_{i=1}^m \frac{|(W_i^{(l)})^\top \alpha^{(l-1)}|^2}{\left\| W_i^{(l)} \right\|_2^4} \left| \left( \sum_{j'=1}^m \frac{W_{ij'}^{(l)}}{\left\| W_i^{(l)} \right\|_2} \mathbf{V}_{1,ij'} \right) \left( \sum_{k'=1}^m \frac{W_{ik'}^{(l)}}{\left\| W_i^{(l)} \right\|_2} \mathbf{V}_{2,ik'} \right) \right|
$$

$$
\leq \frac{\beta_\phi}{m} \left\| \alpha^{(l-1)} \right\|_2^2 \frac{1}{\min_{i\in[m]} \left\| W_i^{(l)} \right\|_2^2} \sup_{\|\mathbf{V}_1\|_F=\|\mathbf{V}_2\|_F=1} \sum_{i=1}^m \left| \left( \sum_{j'=1}^m \frac{W_{ij'}^{(l)}}{\left\| W_i^{(l)} \right\|_2} \mathbf{V}_{1,ij'} \right) \right.
$$

$$
\left. \times \left( \sum_{k'=1}^m \frac{W_{ik'}^{(l)}}{\left\| W_i^{(l)} \right\|_2} \mathbf{V}_{2,ik'} \right) \right|
$$

$$
\leq \frac{\beta_\phi}{m} \left\| \alpha^{(l-1)} \right\|_2^2 \frac{1}{\min_{i\in[m]} \left\| W_i^{(l)} \right\|_2^2} \sup_{\|\mathbf{V}_1\|_F=\|\mathbf{V}_2\|_F=1} \sqrt{\sum_{i=1}^m \|\mathbf{V}_{1,i*}\|_2^2} \sqrt{\sum_{i=1}^m \|\mathbf{V}_{2,i*}\|_2^2}
$$

$$
= \frac{\beta_\phi}{m} \left\| \alpha^{(l-1)} \right\|_2^2 \frac{1}{\min_{i\in[m]} \left\| W_i^{(l)} \right\|_2^2}
$$

$$
\leq \frac{\beta_\phi}{m} (1 + (l-1)|\phi(0)|\sqrt{m})^2 \cdot \frac{1}{\min_{i\in[m]} \left\| W_i^{(l)} \right\|_2^2}
$$

$$
= \frac{\beta_\phi(\frac{1}{\sqrt{m}} + (l-1)|\phi(0)|)^2}{\min_{i\in[m]} \left\| W_i^{(l)} \right\|_2^2}.
$$

Now, for the fifth term,

$$
\sup_{\|\mathbf{V}_1\|_F=\|\mathbf{V}_2\|_F=1} \sum_{i=1}^m \left| \sum_{j'=1}^m \sum_{k'=1}^m \frac{1}{\sqrt{m}} \phi'(\tilde{\alpha}_i^{(l)}) \frac{1}{\left\| W_i^{(l)} \right\|_2^3} W_{ik'}^{(l)} \alpha_{j'}^{(l-1)} \mathbf{V}_{1,ij'} \mathbf{V}_{2,ik'} \right|
$$

$$
\leq \frac{1}{\sqrt{m}} \frac{1}{\min_{i\in[m]} \left\| W_i^{(l)} \right\|_2^2} \sup_{\|\mathbf{V}_1\|_F=\|\mathbf{V}_2\|_F=1} \left\| \mathbf{V}_1 \alpha^{(l-1)} \right\|_2 \sqrt{\sum_{i=1}^m \left( \sum_{k'=1}^m \frac{W_{ik'}^{(l)}}{\left\| W_i^{(l)} \right\|_2} \mathbf{V}_{2,ik'} \right)^2}
$$

$$
\leq \frac{1}{\sqrt{m}} \left\| \alpha^{(l-1)} \right\|_2 \frac{1}{\min_{i\in[m]} \left\| W_i^{(l)} \right\|_2^2} \sup_{\|\mathbf{V}_2\|_F=1} \sqrt{\sum_{i=1}^m \|\mathbf{V}_{2,i*}\|_2^2}
$$

$$
\leq \frac{(\frac{1}{\sqrt{m}} + (l-1)|\phi(0)|)}{\min_{i\in[m]} \left\| W_i^{(l)} \right\|_2^2}.
$$

The sixth term is analyzed very similarly as the fifth term and thus have the same upper bound.

Now, for the seventh term,

$$\sup_{\|\mathbf{V}_1\|_F=\|\mathbf{V}_2\|_F=1} \sum_{i=1}^m \left| \sum_{j'=1}^m \frac{1}{\sqrt{m}} \phi'(\tilde{\alpha}_i^{(l)}) \frac{((W_i^{(l)})^\top \alpha^{(l-1)})}{\left\|W_i^{(l)}\right\|_2^3} \mathbf{V}_{1,ij'} \mathbf{V}_{2,ij'} \right|$$

$$\leq \frac{1}{\sqrt{m}} \frac{1}{\min_{i\in[m]} \left\|W_i^{(l)}\right\|_2^2} \sup_{\|\mathbf{V}_1\|_F=\|\mathbf{V}_2\|_F=1} \sum_{i=1}^m \left| \sum_{j'=1}^m \frac{((W_i^{(l)})^\top \alpha^{(l-1)})}{\left\|W_i^{(l)}\right\|_2} \mathbf{V}_{1,ij'} \mathbf{V}_{2,ij'} \right|$$

$$\leq \frac{1}{\sqrt{m}} \frac{1}{\min_{i\in[m]} \left\|W_i^{(l)}\right\|_2^2} \left\|\alpha^{(l-1)}\right\|_2 \sup_{\|\mathbf{V}_1\|_F=\|\mathbf{V}_2\|_F=1} \sum_{i=1}^m \left| \sum_{j'=1}^m \mathbf{V}_{1,ij'} \mathbf{V}_{2,ij'} \right|$$

$$\leq \frac{1}{\sqrt{m}} \frac{1}{\min_{i\in[m]} \left\|W_i^{(l)}\right\|_2^2} \left\|\alpha^{(l-1)}\right\|_2 \sup_{\|\mathbf{V}_1\|_F=\|\mathbf{V}_2\|_F=1} \sum_{i=1}^m \|\mathbf{V}_{2,i*}\|_2 \|\mathbf{V}_{1,i*}\|_2$$

$$\leq \frac{1}{\sqrt{m}} \frac{1}{\min_{i\in[m]} \left\|W_i^{(l)}\right\|_2^2} \left\|\alpha^{(l-1)}\right\|_2 \sup_{\|\mathbf{V}_1\|_F=\|\mathbf{V}_2\|_F=1} \|\mathbf{V}_2\|_F \|\mathbf{V}_1\|_F$$

$$= \frac{1}{\sqrt{m}} \frac{1}{\min_{i\in[m]} \left\|W_i^{(l)}\right\|_2^2} \left\|\alpha^{(l-1)}\right\|_2$$

$$\leq \frac{(\frac{1}{\sqrt{m}} + (l-1)|\phi(0)|)}{\min_{i\in[m]} \left\|W_i^{(l)}\right\|_2^2}.$$

Finally, for the eight term, following an analysis similar to the fourth term,

$$\sup_{\|\mathbf{V}_1\|_F=\|\mathbf{V}_2\|_F=1} \sum_{i=1}^m \left| \sum_{j'=1}^m \sum_{k'=1}^m \frac{3}{\sqrt{m}} \phi'(\tilde{\alpha}_i^{(l)}) \frac{((W_i^{(l)})^\top \alpha^{(l-1)})}{\left\|W_i^{(l)}\right\|_2^5} W_{ij'}^{(l)} W_{ik'}^{(l)} \mathbf{V}_{1,ij'} \mathbf{V}_{2,ik'} \right|$$

$$\leq \frac{3}{\sqrt{m}} \left\|\alpha^{(l-1)}\right\|_2 \sup_{\|\mathbf{V}_1\|_2=\|\mathbf{V}_2\|_F=1} \sum_{i=1}^m \frac{1}{\left\|W_i^{(l)}\right\|_2^2} \left| \left( \sum_{j'=1}^m \frac{W_{ij'}^{(l)}}{\left\|W_i^{(l)}\right\|_2} \mathbf{V}_{1,ij'} \right) \right.$$

$$\left. \times \left( \sum_{k'=1}^m \frac{W_{ik'}^{(l)}}{\left\|W_i^{(l)}\right\|_2} \mathbf{V}_{2,ik'} \right) \right|$$

$$\leq \frac{3}{\sqrt{m}} \left\|\alpha^{(l-1)}\right\|_2 \frac{1}{\min_{i\in[m]} \left\|W_i^{(l)}\right\|_2^2} \sup_{\|\mathbf{V}_1\|_2=\|\mathbf{V}_2\|_F=1} \sqrt{\sum_{i=1}^m \|\mathbf{V}_{1,i*}\|_2^2} \sqrt{\sum_{i=1}^m \|\mathbf{V}_{2,i*}\|_2^2}$$

$$\leq \frac{3(\frac{1}{\sqrt{m}} + (l-1)|\phi(0)|)}{\min_{i\in[m]} \left\|W_i^{(l)}\right\|_2^2}.$$

Then, taking all the analyzed terms back in (34), we obtain

$$\left\| \frac{\partial^2 \alpha^{(l)}}{(\partial W^{(l)})^2} \right\|_{2,2,1} \leq \frac{4\beta_\phi (\frac{1}{\sqrt{m}} + (l-1)|\phi(0)|)^2}{\min_{i\in[m]} \left\|W_i^{(l)}\right\|_2^2} + \frac{6(\frac{1}{\sqrt{m}} + (l-1)|\phi(0)|)}{\min_{i\in[m]} \left\|W_i^{(l)}\right\|_2^2}.$$

This finishes the proof.

## A.5 Upper bound of the $L_\infty$-norm of $\frac{\partial f}{\partial \alpha^{(l)}}$

As seen before, to upper bound the Hessian, we need to upper bound the quantity $\left\|\frac{\partial f}{\partial \alpha^{(l)}}\right\|_\infty$. For our purposes, we will actually upper bound the $L_2$-norm $\left\|\frac{\partial f}{\partial \alpha^{(l)}}\right\|_2$ and use this to upper bound the $L_\infty$-norm.

**Lemma A.5.** *Consider Assumptions 1 and 2. For any $l \in [L]$, any $\mathbf{x} \in \mathbb{R}^d$ such that $\|\mathbf{x}\|_2 = 1$, and any $\theta \in \mathbb{R}^p$ with $\mathbf{v} \in B_{\rho_1}^{\mathrm{Euc}}(\mathbf{v}_0)$, we have*

$$\left\|\frac{\partial f(\theta; \mathbf{x})}{\partial \alpha^{(l)}(\mathbf{x})}\right\|_2 \le (1 + \rho_1). \tag{35}$$

*Proof.* From Lemma A.2, we know that $\left\|\frac{\partial f}{\partial \alpha^{(L)}}\right\|_2 \le 1 + \rho_1$. Now, for the inductive step, assume $\left\|\frac{\partial f}{\partial \alpha^{(l)}}\right\|_2 \le (1 + \rho_1)$ for any $l < L$. Then,

$$\left\|\frac{\partial f}{\partial \alpha^{(l-1)}}\right\|_2 = \left\|\frac{\partial \alpha^{(l)}}{\partial \alpha^{(l-1)}}\frac{\partial f}{\partial \alpha^{(l)}}\right\|_2 \le \left\|\frac{\partial \alpha^{(l)}}{\partial \alpha^{(l-1)}}\right\|_2 \left\|\frac{\partial f}{\partial \alpha^{(l)}}\right\|_2 \le 1 \cdot (1 + \rho_1)$$

where the last inequality follows again from Lemma A.2. We have finished the proof by induction. $\square$

## A.6 Finishing the proof of Theorem 4.1

Replacing all the previously derived results back in (22), we obtain

$$\left\|\frac{\partial^2 f(\theta)}{\partial W^{(l_1)}\partial W^{(l_2)}}\right\|_2 \leq \frac{2(2\beta_\phi(\frac{1}{\sqrt{m}}+(l_1-1)|\phi(0)|)+3)(\frac{1}{\sqrt{m}}+(l_1-1)|\phi(0)|)}{\min_{i\in[m]}\left\|W_i^{(l_1)}\right\|_2^2}(1+\rho_1)$$

$$+\frac{2(\frac{1}{\sqrt{m}}+(l_1-1)|\phi(0)|)}{\min_{i\in[m]}\left\|W_i^{(l_1)}\right\|_2}\left(\prod_{l=l_1+1}^{l_2-1}1\right)\frac{2(\beta_\phi(\frac{1}{\sqrt{m}}+(l_2-1)|\phi(0)|)+1)}{\min_{i\in[m]}\left\|W_i^{(l_2)}\right\|_2}$$

$$\times(1+\rho_1)$$

$$+\frac{2(\frac{1}{\sqrt{m}}+(l_1-1)|\phi(0)|)}{\min_{i\in[m]}\left\|W_i^{(l_1)}\right\|_2}\frac{2(\frac{1}{\sqrt{m}}+(l_2-1)|\phi(0)|)}{\min_{i\in[m]}\left\|W_i^{(l_2)}\right\|_2}\sum_{l=l_2+1}^{L}\left(\prod_{l'=l_1+1}^{l}1\right)$$

$$\times\beta_\phi\left(\prod_{\hat{l}=l_2+1}^{l}1\right)(1+\rho_1)$$

$$=\frac{2(2\beta_\phi(\frac{1}{\sqrt{m}}+(l_1-1)|\phi(0)|)+3)(\frac{1}{\sqrt{m}}+(l_1-1)|\phi(0)|)}{\min_{i\in[m]}\left\|W_i^{(l_1)}\right\|_2^2}(1+\rho_1)$$

$$+\frac{2(\frac{1}{\sqrt{m}}+(l_1-1)|\phi(0)|)}{\min_{i\in[m]}\left\|W_i^{(l_1)}\right\|_2}\frac{2(\beta_\phi(\frac{1}{\sqrt{m}}+(l_2-1)|\phi(0)|)+1)}{\min_{i\in[m]}\left\|W_i^{(l_2)}\right\|_2}(1+\rho_1) \qquad (36)$$

$$+\frac{2(\frac{1}{\sqrt{m}}+(l_1-1)|\phi(0)|)}{\min_{i\in[m]}\left\|W_i^{(l_1)}\right\|_2}\frac{2(\frac{1}{\sqrt{m}}+(l_2-1)|\phi(0)|)}{\min_{i\in[m]}\left\|W_i^{(l_2)}\right\|_2}\cdot(L-l_2)\cdot\beta_\phi(1+\rho_1)$$

$$\leq\frac{2(2\beta_\phi(\frac{1}{\sqrt{m}}+L|\phi(0)|)+3)(\frac{1}{\sqrt{m}}+L|\phi(0)|)}{\min_{\substack{i\in[m]\\l\in[L]}}\left\|W_i^{(l)}\right\|_2^2}(1+\rho_1)$$

$$+\frac{4(\frac{1}{\sqrt{m}}+L|\phi(0)|)(\beta_\phi(\frac{1}{\sqrt{m}}+L|\phi(0)|)+1)}{\min_{\substack{i\in[m]\\l\in[L]}}\left\|W_i^{(l)}\right\|_2^2}(1+\rho_1)$$

$$+\frac{4L\beta_\phi(\frac{1}{\sqrt{m}}+L|\phi(0)|)^2}{\min_{\substack{i\in[m]\\l\in[L]}}\left\|W_i^{(l)}\right\|_2^2}(1+\rho_1)$$

$$\leq\mathcal{O}\left(\frac{L(1+\rho_1)(\frac{1}{\sqrt{m}}+L^2\max\{|\phi(0)|,|\phi(0)|^2\})}{\min_{\substack{i\in[m]\\l\in[L]}}\left\|W_i^{(l)}\right\|_2^2}\right)$$

Now, notice that $\frac{\partial^2 f(\theta)}{\partial W^{(l_1)}}=\frac{\partial\alpha^{(l_1)}}{\partial W^{(l_1)}}\prod_{l'=l_1+1}^{L}\frac{\partial\alpha^{(l')}}{\partial\alpha^{(l'_1-1)}}\cdot\mathbf{v}$, then

$$\left\|\frac{\partial^2 f(\theta)}{\partial W^{(l_1)}\partial W^{(L+1)}}\right\|_2 \leq \left\|\frac{\partial\alpha^{(l_1)}}{\partial W^{(l_1)}}\right\|_2\prod_{l'=l_1+1}^{L}\left\|\frac{\partial\alpha^{(l')}}{\partial\alpha^{(l'_1-1)}}\right\|_2$$

$$\leq\frac{2(\frac{1}{\sqrt{m}}+(l_1-1)|\phi(0)|)}{\min_{i\in[m]}\left\|W_i^{(l_1)}\right\|_2}\leq\frac{2(\frac{1}{\sqrt{m}}+L|\phi(0)|)}{\min_{i\in[m]}\left\|W_i^{(l_1)}\right\|_2}\leq\mathcal{O}\left(\frac{(\frac{1}{\sqrt{m}}+L|\phi(0)|)}{\min_{\substack{i\in[m]\\l\in[L]}}\left\|W_i^{(l)}\right\|_2}\right) \qquad (37)$$

Replacing (36) along with (37) back in (21), we obtain

$$
\left\|\frac{\partial^2 f(\theta)}{(\partial\theta)^2}\right\|_2 \le L^2 \mathcal{O}\left(\frac{L(1+\rho_1)(\frac{1}{\sqrt{m}} + L^2 \max\{|\phi(0)|, |\phi(0)|^2\})}{\min_{\substack{i\in[m]\\l\in[L]}}\left\|W_i^{(l)}\right\|_2^2}\right) + 2L\,\mathcal{O}\left(\frac{\frac{1}{\sqrt{m}} + L|\phi(0)|}{\min_{\substack{i\in[m]\\l\in[L]}}\left\|W_i^{(l)}\right\|_2}\right)
$$

$$
\le \mathcal{O}\left(\frac{(1+\rho_1)L^3(\frac{1}{\sqrt{m}} + L^2 \max\{|\phi(0)|, |\phi(0)|^2\})}{\min\left\{\min_{\substack{i\in[m]\\l\in[L]}}\left\|W_i^{(l)}\right\|_2, \min_{\substack{i\in[m]\\l\in[L]}}\left\|W_i^{(l)}\right\|_2^2\right\}}\right). \tag{38}
$$

This finishes the proof.

## B  Upper bounds on gradients of the predictor and empirical loss for WeightNorm

**Lemma 4.1** (**Predictor gradient bounds**). *Under Assumptions 1 and 2, for any $\theta \in \mathbb{R}^p$ with $\mathbf{v} \in B_{\rho_1}^{\mathrm{Euc}}(\mathbf{v}_0)$, and any $\mathbf{x}_i, i \in [n]$, we have*

$$
\|\nabla_\theta f(\theta; \mathbf{x}_i)\|_2 \le \varrho_\theta \le \mathcal{O}\left((1 + L|\phi(0)|\sqrt{m})\left(1 + \frac{\sqrt{L}(1+\rho_1)}{\|\bar{W}\|_2}\right)\right) \tag{6}
$$

*where $\varrho_\theta^2 := (1 + L|\phi(0)|\sqrt{m})^2 + 4(1+\rho_1)^2 \sum_{l=1}^{L} \frac{1}{\|\bar{W}\|_2^2}\left(\frac{1}{\sqrt{m}} + (l-1)|\phi(0)|\right)^2$, and*

$$
\|\nabla_\mathbf{x} f(\theta; \mathbf{x}_i)\|_2 \le 1 + \rho_1. \tag{7}
$$

*Proof.* We first prove the bound on the gradient with respect to the weights. Using the chain rule,

$$
\frac{\partial f}{\partial W^{(l)}} = \frac{\partial \alpha^{(l)}}{\partial W^{(l)}} \prod_{l'=l+1}^{L} \frac{\partial \alpha^{(l')}}{\partial \alpha^{(l'-1)}} \frac{\partial f}{\partial \alpha^{(L)}}
$$

and so

$$
\left\|\frac{\partial f}{\partial W^{(l)}}\right\|_2 \le \left\|\frac{\partial \alpha^{(l)}}{\partial W^{(l)}}\right\|_2 \left\|\frac{\partial f}{\partial \alpha^{(l)}}\right\|_2 \overset{(a)}{\le} \left\|\frac{\partial \alpha^{(l)}}{\partial W^{(l)}}\right\|_2 \cdot (1 + \rho_1)
$$

$$
\overset{(b)}{\le} \left(\frac{2}{\min_{i\in[m]}\left\|W_i^{(l)}\right\|_2}\left(\frac{1}{\sqrt{m}} + (l-1)|\phi(0)|\right)\right) \cdot (1 + \rho_1)
$$

for $l \in [L]$, where (a) follows from Lemma A.5, (b) follows from Lemma A.3. Similarly,

$$
\left\|\frac{\partial f}{\partial W^{(L+1)}}\right\|_2 = \left\|\alpha^{(L)}\right\|_2 \le (1 + L|\phi(0)|\sqrt{m}) \;,
$$

where we used Lemma A.1 for the inequality. Now,

$$
\|\nabla_\theta f\|_2^2 = \sum_{l=1}^{L+1} \left\|\frac{\partial f}{\partial W^{(l)}}\right\|_2^2
$$

$$
\le (1 + L|\phi(0)|\sqrt{m})^2 + 4(1+\rho_1)^2 \sum_{l=1}^{L} \frac{1}{\min_{i\in[m]}\left\|W_i^{(l)}\right\|_2^2}\left(\frac{1}{\sqrt{m}} + (l-1)|\phi(0)|\right)^2
$$

$$
= \varrho_\theta^2.
$$

We further note that,

$$\varrho_\theta^2 \leq \mathcal{O}\left( (1 + L^2|\phi(0)|^2 m) + L(1+\rho_1)^2 \frac{1}{\min_{\substack{i\in[m]\\l\in[L]}} \left\|W_i^{(l)}\right\|_2^2} \left(\frac{1}{m} + L^2|\phi(0)|^2\right) \right)$$

$$\leq \mathcal{O}\left( (1 + L^2|\phi(0)|^2 m)\left( 1 + \frac{L(1+\rho_1)^2}{\min_{\substack{i\in[m]\\l\in[L]}} \left\|W_i^{(l)}\right\|_2^2} \right) \right)$$

$$\Rightarrow \varrho_\theta \leq \mathcal{O}\left( (1 + L|\phi(0)|\sqrt{m})\left( 1 + \frac{\sqrt{L}(1+\rho_1)}{\min_{\substack{i\in[m]\\l\in[L]}} \left\|W_i^{(l)}\right\|_2} \right) \right),$$

where the implication follows from the property $\sqrt{a+b} \leq \sqrt{a} + \sqrt{b}$ for $a, b \geq 0$.

We now prove the bound on the gradient with respect to the input data. Again, using the chain rule,

$$\frac{\partial f}{\partial \mathbf{x}} = \frac{\partial f}{\partial \alpha^{(0)}} = \frac{\partial \alpha^{(1)}}{\partial \alpha^{(0)}}\left(\prod_{l'=2}^{L} \frac{\partial \alpha^{(l')}}{\partial \alpha^{(l'-1)}}\right)\frac{\partial f}{\partial \alpha^{(L)}}$$

and so

$$\left\|\frac{\partial f}{\partial \mathbf{x}}\right\|_2 \leq \left\|\frac{\partial \alpha^{(1)}}{\partial \alpha^{(0)}}\right\|_2 \left\|\left(\prod_{l'=2}^{L} \frac{\partial \alpha^{(l')}}{\partial \alpha^{(l'-1)}}\right)\frac{\partial f}{\partial \alpha^{(L)}}\right\|_2$$

$$\leq \left\|\frac{\partial \alpha^{(1)}}{\partial \alpha^{(0)}}\right\|_2 \left(\prod_{l'=2}^{L} \left\|\frac{\partial \alpha^{(l')}}{\partial \alpha^{(l'-1)}}\right\|_2\right)\left\|\frac{\partial f}{\partial \alpha^{(L)}}\right\|_2$$

$$\overset{(a)}{\leq} 1 + \rho_1$$

where (a) follows from Lemma A.2 and Lemma A.5. This completes the proof. $\qquad\square$

**Proposition 4.1** (**Empirical loss and empirical loss gradient bounds**). *Consider the square loss. Under Assumptions 1 and 2, the following inequality holds for any $\theta \in \mathbb{R}^p$ with $\mathbf{v} \in B_{\rho_1}^{\text{Euc}}(\mathbf{v}_0)$,*

$$\mathcal{L}(\theta) \leq \varphi \leq \mathcal{O}((1+\rho_1)^2(1 + L^2|\phi(0)|^2 m)), \tag{9}$$

*where $\varphi := \frac{2}{n}\sum_{i=1}^{n} y_i^2 + 2(1+\rho_1)^2(1 + L|\phi(0)|\sqrt{m})^2$. Moreover,*

$$\|\nabla_\theta \mathcal{L}(\theta)\|_2 \leq 2\sqrt{\mathcal{L}(\theta)}\varrho_\theta \leq 2\varrho_\theta\sqrt{\varphi}, \tag{10}$$

*with $\varrho_\theta$ as in Lemma 4.1.*

*Proof.* We start by noticing that for $\theta \in \mathbb{R}^p$,

$$\mathcal{L}(\theta) = \frac{1}{n}\sum_{i=1}^{n}(y_i - f(\theta; \mathbf{x}_i))^2 \leq \frac{1}{n}\sum_{i=1}^{n}(2y_i^2 + 2|f(\theta; \mathbf{x}_i)|^2). \tag{39}$$

Now, using the assumption on the weight of the output layer,

$$|f(\theta; \mathbf{x})| = \mathbf{v}^\top \alpha^{(L)}(\mathbf{x})$$

$$\leq \|\mathbf{v}\|_2 \left\|\alpha^{(L)}(\mathbf{x})\right\|_2$$

$$\overset{(a)}{\leq} (1+\rho_1)\left\|\alpha^{(L)}(\mathbf{x})\right\|_2 \tag{40}$$

$$\overset{(b)}{\leq} (1+\rho_1)\left(1 + L|\phi(0)|\sqrt{m}\right),$$

where (a) follows from $\|v\|_2 \leq \|\mathbf{v}_0\|_2 + \|\mathbf{v} - \mathbf{v}_0\|_2 \leq 1 + \rho_1$, and (b) follows from Lemma A.1. Now, replacing this result back in (39) we obtain $\mathcal{L}(\theta) \leq \frac{1}{n}\sum_{i=1}^{n}(2y_i^2 + 2(1+\rho_1)^2(1+L|\phi(0)|\sqrt{m})^2)$. This finishes the proof for inequality (9).

Now, we observe that $\|\nabla_\theta \mathcal{L}(\theta)\|_2 = \left\|\frac{1}{n}\sum_{i=1}^{n}\ell_i'\nabla_\theta f\right\|_2 \leq \frac{1}{n}\sum_{i=1}^{n}|\ell_i'|\,\|\nabla_\theta f\|_2 \overset{(a)}{\leq} \frac{2\varrho_\theta}{n}\sum_{i=1}^{n}|y_i - \hat{y}_i| \leq 2\varrho_\theta\sqrt{\mathcal{L}(\theta)} \overset{(b)}{\leq} 2\sqrt{\varphi}\varrho_\theta$ where (a) follows from Lemma 4.1 and (b) from inequality (9). This finishes the proof for inequality (10). This completes the proof. $\qquad\square$

## C  Training using restricted strong convexity for WeightNorm

We establish the results from Section 5.

### C.1  Restricted Strong Convexity and Smoothness

**Lemma 5.1** (**RSC for WeightNorm under Square Loss**). *For square loss, under Assumptions 1 and 2, for every $\theta' \in Q_\kappa^\theta \cap B_{\rho_2}^{\mathrm{Euc}}(\theta)$ with $\theta \in \mathbb{R}^p$ and $\mathbf{v}, \mathbf{v}' \in B_{\rho_1}^{\mathrm{Euc}}(\mathbf{v}_0)$,*

$$\mathcal{L}(\theta') \geq \mathcal{L}(\theta) + \langle \theta' - \theta, \nabla_\theta \mathcal{L}(\theta)\rangle + \frac{\alpha_{\theta,\theta'}}{2}\|\theta' - \theta\|_2^2 \,, \tag{12}$$

*with*

$$\begin{aligned}
\alpha_{\theta,\theta'} = &\frac{\kappa^2}{2}\frac{\|\nabla_\theta \mathcal{L}(\theta)\|_2^2}{\mathcal{L}(\theta)}\\
&- \mathcal{O}\left(\left(1 + \frac{\sqrt{L}}{\|\bar{W}\|_2}\right)\frac{(1+\rho_2)(1+\rho_1)^3 L^3 A(\frac{1}{\sqrt{m}}, L^2, |\phi(0)|)B(1, L, |\phi(0)|\sqrt{m})}{\min\left\{\|\bar{f}(W', W)_\xi\|_2, \|\bar{f}(W', W)_\xi\|_2^2\right\}}\right) \,,
\end{aligned} \tag{13}$$

*where $f(W', W)_\xi := \xi W' + (1-\xi)W$ for some $\xi \in [0,1]$, $A(\frac{1}{\sqrt{m}}, L^2, |\phi(0)|) := \frac{1}{\sqrt{m}} + L^2 \max\{|\phi(0)|, |\phi(0)|^2\}$ and $B(1, L, |\phi(0)|\sqrt{m}) := 1 + L|\phi(0)|\sqrt{m}$. We say that the empirical loss $\mathcal{L}$ satisfies the RSC property w.r.t. $(Q_\kappa^\theta \cap B_{\rho_2}^{\mathrm{Euc}}(\theta), \theta)$ whenever $\alpha_{\theta,\theta'} > 0$.*

*Proof.* For any $\theta' \in Q_{\kappa/2}^t \cap B_{\rho_2}(\theta)$ with $\mathbf{v}' \in B_{\rho_1}^{\mathrm{Euc}}(\mathbf{v}_0)$, by the second order Taylor expansion around $\theta$, we have

$$\mathcal{L}(\theta') = \mathcal{L}(\theta) + \langle \theta' - \theta, \nabla_\theta \mathcal{L}(\theta)\rangle + \frac{1}{2}(\theta' - \theta)^\top \frac{\partial^2 \mathcal{L}(\tilde{\theta})}{\partial\theta^2}(\theta' - \theta) \,,$$

where $\tilde{\theta} = \xi\theta' + (1-\xi)\theta$ for some $\xi \in [0,1]$. We note that $\tilde{\mathbf{v}} \in B_{\rho_1}^{\mathrm{Euc}}(\mathbf{v}_0)$ since, $\|\tilde{\mathbf{v}} - \mathbf{v}_0\|_2 = \|\xi\mathbf{v}' - \xi\mathbf{v}_0 + (1-\xi)\mathbf{v} - (1-\xi)\mathbf{v}_0\|_2 \leq \xi\|\mathbf{v}' - \mathbf{v}_0\|_2 + (1-\xi)\|\mathbf{v} - \mathbf{v}_0\|_2 \leq \rho_1$.

Focusing on the quadratic form in the Taylor expansion, it can be shown that

$$(\theta' - \theta)^\top \frac{\partial^2 \mathcal{L}(\tilde{\theta})}{\partial\theta^2}(\theta' - \theta)$$
$$= \underbrace{\frac{1}{n}\sum_{i=1}^{n}\ell_i''\left\langle \theta' - \theta, \frac{\partial f(\tilde{\theta}; \mathbf{x}_i)}{\partial\theta}\right\rangle^2}_{I_1} + \underbrace{\frac{1}{n}\sum_{i=1}^{n}\ell_i'(\theta' - \theta)^\top \frac{\partial^2 f(\tilde{\theta}; \mathbf{x}_i)}{\partial\theta^2}(\theta' - \theta)}_{I_2} \,,$$

where $\ell_i = \ell(y_i, f(\tilde{\theta}, \mathbf{x}_i))$, $\ell_i' = \left.\frac{\partial\ell(y_i, z)}{\partial z}\right|_{z=f(\tilde{\theta}, \mathbf{x}_i)}$, and $\ell_i'' = \left.\frac{\partial^2\ell(y_i, z)}{\partial z^2}\right|_{z=f(\tilde{\theta}, \mathbf{x}_i)}$. Likewise, we set $\bar{\ell}_i = \ell(y_i, f(\theta, \mathbf{x}_i))$, $\bar{\ell}_i' = \left.\frac{\partial\ell(y_i, z)}{\partial z}\right|_{z=f(\theta, \mathbf{x}_i)}$, and $\bar{\ell}_i'' = \left.\frac{\partial^2\ell(y_i, z)}{\partial z^2}\right|_{z=f(\theta, \mathbf{x}_i)}$ Then, following a similar analysis to (Banerjee et al., 2023),

$$I_1 = \frac{1}{n} \sum_{i=1}^{n} \ell_i'' \left\langle \theta' - \theta, \frac{\partial f(\tilde{\theta}; \mathbf{x}_i)}{\partial \theta} \right\rangle^2$$

$$\overset{(a)}{\geq} \frac{2}{n} \sum_{i=1}^{n} \left\langle \theta' - \theta, \frac{\partial f(\theta; \mathbf{x}_i)}{\partial \theta} \right\rangle^2 - \frac{4}{n} \sum_{i=1}^{n} \left\| \frac{\partial f(\theta; \mathbf{x}_i)}{\partial \theta} \right\|_2 \left\| \frac{\partial f(\tilde{\theta}; \mathbf{x}_i)}{\partial \theta} - \frac{\partial f(\theta; \mathbf{x}_i)}{\partial \theta} \right\|_2$$
$$\times \| \theta' - \theta \|_2^2$$

$$= \frac{2}{n} \sum_{i=1}^{n} \frac{1}{(\bar{\ell}_i')^2} \left\langle \theta' - \theta, \bar{\ell}_i' \frac{\partial f(\theta; \mathbf{x}_i)}{\partial \theta} \right\rangle^2 - \frac{4}{n} \sum_{i=1}^{n} \left\| \frac{\partial f(\theta; \mathbf{x}_i)}{\partial \theta} \right\|_2 \left\| \frac{\partial f(\tilde{\theta}; \mathbf{x}_i)}{\partial \theta} - \frac{\partial f(\theta; \mathbf{x}_i)}{\partial \theta} \right\|_2$$
$$\times \| \theta' - \theta \|_2^2$$

$$\geq \frac{2}{n} \sum_{i=1}^{n} \frac{1}{\sum_{j=1}^{n} (\bar{\ell}_j')^2} \left\langle \theta' - \theta, \bar{\ell}_i' \frac{\partial f(\theta; \mathbf{x}_i)}{\partial \theta} \right\rangle^2 - \frac{4}{n} \sum_{i=1}^{n} \left\| \frac{\partial f(\theta; \mathbf{x}_i)}{\partial \theta} \right\|_2$$
$$\times \left\| \frac{\partial f(\tilde{\theta}; \mathbf{x}_i)}{\partial \theta} - \frac{\partial f(\theta; \mathbf{x}_i)}{\partial \theta} \right\|_2 \| \theta' - \theta \|_2^2$$

$$\overset{(b)}{=} \frac{1}{2n^2 \mathcal{L}(\theta)} \sum_{i=1}^{n} \left\langle \theta' - \theta, \bar{\ell}_i' \frac{\partial f(\theta; \mathbf{x}_i)}{\partial \theta} \right\rangle^2 - \frac{4}{n} \sum_{i=1}^{n} \left\| \frac{\partial f(\theta; \mathbf{x}_i)}{\partial \theta} \right\|_2 \left\| \frac{\partial f(\tilde{\theta}; \mathbf{x}_i)}{\partial \theta} - \frac{\partial f(\theta; \mathbf{x}_i)}{\partial \theta} \right\|_2$$
$$\times \| \theta' - \theta \|_2^2$$

$$\overset{(c)}{\geq} \frac{1}{2\mathcal{L}(\theta)} \left\langle \theta' - \theta, \frac{1}{n} \sum_{i=1}^{n} \bar{\ell}_i' \frac{\partial f(\theta; \mathbf{x}_i)}{\partial \theta} \right\rangle^2 - \frac{4}{n} \sum_{i=1}^{n} \varrho_\theta \left\| \frac{\partial^2 f(\bar{\theta}; \mathbf{x}_i)}{\partial \theta^2} \right\|_2 \| \tilde{\theta} - \theta \|_2 \| \theta' - \theta \|_2^2$$

$$= \frac{1}{2\mathcal{L}(\theta)} \left\langle \theta' - \theta, \nabla_\theta \mathcal{L}(\theta) \right\rangle^2 - \frac{4}{n} \sum_{i=1}^{n} \varrho_\theta \left\| \frac{\partial^2 f(\bar{\theta}; \mathbf{x}_i)}{\partial \theta^2} \right\|_2 \| \tilde{\theta} - \theta \|_2 \| \theta' - \theta \|_2^2$$

$$\overset{(d)}{\geq} \frac{1}{2\mathcal{L}(\theta)} \left\langle \theta' - \theta, \nabla_\theta \mathcal{L}(\theta) \right\rangle^2 - 4\varrho_\theta \| \theta' - \theta \|_2^3 \cdot \frac{1}{n} \sum_{i=1}^{n} \left\| \frac{\partial^2 f(\bar{\theta}; \mathbf{x}_i)}{\partial \theta^2} \right\|_2$$

$$\overset{(e)}{\geq} \frac{\kappa^2}{2\mathcal{L}(\theta)} \| \nabla_\theta \mathcal{L}(\theta) \|_2^2 \| \theta' - \theta \|_2^2 - 4\varrho_\theta \| \theta' - \theta \|_2^3 \cdot \frac{1}{n} \sum_{i=1}^{n} \left\| \frac{\partial^2 f(\bar{\theta}; \mathbf{x}_i)}{\partial \theta^2} \right\|_2$$

$$= \left( \frac{\kappa^2}{2\mathcal{L}(\theta)} \| \nabla_\theta \mathcal{L}(\theta) \|_2^2 - 4\varrho_\theta \| \theta' - \theta \|_2 \cdot \frac{1}{n} \sum_{i=1}^{n} \left\| \frac{\partial^2 f(\bar{\theta}; \mathbf{x}_i)}{\partial \theta^2} \right\|_2 \right) \| \theta' - \theta \|_2^2 \, ,$$

where $\bar{\theta}$ is some point in the segment that joins $\tilde{\theta}$ and $\theta$; and where (a) is the inequality derived in the proof of (Banerjee et al., 2023, Theorem 5.1); (b) follows by $\sum_{i=1}^{n} (\bar{\ell}_i')^2 = 4 \sum_{i=1}^{n} (f(\theta; \mathbf{x}_i) - y_i)^2 = 4n\mathcal{L}(\theta)$; (c) follows by Jensen's inequality on the first term and on the second term by the use of Lemma 4.1 due to $\tilde{\mathbf{v}} \in B_{\rho_1}^{\text{Spec}}(\mathbf{v}_0)$ and the mean value theorem; (d) follows by $\| \tilde{\theta} - \theta \|_2 = \| \xi \theta' + (1 - \xi)\theta - \theta \|_2 = \xi \| \theta' - \theta \|_2 \leq \| \theta' - \theta \|_2$; (e) follows since $\theta' \in Q_\kappa^\theta$ and from the fact that $p^\top q = \cos(p, q) \| p \| \| q \|$ for any vectors $p, q$.

Regarding $I_2$, note that

$$
\begin{aligned}
I_2 &= \frac{1}{n} \sum_{i=1}^{n} \ell_i'(\theta' - \theta)^\top \frac{\partial^2 f(\tilde{\theta}; \mathbf{x}_i)}{\partial \theta^2}(\theta' - \theta) \\
&\overset{(a)}{\geq} -\left| \sum_{i=1}^{n} \left( \frac{1}{\sqrt{n}} \ell_i' \right) \left( \frac{1}{\sqrt{n}} \|\theta' - \theta\|_2^2 \left\| \frac{\partial^2 f(\tilde{\theta}; \mathbf{x}_i)}{\partial \theta^2} \right\|_2 \right) \right| \\
&\overset{(b)}{\geq} -\left( \frac{1}{n} \|[\ell_i']\|_2^2 \right)^{1/2} \left( \frac{1}{n} \sum_{i=1}^{n} \left\| \frac{\partial^2 f(\tilde{\theta}; \mathbf{x}_i)}{\partial \theta^2} \right\|_2^2 \right)^{1/2} \|\theta' - \theta\|_2^2 \\
&\overset{(c)}{=} -2\sqrt{\mathcal{L}(\tilde{\theta})} \left( \frac{1}{n} \sum_{i=1}^{n} \left\| \frac{\partial^2 f(\tilde{\theta}; \mathbf{x}_i)}{\partial \theta^2} \right\|_2^2 \right)^{1/2} \|\theta' - \theta\|_2^2 \ ,
\end{aligned}
$$

where (a) follows from the Cauchy-Schwartz inequality and the submultiplicative property of the induced norm; (b) follows by Cauchy-Schwartz inequality again; and (c) from $\frac{1}{n}\|[\ell_i']_i\|_2^2 = \frac{4}{n}\sum_{i=1}^{n}(\tilde{y}_i - y_i)^2 = 4\mathcal{L}(\tilde{\theta})$.

Replacing the lower bounds on $I_1$ and $I_2$, and using Lemma 4.1 and the fact that $\theta' \in B_{\rho_2}^{\mathrm{Euc}}(\theta)$, we finally obtain,

$$
\begin{aligned}
(\theta' - \theta)^\top \frac{\partial^2 \mathcal{L}(\tilde{\theta})}{\partial \theta^2}(\theta' - \theta) \geq &\left( \frac{\kappa^2}{2} \frac{\|\nabla_\theta \mathcal{L}(\theta)\|_2^2}{\mathcal{L}(\theta)} - 4\varrho_\theta \rho_2 \frac{1}{n} \sum_{i=1}^{n} \left\| \frac{\partial^2 f(\bar{\theta}; \mathbf{x}_i)}{\partial \theta^2} \right\|_2 \right. \\
&\left. -2\sqrt{\varphi} \left( \frac{1}{n} \sum_{i=1}^{n} \left\| \frac{\partial^2 f(\tilde{\theta}; \mathbf{x}_i)}{\partial \theta^2} \right\|_2^2 \right)^{1/2} \right) \|\theta' - \theta\|_2^2 \ .
\end{aligned}
$$

Now, we note that we can use Theorem 4.1 to upper bound the norm of the Hessian of the predictor because $\bar{\mathbf{v}}$ belongs to the segment joining $\tilde{\mathbf{v}}$ and $\mathbf{v}$, two vectors which belong to the convex set $B_{\rho_1}^{\mathrm{Euc}}(\mathbf{v}_0)$, and so it follows that $\bar{\mathbf{v}} \in B_{\rho_1}^{\mathrm{Euc}}(\mathbf{v}_0)$.

Then, using Theorem 4.1, Lemma 4.1, and Proposition 4.1,

$$4\varrho_\theta \rho_2 \frac{1}{n} \sum_{i=1}^n \left\| \frac{\partial^2 f(\bar{\theta}; \mathbf{x}_i)}{\partial \theta^2} \right\|_2 + 2\sqrt{\varphi} \left( \frac{1}{n} \sum_{i=1}^n \left\| \frac{\partial^2 f(\tilde{\theta}; \mathbf{x}_i)}{\partial \theta^2} \right\|_2^2 \right)^{1/2}$$

$$\leq (1+\rho_1)L^3 (\frac{1}{\sqrt{m}} + L^2 \max\{|\phi(0)|, |\phi(0)|^2\}) \mathcal{O} \left( \frac{4\varrho_\theta \rho_2}{\min\left\{ \min_{\substack{i\in[m]\\l\in[L]}} \left\| \bar{W}_i^{(l)} \right\|_2, \min_{\substack{i\in[m]\\l\in[L]}} \left\| \bar{W}_i^{(l)} \right\|_2^2 \right\}} \right.$$

$$\left. + \frac{2\sqrt{\varphi}}{\min\left\{ \min_{\substack{i\in[m]\\l\in[L]}} \left\| \tilde{W}_i^{(l)} \right\|_2, \min_{\substack{i\in[m]\\l\in[L]}} \left\| \tilde{W}_i^{(l)} \right\|_2 \right\}} \right)$$

$$\leq (4\varrho_\theta \rho_2 + 2\sqrt{\varphi}) \mathcal{O} \left( \frac{(1+\rho_1)L^3 (\frac{1}{\sqrt{m}} + L^2 \max\{|\phi(0)|, |\phi(0)|^2\})}{\min\left\{ \min_{\substack{i\in[m]\\l\in[L]}} \left\| \bar{W}_i^{(l)} \right\|_2, \min_{\substack{i\in[m]\\l\in[L]}} \left\| \bar{W}_i^{(l)} \right\|_2^2, \min_{\substack{i\in[m]\\l\in[L]}} \left\| \tilde{W}_i^{(l)} \right\|_2, \min_{\substack{i\in[m]\\l\in[L]}} \left\| \tilde{W}_i^{(l)} \right\|_2^2 \right\}} \right)$$

$$\leq \mathcal{O} \left( (\varrho_\theta \rho_2 + \sqrt{\varphi}) \frac{(1+\rho_1)L^3 (\frac{1}{\sqrt{m}} + L^2 \max\{|\phi(0)|, |\phi(0)|^2\})}{\min\left\{ \min_{\substack{i\in[m]\\l\in[L]}} \left\| \bar{W}_i^{(l)} \right\|_2, \min_{\substack{i\in[m]\\l\in[L]}} \left\| \bar{W}_i^{(l)} \right\|_2^2, \min_{\substack{i\in[m]\\l\in[L]}} \left\| \tilde{W}_i^{(l)} \right\|_2, \min_{\substack{i\in[m]\\l\in[L]}} \left\| \tilde{W}_i^{(l)} \right\|_2^2 \right\}} \right)$$

$$\overset{(a)}{\leq} \mathcal{O} \left( \left( 1 + \frac{\sqrt{L}}{\min_{\substack{i\in[m]\\l\in[L]}} \left\| W_i^{(l)} \right\|_2} \right) \right.$$

$$\times \frac{\max\{(1+\rho_1), \rho_2\}(1+\rho_1)^2 L^3 (\frac{1}{\sqrt{m}} + L^2 \max\{|\phi(0)|, |\phi(0)|^2\})(1+L|\phi(0)|\sqrt{m})}{\min\left\{ \min_{\substack{i\in[m]\\l\in[L]}} \left\| \bar{W}_i^{(l)} \right\|_2, \min_{\substack{i\in[m]\\l\in[L]}} \left\| \bar{W}_i^{(l)} \right\|_2^2, \min_{\substack{i\in[m]\\l\in[L]}} \left\| \tilde{W}_i^{(l)} \right\|_2, \min_{\substack{i\in[m]\\l\in[L]}} \left\| \tilde{W}_i^{(l)} \right\|_2^2 \right\}} \right)$$

$$\leq \mathcal{O} \left( \left( 1 + \frac{\sqrt{L}}{\min_{\substack{i\in[m]\\l\in[L]}} \left\| W_i^{(l)} \right\|_2} \right) \right.$$

$$\times \frac{(1+\rho_2)(1+\rho_1)^3 L^3 (\frac{1}{\sqrt{m}} + L^2 \max\{|\phi(0)|, |\phi(0)|^2\})(1+L|\phi(0)|\sqrt{m})}{\min\left\{ \min_{\substack{i\in[m]\\l\in[L]}} \left\| \bar{W}_i^{(l)} \right\|_2, \min_{\substack{i\in[m]\\l\in[L]}} \left\| \bar{W}_i^{(l)} \right\|_2^2, \min_{\substack{i\in[m]\\l\in[L]}} \left\| \tilde{W}_i^{(l)} \right\|_2, \min_{\substack{i\in[m]\\l\in[L]}} \left\| \tilde{W}_i^{(l)} \right\|_2^2 \right\}} \right)$$

where (a) follows from

$$(\varrho_\theta \rho_2 + \sqrt{\varphi}) \le \mathcal{O}\left(\rho_2(1 + L|\phi(0)|\sqrt{m})\left(1 + \frac{\sqrt{L}(1+\rho_1)}{\min_{\substack{i\in[m]\\l\in[L]}}\left\|W_i^{(l)}\right\|_2}\right) + (1+\rho_1)(1+L|\phi(0)|\sqrt{m})\right)$$

$$\le \mathcal{O}\left(\max\{\rho_2, (1+\rho_1)\}(1 + L|\phi(0)|\sqrt{m})\left(1 + \frac{\sqrt{L}(1+\rho_1)}{\min_{\substack{i\in[m]\\l\in[L]}}\left\|W_i^{(l)}\right\|_2}\right)\right).$$

That completes the proof. $\qquad\square$

**Lemma 5.2 (Smoothness-like property for WeightNorm under Square Loss).** *For square loss, under Assumptions 1 and 2, for every $\theta, \theta' \in \mathbb{R}^p$ with $\mathbf{v}, \mathbf{v}' \in B_{\rho_1}^{\text{Euc}}(\mathbf{v}_0)$,*

$$\mathcal{L}(\theta') \le \mathcal{L}(\theta) + \langle \theta' - \theta, \nabla_\theta \mathcal{L}(\theta) \rangle + \frac{\beta_{\theta,\theta'}}{2}\|\theta' - \theta\|_2^2 , \tag{15}$$

*with*

$$\beta_{\theta,\theta'} \le \mathcal{O}\left(L^4 A(1, L^2, |\phi(0)|)(1+\rho_1)^2 \left(1 + \frac{1}{\min\{\left\|\bar{f}(W', W)_\xi\right\|_2, \left\|\bar{f}(W', W)_\xi\right\|_2^2\}}\right)\right) , \tag{16}$$

*where $f(W', W)_\xi := \xi W' + (1-\xi)W$ for some $\xi \in [0,1]$, $A(1, L^2, |\phi(0)|) = (1 + L^2 \max\{|\phi(0)|, |\phi(0)|^2\}m)^2$.*

*Proof.* By the second order Taylor expansion about $\theta \in \mathbb{R}^p$ with $\bar{\mathbf{v}} \in B_{\rho_1}^{\text{Euc}}(\mathbf{v}_0)$, we have $\mathcal{L}(\theta') = \mathcal{L}(\theta) + \langle \theta' - \theta, \nabla_\theta \mathcal{L}(\theta) \rangle + \frac{1}{2}(\theta' - \theta)^\top \frac{\partial^2 \mathcal{L}(\tilde{\theta})}{\partial \theta^2}(\theta' - \theta)$, where $\tilde{\theta} = \xi \theta' + (1-\xi)\theta$ for some $\xi \in [0,1]$. Similarly to what was proved in Theorem 5.1, $\tilde{\mathbf{v}} \in B_{\rho_1}^{\text{Euc}}(\mathbf{v}_0)$.

Now,

$$(\theta' - \theta)^\top \frac{\partial^2 \mathcal{L}(\tilde{\theta})}{\partial \theta^2}(\theta' - \theta)$$

$$= \underbrace{\frac{1}{n}\sum_{i=1}^n \ell_i'' \left\langle \theta' - \theta, \frac{\partial f(\tilde{\theta}; \mathbf{x}_i)}{\partial \theta}\right\rangle^2}_{I_1} + \underbrace{\frac{1}{n}\sum_{i=1}^n \ell_i'(\theta' - \theta)^\top \frac{\partial^2 f(\tilde{\theta}; \mathbf{x}_i)}{\partial \theta^2}(\theta' - \theta)}_{I_2} ,$$

where $\ell_i = \ell(y_i, f(\tilde{\theta}, \mathbf{x}_i))$, $\ell_i' = \left.\frac{\partial \ell(y_i, z)}{\partial z}\right|_{z=f(\tilde{\theta}, \mathbf{x}_i)}$, and $\ell_i'' = \left.\frac{\partial^2 \ell(y_i, z)}{\partial z^2}\right|_{z=f(\tilde{\theta}, \mathbf{x}_i)}$. It is easy to note that $I_1 \le 2\varrho_{\tilde{\theta}}^2 \|\theta' - \theta\|_2^2$. For $I_2$, we have that

$$I_2 = \frac{1}{n}\sum_{i=1}^n \ell_i'(\theta' - \theta)^\top \frac{\partial^2 f(\tilde{\theta}; \mathbf{x}_i)}{\partial \theta^2}(\theta' - \theta)$$

$$\le \left|\sum_{i=1}^n \left(\frac{1}{\sqrt{n}}\ell_i'\right)\left(\frac{1}{\sqrt{n}}\|\theta' - \theta\|_2^2 \left\|\frac{\partial^2 f(\tilde{\theta}; \mathbf{x}_i)}{\partial \theta^2}\right\|_2\right)\right|$$

$$\overset{(a)}{\le} \left(\frac{1}{n}\|[\ell_i']_i\|_2^2\right)^{1/2}\left(\frac{1}{n}\sum_{i=1}^n \|\theta' - \theta\|_2^4 \left\|\frac{\partial^2 f(\tilde{\theta}; \mathbf{x}_i)}{\partial \theta^2}\right\|_2^2\right)^{1/2}$$

$$\overset{(b)}{\le} 2\sqrt{\mathcal{L}(\tilde{\theta})}\left(\frac{1}{n}\sum_{i=1}^n \left\|\frac{\partial^2 f(\tilde{\theta}; \mathbf{x}_i)}{\partial \theta^2}\right\|_2^2\right)^{1/2}\|\theta' - \theta\|_2^2 ,$$

where (a) follows by Cauchy-Schwartz, and (b) from $\frac{1}{n}\|[\ell'_i]_i\|_2^2 = \frac{4}{n}\sum_{i=1}^n(\tilde{y}_i - y_i)^2 = 4\mathcal{L}(\tilde{\theta})$. Replacing the bounds on $I_1$ and $I_2$ on the original expression, and using Lemma 4.1,

$$(\theta' - \theta)^\top \frac{\partial^2 \mathcal{L}(\tilde{\theta})}{\partial \theta^2}(\theta' - \theta) \leq \left[2\varrho_{\tilde{\theta}}^2 + 2\sqrt{\varphi}\left(\frac{1}{n}\sum_{i=1}^n\left\|\frac{\partial^2 f(\tilde{\theta};\mathbf{x}_i)}{\partial\theta^2}\right\|_2^2\right)^{1/2}\right]\|\theta' - \theta\|_2^2 \ .$$

Now, note that from the proof of Lemma 4.1,

$$\varrho_{\tilde{\theta}}^2 \leq \mathcal{O}\left((1 + L^2|\phi(0)|^2 m)\left(1 + \frac{L(1+\rho_1)^2}{\min_{\substack{i\in[m]\\l\in[L]}}\left\|\tilde{W}_i^{(l)}\right\|_2^2}\right)\right)$$

and so,

$$2\varrho_{\tilde{\theta}}^2 + 2\sqrt{\varphi}\left(\frac{1}{n}\sum_{i=1}^n\left\|\frac{\partial^2 f(\tilde{\theta};\mathbf{x}_i)}{\partial\theta^2}\right\|_2^2\right)^{1/2}$$

$$\leq \mathcal{O}\left((1 + L^2|\phi(0)|^2 m)\left(1 + \frac{L(1+\rho_1)^2}{\min_{\substack{i\in[m]\\l\in[L]}}\left\|\tilde{W}_i^{(l)}\right\|_2^2}\right)\right)$$

$$+ \mathcal{O}\left(\frac{(1+\rho_1)(1 + L|\phi(0)|\sqrt{m})\cdot(1+\rho_1)L^3(\frac{1}{\sqrt{m}} + L^2\max\{|\phi(0)|, |\phi(0)|^2\})}{\min\{\min_{\substack{i\in[m]\\l\in[L]}}\left\|\tilde{W}_i^{(l)}\right\|_2, \min_{\substack{i\in[m]\\l\in[L]}}\left\|\tilde{W}_i^{(l)}\right\|_2^2\}}\right)$$

$$\leq \mathcal{O}\left(L^3(1 + L^2\max\{|\phi(0)|, |\phi(0)|^2\}m)(1+\rho_1)^2\left(1 + \frac{L}{\min_{\substack{i\in[m]\\l\in[L]}}\left\|\tilde{W}_i^{(l)}\right\|_2^2}\right.\right.$$

$$\left.\left.+ \frac{(\frac{1}{\sqrt{m}} + L^2\max\{|\phi(0)|, |\phi(0)|^2\})}{\min\{\min_{\substack{i\in[m]\\l\in[L]}}\left\|\tilde{W}_i^{(l)}\right\|_2, \min_{\substack{i\in[m]\\l\in[L]}}\left\|\tilde{W}_i^{(l)}\right\|_2^2\}}\right)\right)$$

$$\leq \mathcal{O}\left(L^4(1 + L^2\max\{|\phi(0)|, |\phi(0)|^2\}m)^2(1+\rho_1)^2\left(1 + \frac{1}{\min\{\min_{\substack{i\in[m]\\l\in[L]}}\left\|\tilde{W}_i^{(l)}\right\|_2, \min_{\substack{i\in[m]\\l\in[L]}}\left\|\tilde{W}_i^{(l)}\right\|_2^2\}}\right)\right) \ .$$

This completes the proof. $\qquad\square$

### C.2 About the adaptation of (Banerjee et al., 2023, Theorem 5.3)

To adapt (Banerjee et al., 2023, Theorem 5.3) to our setting, we need to adapt the supporting lemma (Banerjee et al., 2023, Lemma 5.1) to our setting as follows:

**Lemma C.1 (Auxiliary result using RSC, (Banerjee et al., 2023, Lemma 5.1)).** *Let $B_t := Q_\kappa^{\theta_t} \cap B_{\rho_2}^{\text{Euc}}(\theta_t) \cap \{\theta \in \mathbb{R}^p \mid \mathbf{v} \in B_{\rho_1}^{\text{Euc}}(\mathbf{v}_0)\}$ and $\overline{\theta}_t \in \operatorname{arginf}_{\theta \in B_t}\mathcal{L}(\theta)$. Using the setting of Lemma 5.1, if $\alpha_{\theta_t,\overline{\theta}_t} > 0$, then*

$$\mathcal{L}(\theta_t) - \inf_{\theta \in B_t}\mathcal{L}(\theta) \leq \frac{1}{2\alpha_{\theta_t,\overline{\theta}_t}}\|\nabla_\theta\mathcal{L}(\theta_t)\|_2^2 \ . \tag{41}$$

*Proof.* By Lemma 5.1 we have

$$\mathcal{L}(\bar{\theta}_t) \geq \mathcal{L}(\theta_t) + \langle \bar{\theta}_t - \theta_t, \nabla_\theta \mathcal{L}(\theta_t) \rangle + \frac{\alpha_{\theta_t, \bar{\theta}_t}}{2} \|\bar{\theta}_t - \theta_t\|_2^2 \ . \tag{42}$$

Now, let us take $\alpha_{\theta_t, \bar{\theta}_t}$ and define

$$\hat{\mathcal{L}}_t(\theta) := \mathcal{L}(\theta_t) + \langle \theta - \theta_t, \nabla_\theta \mathcal{L}(\theta_t) \rangle + \frac{\alpha_{\theta_t, \bar{\theta}_t}}{2} \|\theta - \theta_t\|_2^2 \ .$$

Note that $\hat{\mathcal{L}}_t(\theta)$ is minimized at $\hat{\theta}_{t+1} := \theta_t - \nabla_\theta \mathcal{L}(\theta_t)/\alpha_{\theta_t, \bar{\theta}_t}$ and the minimum value is:

$$\inf_{\theta \in \mathbb{R}^p} \hat{\mathcal{L}}_{\theta_t}(\theta) = \hat{\mathcal{L}}_t(\hat{\theta}_{t+1}) = \mathcal{L}(\theta_t) - \frac{1}{2\alpha_{\theta_t, \bar{\theta}_t}} \|\nabla_\theta \mathcal{L}(\theta_t)\|_2^2 \ .$$

Then, we have

$$\inf_{\theta \in B_t} \mathcal{L}(\theta) = \mathcal{L}(\bar{\theta}_t) \overset{(a)}{\geq} \hat{\mathcal{L}}_t(\bar{\theta}_t) \geq \inf_{\theta \in \mathbb{R}^p} \hat{\mathcal{L}}_t(\theta) = \mathcal{L}(\theta_t) - \frac{1}{2\alpha_{\theta_t, \bar{\theta}_t}} \|\nabla_\theta \mathcal{L}(\theta_t)\|_2^2 \ ,$$

where (a) follows from (42). Rearranging the terms completes the proof. $\qquad\square$

Finally, we note that Theorem 5.1 makes the claim that $\frac{\alpha_{\theta_t, \bar{\theta}_t}}{\beta_{\theta_t}} \in (0, 1]$ (using the notation from the same theorem). We now provide the proof for this result.

**Proposition C.1 (The RSC to smoothness ratio).** *Consider the setting of Theorem 5.1. Then,* $\frac{\alpha_{\theta_t, \bar{\theta}_t}}{\beta_{\theta_t}} \in (0, 1]$.

*Proof.* From Lemma C.1 we have

$$\mathcal{L}(\theta_t) - \mathcal{L}(\bar{\theta}_t) \leq \frac{1}{2\alpha_{\theta_t, \bar{\theta}_t}} \|\nabla_\theta \mathcal{L}(\theta_t)\|_2^2 . \tag{43}$$

Now, using Lemma 5.2, we obtain

$$\begin{aligned} \mathcal{L}(\theta_{t+1}) &\leq \mathcal{L}(\theta_t) + \langle \theta_{t+1} - \theta_t, \nabla_\theta \mathcal{L}(\theta_t) \rangle + \frac{\beta_{\theta_t}}{2} \|\theta_{t+1} - \theta_t\|_2^2 \\ &= \mathcal{L}(\theta_t) - \eta_t \|\nabla_\theta \mathcal{L}(\theta_t)\|_2^2 + \frac{\beta_{\theta_t}}{2} \eta_t^2 \|\nabla_\theta \mathcal{L}(\theta_t)\|_2^2 \\ &= \mathcal{L}(\theta_t) - \frac{1}{2\beta_{\theta_t}} \|\nabla_\theta \mathcal{L}(\theta_t)\|_2^2 \end{aligned} \tag{44}$$

where the last equality follows from $\eta_t = \frac{1}{\beta_{\theta_t}}$.

Now, we notice that $\theta_{t+1} \in Q_\kappa^{\theta_t}$ by Definition 5.1, $\theta_{t+1} \in B_{\rho_2}^{\text{Euc}}(\theta_t) \cap \{\theta \in \mathbb{R}^p \,|\, \mathbf{v} \in B_{\rho_1}^{\text{Euc}}(\mathbf{v}_0)\}$ by Assumption 3, and so $\theta_{t+1} \in B_t$. This implies that $\mathcal{L}(\bar{\theta}_t) \leq \mathcal{L}(\theta_{t+1})$. Thus, using (44), we obtain

$$\frac{1}{2\beta_{\theta_t}} \|\nabla_\theta \mathcal{L}(\theta_t)\|_2^2 \leq \mathcal{L}(\theta_t) - \mathcal{L}(\bar{\theta}_t) \ .$$

Using this inequality along with (43) leads to $\frac{\alpha_{\theta_t, \bar{\theta}_t}}{\beta_{\theta_t}} \leq 1$. Finally, by assumption we have $\alpha_{\theta_t, \bar{\theta}_t} > 0$ and by (17) we have $\beta_{\theta_t} > 0$, and so $\frac{\alpha_{\theta_t, \bar{\theta}_t}}{\beta_{\theta_t}} > 0$. This completes the proof. $\qquad\square$

### C.3 Distance between consecutive iterates

**Proposition C.2** (**Bound on the distance between consecutive iterates for WeightNorm under** $\phi(0) = 0$)**.** *Under Assumptions 1 and 2, and assuming* (**A3.1**) *from Assumption 3 and that the activation function satisfies* $\phi(0) = 0$, *we have*

$$\|\theta_{t+1} - \theta_t\|_2 \leq \eta_t \cdot \mathcal{O}\left((1 + \rho_1)^2 \left(1 + \frac{\sqrt{L}}{\sqrt{m}\,\|\bar{W}_t\|_2}\right)\right).$$

*Proof.* We have that $\|\theta_{t+1} - \theta_t\|_2 = \|\theta_t - \eta_t \nabla_\theta \mathcal{L}(\theta_t) - \theta_t\|_2 = \eta_t \|\nabla_\theta \mathcal{L}(\theta_t)\|_2 \overset{(a)}{\leq} \eta_t \cdot 2\varrho_{\theta_t}\sqrt{\varphi} \overset{(b)}{\leq} \eta_t \cdot \mathcal{O}\left((1 + \rho_1)\left(1 + \frac{\sqrt{L}(1+\rho_1)}{\sqrt{m}\|\bar{W}_t\|_2}\right)\right) \leq \eta_t \cdot \mathcal{O}\left((1 + \rho_1)^2 \left(1 + \frac{\sqrt{L}}{\sqrt{m}\|\bar{W}_t\|_2}\right)\right)$, where (a) follows from using (10) from Proposition 4.1 and the definitions of $\varrho_{\theta_t}$ and $\varphi$ presented therein; and (b) follows from Corollary 4.2 and Corollary 4.3. $\qquad\square$

## D Comparison between networks with and without WeightNorm

We summarize in Table 1 the comparison between our bounds and optimization results and those for networks without WeightNorm by Banerjee et al. (2023).

| with WeightNorm | without WeightNorm |
|---|---|
| Deterministic results | High probability results |
| Hidden layer weights can take any value | Hidden layer weights restricted to a neighborhood around their initialization |
| Appearance of normalization terms, i.e., the minimum weight vector norm, in the Hessian bounds and the RSC and smoothness parameters | No appearance of such normalization terms |
| The polynomial dependence on the depth in our bounds and parameters is independent from the initialization variance of the weights | Polynomial dependence on the depth requires careful choice of initialization variance |
| Lipschitz constant bound is independent from the depth | Lipschitz constant bound exponentially grows with the depth unless a careful choice of initialization variance is done |

Table 1: Comparison of results for neural networks with and without WeightNorm.

## E Generalization bound for WeightNorm

We first state the following auxiliary technical result, which is based on (Golowich et al., 2018, Lemma 1). We let $\mathcal{A}^{(l)} := \{(W^{(1)}, W^{(2)}, \ldots, W^{(l)}) \in \mathbb{R}^{m \times d} \times \mathbb{R}^{m \times m} \times \cdots \times \mathbb{R}^{m \times m}\}$.

**Lemma E.1.** Under Assumptions 1 and 2, assuming the activation function satisfies $\phi(0) = 0$, consider any WeightNorm network $f(\theta; \cdot)$ of the form (1) with any fixed $\theta \in \mathbb{R}^p$ and $\mathbf{v} \in B_{\rho_1}^{\text{Euc}}(\mathbf{v}_0)$, along with some input data set $\{\mathbf{x}_i\}_{i=1}^n$. Consider also any convex and monotonically increasing function $g : \mathbb{R} \to [0, \infty)$. For any

$l \in \{1, \ldots, L-1\}$,

$$\mathbb{E}_{\{\epsilon_i\}_{i=1}^n}\left[\sup_{\substack{\bar{W}\in\mathcal{A}^{(l)} \\ w\in\mathbb{R}^m}} g\left(\left|\sum_{i=1}^n \epsilon_i \phi\left(\frac{1}{\sqrt{m}}\frac{w^\top}{\|w\|}\alpha^{(l)}(\mathbf{x}_i)\right)\right|\right)\right]$$

$$\leq 2\,\mathbb{E}_{\{\epsilon_i\}_{i=1}^n}\left[\sup_{\bar{W}\in\mathcal{A}^{(l)}} g\left(\frac{1}{\sqrt{m}}\left\|\sum_{i=1}^n \epsilon_i\alpha^{(l)}(\mathbf{x}_i)\right\|_2\right)\right] \quad (45)$$

*Proof.* Since $g$ is positive, $g(|z|) \leq g(z) + g(-z)$, and so

$$\mathbb{E}_{\{\epsilon_i\}_{i=1}^n}\left[\sup_{\substack{\bar{W}\in\mathcal{A}^{(l)} \\ w\in\mathbb{R}^m}} g\left(\left|\sum_{i=1}^n \epsilon_i \phi\left(\frac{1}{\sqrt{m}}\frac{w^\top}{\|w\|}\alpha^{(l)}(\mathbf{x}_i)\right)\right|\right)\right]$$

$$\leq \mathbb{E}_{\{\epsilon_i\}_{i=1}^n}\left[\sup_{\substack{\bar{W}\in\mathcal{A}^{(l)} \\ w\in\mathbb{R}^m}} g\left(\sum_{i=1}^n \epsilon_i \phi\left(\frac{1}{\sqrt{m}}\frac{w^\top}{\|w\|}\alpha^{(l)}(\mathbf{x}_i)\right)\right)\right]$$

$$+ \mathbb{E}_{\{\epsilon_i\}_{i=1}^n}\left[\sup_{\substack{\bar{W}\in\mathcal{A}^{(l)} \\ w\in\mathbb{R}^m}} g\left(-\left(\sum_{i=1}^n \epsilon_i \phi\left(\frac{1}{\sqrt{m}}\frac{w^\top}{\|w\|}\alpha^{(l)}(\mathbf{x}_i)\right)\right)\right)\right]$$

$$\overset{(a)}{=} 2\,\mathbb{E}_{\{\epsilon_i\}_{i=1}^n}\left[\sup_{\substack{\bar{W}\in\mathcal{A}^{(l)} \\ w\in\mathbb{R}^m}} g\left(\sum_{i=1}^n \epsilon_i \phi\left(\frac{1}{\sqrt{m}}\frac{w^\top}{\|w\|}\alpha^{(l)}(\mathbf{x}_i)\right)\right)\right]$$

$$\overset{(b)}{=} 2\,\mathbb{E}_{\{\epsilon_i\}_{i=1}^n}\left[g\left(\sup_{\substack{\bar{W}\in\mathcal{A}^{(l)} \\ w\in\mathbb{R}^m}} \sum_{i=1}^n \epsilon_i \phi\left(\frac{1}{\sqrt{m}}\frac{w^\top}{\|w\|}\alpha^{(l)}(\mathbf{x}_i)\right)\right)\right]$$

$$\overset{(c)}{\leq} 2\,\mathbb{E}_{\{\epsilon_i\}_{i=1}^n}\left[g\left(\sup_{\substack{\bar{W}\in\mathcal{A}^{(l)} \\ w\in\mathbb{R}^m}} \sum_{i=1}^n \epsilon_i \left(\frac{1}{\sqrt{m}}\frac{w^\top}{\|w\|}\alpha^{(l)}(\mathbf{x}_i)\right)\right)\right]$$

$$= 2\,\mathbb{E}_{\{\epsilon_i\}_{i=1}^n}\left[\sup_{\substack{\bar{W}\in\mathcal{A}^{(l)} \\ w\in\mathbb{R}^m}} g\left(\sum_{i=1}^n \epsilon_i \left(\frac{1}{\sqrt{m}}\frac{w^\top}{\|w\|}\alpha^{(l)}(\mathbf{x}_i)\right)\right)\right]$$

$$\overset{(d)}{\leq} 2\,\mathbb{E}_{\{\epsilon_i\}_{i=1}^n}\left[\sup_{\substack{\bar{W}\in\mathcal{A}^{(l)} \\ w\in\mathbb{R}^m}} g\left(\frac{1}{\sqrt{m}}\left|\sum_{i=1}^n \epsilon_i\left(\frac{w^\top}{\|w\|}\alpha^{(l)}(\mathbf{x}_i)\right)\right|\right)\right]$$

$$= 2\,\mathbb{E}_{\{\epsilon_i\}_{i=1}^n}\left[\sup_{\bar{W}\in\mathcal{A}^{(l)}} g\left(\frac{1}{\sqrt{m}}\left\|\sum_{i=1}^n \epsilon_i\alpha^{(l)}(\mathbf{x}_i)\right\|_2\right)\right],$$

where (a) follows from the fact that $\epsilon_i$ and $-\epsilon_i$ have the same (symmetrical) distribution; (b) from $g$ being monotonically increasing; (c) from (Ledoux and Talagrand, 1991, equation (4.20)) which makes use of $g$ being convex and increasing, and of $\phi$ being 1-Lipschitz and $\phi(0) = 0$; and (d) follows from the increasing monotonicity of $g$. $\qquad\square$

We now prove our generalization result.

**Theorem 6.1 (Generalization Bound for WeightNorm under Square Loss).** *Consider the square loss and the training set $S = \{(x_i, y_i) \overset{i.i.d.}{\sim} \mathcal{D}, i \in [n]\}$ and $|y| \leq 1$ for any $y \sim \mathcal{D}_y$ with probability one. Under*

*Assumptions 1 and 2, assuming the activation function satisfies $\phi(0) = 0$, with probability at least $(1 - \delta)$ over the choice of the training data $\mathbf{x}_i \sim \mathcal{D}_{\mathbf{x}}, i \in [n]$, for any WeightNorm network $f(\theta; \cdot)$ of the form (1) with any fixed $\theta \in \mathbb{R}^p$ and $\mathbf{v} \in B_{\rho_1}^{Euc}(\mathbf{v}_0)$, we have*

$$\mathcal{L}_D(\theta) - \mathcal{L}_S(\theta) \leq 4(2 + \rho_1)(1 + \rho_1)\frac{\sqrt{2\log(2)L} + 1}{\sqrt{n}} + 2(1 + (1 + \rho_1)^2)\frac{\sqrt{2\log(2/\delta)}}{\sqrt{n}}. \tag{19}$$

*Proof.* Recall that for a class of functions $\mathcal{F}$, the Rademacher complexity is given by

$$R_n(\mathcal{F}) := \mathbb{E}_{\{\mathbf{x}_i\}_{i=1}^n, \{\epsilon_i\}_{i=1}^n} \left[ \sup_{f \in \mathcal{F}} \left| \frac{1}{n} \sum_{i=1}^n \epsilon_i f(\mathbf{x}_i) \right| \right] , \tag{46}$$

where samples $\mathbf{x}_i$, $i \in [n]$, are drawn i.i.d. from $\mathcal{D}_{\mathbf{x}}$, and $\epsilon_i$, $i \in [n]$, are drawn i.i.d. from the Rademacher distribution, i.e., $+1$ or $-1$ with probability $\frac{1}{2}$. Assuming $\|f\|_\infty \leq B, \forall f \in \mathcal{F}$, a standard uniform convergence argument implies

$$\mathbb{P}\left( \sup_{f \in \mathcal{F}} \left| \frac{1}{n} \sum_{i=1}^n f(\mathbf{x}_i) - \mathbb{E}_{\mathbf{x} \sim \mathcal{D}_{\mathbf{x}}}[f(\mathbf{x})] \right| \geq 2R_n(\mathcal{F}) + t \right) \leq 2\exp\left( -\frac{nt^2}{2B^2} \right) . \tag{47}$$

In other words, with probability at least $(1 - \delta)$ over the draw of the samples, we have

$$\forall f \in \mathcal{F}, \quad \mathbb{E}_{\mathbf{x} \sim \mathcal{D}_{\mathbf{x}}}[f(\mathbf{x})] \leq \frac{1}{n}\sum_{i=1}^n f(\mathbf{x}_i) + 2R_n(\mathcal{F}) + B\sqrt{\frac{2\log\frac{2}{\delta}}{n}} . \tag{48}$$

Now, consider a given loss function $\ell$ so that, for a given constant $y$ such that $|y| \leq 1$, it computes $\ell(y, f(\mathbf{x}))$. If $\hat{y} \mapsto \ell(y, \hat{y})$ has a Lipschitz constant of $\lambda$, then by using the Rademacher contraction lemma (e.g., an adaptation from (Shalev-Shwartz and Ben-David, 2014, Lemma 26.9)), the uniform convergence result can be extended to the loss function: with probability at least $(1 - \delta)$ over the draw of the samples, we have

$$\forall f \in \mathcal{F}, \quad \mathcal{L}_{\mathcal{D}}[f] \leq \mathcal{L}_S[f] + 2\lambda R_n(\mathcal{F}) + \bar{B}\sqrt{\frac{2\log\frac{2}{\delta}}{n}} . \tag{49}$$

with $\mathcal{L}_{\mathcal{D}}[f] := \mathbb{E}_{(\mathbf{x},y) \sim \mathcal{D}}[\ell(y, f(\mathbf{x}))]$, $\mathcal{L}_S[f] := \frac{1}{n}\sum_{i=1}^n \ell(y_i, f(\mathbf{x}_i))$, and $\sup_{\substack{f \in \mathcal{F} \\ |y| \leq 1}} |\ell(y, f(\cdot))| \leq \bar{B}$.

Now we put everything according to the context of our setting. We let $\mathcal{F}$ be the set of all WeightNorm networks $f(\theta; \cdot)$ of the form (1) with any fixed $\theta \in \mathbb{R}^p$ and $\mathbf{v} \in B_{\rho_1}^{\text{Euc}}(\mathbf{v}_0)$ under Assumptions 1 and 2, and assuming the activation function satisfies $\phi(0) = 0$. We also set $\mathcal{L}_{\mathcal{D}}(\theta) := \mathcal{L}_{\mathcal{D}}[f(\theta; \cdot)]$ and $\mathcal{L}_S(\theta) := \mathcal{L}_S[f(\theta; \cdot)]$.

Consider any $\mathbf{x} \in \mathbb{R}^d$ with $\|\mathbf{x}\|_2 = 1$. Now, using the assumption $\phi(0) = 0$, we obtain $|f(\theta; \mathbf{x})| \leq (1 + \rho_1)$ from equation (40), and so $\ell(y, f(\theta; \mathbf{x})) = (y - f(\theta; \mathbf{x}))^2 \leq 2|y|^2 + 2|f(\theta; \mathbf{x})|^2 \leq 2(1 + (1 + \rho_1)^2)$, i.e.,

$$B = 2(1 + (1 + \rho_1)^2) . \tag{50}$$

Likewise, this allows us to obtain, for $|y| \leq 1$,

$$\left. \frac{d\ell(y, \hat{y})}{d\hat{y}} \right|_{\hat{y} = f(\theta; \mathbf{x})} \leq 2(|y| + |f(\theta, \mathbf{x})|) \leq 2(1 + (1 + \rho_1)) = 2(2 + \rho_1),$$

and so

$$\lambda = 2(2 + \rho_1) . \tag{51}$$

Next, we focus on bounding $R_n(\mathcal{F})$.

$$
\begin{aligned}
R_n(\mathcal{F}) &= \frac{1}{n} E_{\{\mathbf{x}_i\}_{i=1}^n, \{\epsilon_i\}_{i=1}^n} \left[ \sup_{f \in \mathcal{F}} \left| \sum_{i=1}^n \epsilon_i f(\mathbf{x}_i) \right| \right] \\
&= \frac{1}{n} E_{\{\mathbf{x}_i\}_{i=1}^n, \{\epsilon_i\}_{i=1}^n} \left[ \sup_{\substack{\bar{W} \in \mathcal{A}^{(L)} \\ \mathbf{v} \in \mathbb{R}^m, \|\mathbf{v}\|_2 \leq 1+\rho_1}} \left| \mathbf{v}^\top \left( \sum_{i=1}^n \epsilon_i \alpha^{(L)}(\mathbf{x}_i) \right) \right| \right] \\
&= \frac{(1+\rho_1)}{n} E_{\{\mathbf{x}_i\}_{i=1}^n, \{\epsilon_i\}_{i=1}^n} \left[ \sup_{\bar{W} \in \mathcal{A}^{(L)}} \left\| \sum_{i=1}^n \epsilon_i \alpha^{(L)}(\mathbf{x}_i) \right\|_2 \right] \\
&\leq \frac{(1+\rho_1)}{n} E_{\{\mathbf{x}_i\}_{i=1}^n, \{\epsilon_i\}_{i=1}^n} \left[ \sup_{\bar{W} \in \mathcal{A}^{(L)}} \sqrt{m} \left\| \sum_{i=1}^n \epsilon_i \alpha^{(L)}(\mathbf{x}_i) \right\|_\infty \right] \\
&= \frac{(1+\rho_1)}{n} E_{\{\mathbf{x}_i\}_{i=1}^n, \{\epsilon_i\}_{i=1}^n} \left[ \sup_{\bar{W} \in \mathcal{A}^{(L)}} \max_{k \in [m]} \sqrt{m} \left| \sum_{i=1}^n \epsilon_i \phi \left( \frac{1}{\sqrt{m}} \frac{(W_k^{(L)})^\top}{\|W_k^{(L)}\|_2} \alpha^{(L-1)}(\mathbf{x}_i) \right) \right| \right] \\
&= \frac{(1+\rho_1)}{n} E_{\{\mathbf{x}_i\}_{i=1}^n, \{\epsilon_i\}_{i=1}^n} \left[ \sup_{\bar{W} \in \mathcal{A}^{(L-1)}} \sup_{w \in \mathbb{R}^m} \sqrt{m} \left| \sum_{i=1}^n \epsilon_i \phi \left( \frac{1}{\sqrt{m}} \frac{w^\top}{\|w\|_2} \alpha^{(L-1)}(\mathbf{x}_i) \right) \right| \right].
\end{aligned}
\tag{52}
$$

Then,

$$
\begin{aligned}
\Longrightarrow \frac{n}{(1+\rho_1)} R_n(\mathcal{F}) &\leq E_{\{\mathbf{x}_i\}_{i=1}^n, \{\epsilon_i\}_{i=1}^n} \left[ \sup_{\bar{W} \in \mathcal{A}^{(L-1)}} \sup_{w \in \mathbb{R}^m} \sqrt{m} \left| \sum_{i=1}^n \epsilon_i \phi \left( \frac{1}{\sqrt{m}} \frac{w^\top}{\|w\|_2} \alpha^{(L-1)}(\mathbf{x}_i) \right) \right| \right] \\
&= \frac{1}{\lambda} \log \exp \left( \lambda E_{\{\mathbf{x}_i\}_{i=1}^n, \{\epsilon_i\}_{i=1}^n} \left[ \sup_{\bar{W} \in \mathcal{A}^{(L-1)}} \sup_{w \in \mathbb{R}^m} \sqrt{m} \left| \sum_{i=1}^n \epsilon_i \phi \left( \frac{1}{\sqrt{m}} \frac{w^\top}{\|w\|_2} \alpha^{(L-1)}(\mathbf{x}_i) \right) \right| \right] \right) \\
&\overset{(a)}{\leq} \frac{1}{\lambda} \log E_{\{\mathbf{x}_i\}_{i=1}^n, \{\epsilon_i\}_{i=1}^n} \left[ \sup_{\substack{\bar{W} \in \mathcal{A}^{(L-1)} \\ w \in \mathbb{R}^m}} \exp \left( \lambda \sqrt{m} \left| \sum_{i=1}^n \epsilon_i \phi \left( \frac{1}{\sqrt{m}} \frac{w^\top}{\|w\|_2} \alpha^{(L-1)}(\mathbf{x}_i) \right) \right| \right) \right]
\end{aligned}
\tag{53}
$$

where (a) follows from Jensen's inequality. Now, using Lemma E.1 with the function $g(z) = \exp(\lambda \sqrt{m} \cdot z)$,

$$
\begin{aligned}
\frac{1}{\lambda} \log E_{\{\mathbf{x}_i\}_{i=1}^n, \{\epsilon_i\}_{i=1}^n} &\left[ \sup_{\substack{\bar{W} \in \mathcal{A}^{(L-1)} \\ w \in \mathbb{R}^m}} \exp \left( \lambda \sqrt{m} \left| \sum_{i=1}^n \epsilon_i \phi \left( \frac{1}{\sqrt{m}} \frac{w^\top}{\|w\|_2} \alpha^{(L-1)}(\mathbf{x}_i) \right) \right| \right) \right] \\
&\leq \frac{1}{\lambda} \log 2 E_{\{\mathbf{x}_i\}_{i=1}^n, \{\epsilon_i\}_{i=1}^n} \left[ \sup_{\bar{W} \in \mathcal{A}^{(L-1)}} \exp \left( \lambda \sqrt{m} \cdot \frac{1}{\sqrt{m}} \left\| \sum_{i=1}^n \epsilon_i \alpha^{(L-1)}(\mathbf{x}_i) \right\|_2 \right) \right] \\
&= \frac{1}{\lambda} \log 2 E_{\{\mathbf{x}_i\}_{i=1}^n, \{\epsilon_i\}_{i=1}^n} \left[ \sup_{\bar{W} \in \mathcal{A}^{(L-1)}} \exp \left( \lambda \left\| \sum_{i=1}^n \epsilon_i \alpha^{(L-1)}(\mathbf{x}_i) \right\|_2 \right) \right].
\end{aligned}
$$

Now we realize the expression on the right-hand side of the above equality is very similar to the second inequality of equation (53); therefore, we can iterate the same procedure using Lemma E.1 until we eventually obtain,

$$
\begin{aligned}
\Longrightarrow \frac{n}{(1+\rho_1)} R_n(\mathcal{F}) &\leq \frac{1}{\lambda} \log \left( 2^L \cdot E_{\{\mathbf{x}_i\}_{i=1}^n, \{\epsilon_i\}_{i=1}^n} \left[ \exp \left( \lambda \left\| \sum_{i=1}^n \epsilon_i \alpha^{(0)}(\mathbf{x}_i) \right\|_2 \right) \right] \right) \\
&= \frac{1}{\lambda} \log \left( 2^L \cdot E_{\{\mathbf{x}_i\}_{i=1}^n, \{\epsilon_i\}_{i=1}^n} \left[ \exp \left( \lambda \left\| \sum_{i=1}^n \epsilon_i \mathbf{x}_i \right\|_2 \right) \right] \right).
\end{aligned}
\tag{54}
$$

Now, we can closely follow the proof of (Golowich et al., 2018, Theorem 1) after the equation (9) from that same paper. We first define the random variable $Z = \left\| \sum_{i=1}^{n} \epsilon_i \mathbf{x}_i \right\|_2$ and obtain

$$\frac{1}{\lambda} \log \left( 2^L \cdot E_{\{\mathbf{x}_i\}_{i=1}^n, \{\epsilon_i\}_{i=1}^n} \left[ \exp(\lambda Z) \right] \right)$$
$$= \frac{L \log(2)}{\lambda} + \frac{\log \left( E_{\{\mathbf{x}_i\}_{i=1}^n, \{\epsilon_i\}_{i=1}^n} \left[ \exp \left( \lambda \left( Z - E_{\{\epsilon_i\}_{i=1}^n}[Z] \right) \right) \right] \right)}{\lambda} + \frac{\log \left( E_{\{\mathbf{x}_i\}_{i=1}^n} \left[ \exp \left( \lambda E_{\{\epsilon_i\}_{i=1}^n}[Z] \right) \right] \right)}{\lambda}.$$
(55)

Now,

$$E_{\{\epsilon_i\}_{i=1}^n}[Z] \stackrel{(a)}{\leq} \sqrt{ E_{\{\epsilon_i\}_{i=1}^n} \left[ \left\| \sum_{i=1}^{n} \epsilon_i \mathbf{x}_i \right\|_2^2 \right] }$$

$$= \sqrt{ E_{\{\epsilon_i\}_{i=1}^n} \left[ \sum_{i=1}^{n} \epsilon_i^2 \mathbf{x}_i^\top \mathbf{x}_i + \sum_{i=1}^{n} \sum_{\substack{j=1 \\ j \neq i}}^{n} \epsilon_i \epsilon_j \mathbf{x}_i^\top \mathbf{x}_j \right] }$$

$$\stackrel{(b)}{=} \sqrt{ \sum_{i=1}^{n} \left\| \mathbf{x}_i \right\|_2^2 }$$
(56)

$$\implies \frac{1}{\lambda} \log \left( E_{\{\mathbf{x}_i\}_{i=1}^n} \left[ \exp \left( \lambda E_{\{\epsilon_i\}_{i=1}^n}[Z] \right) \right] \right) \leq \frac{1}{\lambda} \log \left( E_{\{\mathbf{x}_i\}_{i=1}^n} \left[ \exp \left( \lambda \sqrt{ \sum_{i=1}^{n} \left\| \mathbf{x}_i \right\|_2^2 } \right) \right] \right)$$

$$= \frac{1}{\lambda} \log \left( \exp \left( \lambda \sqrt{n} \right) \right)$$

$$= \sqrt{n}$$

where (a) follows by Jensen's inequality and (b) from the fact that $\epsilon_i$ is independent from $\epsilon_j$, $j \neq i$.

If we consider $\{x_i\}_{i=1}^n$ as constants, then we have that $Z$ is a deterministic function of the i.i.d variables $(\epsilon_i)_{i=1}^n$ which satisfies

$$Z(\epsilon_1, \ldots, \epsilon_i, \ldots, \epsilon_n) - Z(\epsilon_1, \ldots, -\epsilon_i, \ldots, \epsilon_n) \leq 2 \left\| \mathbf{x}_i \right\|_2 ,$$

obtained by using the reverse triangle inequality. This inequality is a bounded-difference condition which implies that $Z$ (considering $\{x_i\}_{i=1}^n$ as constants) is sub-Gaussian with variance factor $v = \frac{1}{4} \sum_{i=1}^{n} 4 \left\| \mathbf{x}_i \right\|_2^2 = \sum_{i=1}^{n} \left\| \mathbf{x}_i \right\|_2^2$; see (Boucheron et al., 2013, Theorem 6.2); which then implies $E_{\{\epsilon_i\}_{i=1}^n} \left[ \exp \left( \lambda \left( Z - E_{\{\epsilon_i\}_{i=1}^n}[Z] \right) \right) \right] \leq \exp \left( \frac{1}{2} \lambda^2 \sum_{i=1}^{n} \left\| \mathbf{x}_i \right\|_2^2 \right)$. Then, we obtain

$$\frac{\log \left( E_{\{\mathbf{x}_i\}_{i=1}^n, \{\epsilon_i\}_{i=1}^n} \left[ \exp \left( \lambda \left( Z - E_{\{\epsilon_i\}_{i=1}^n}[Z] \right) \right) \right] \right)}{\lambda} \leq \frac{\log \left( E_{\{\mathbf{x}_i\}_{i=1}^n} \left[ \exp \left( \frac{\lambda^2}{2} \sum_{i=1}^{n} \left\| x_i \right\|_2^2 \right) \right] \right)}{\lambda} = \frac{1}{\lambda} \cdot \frac{\lambda^2 n}{2} = \frac{\lambda n}{2}.$$
(57)

Replacing equations (56) and (57) back in (55) with $\lambda = \frac{\sqrt{2 \log(2) L}}{\sqrt{n}}$, let us obtain

$$\frac{1}{\lambda} \log \left( 2^L \cdot E_{\{\mathbf{x}_i\}_{i=1}^n, \{\epsilon_i\}_{i=1}^n}[Z] \right) \leq \frac{\sqrt{2 \log(2) L n}}{2} + \frac{\sqrt{2 \log(2) L n}}{2} + \sqrt{n} = (\sqrt{2 \log(2) L} + 1) \sqrt{n}$$
(58)

which replacing back in (54) let us obtain,

$$\frac{n}{(1 + \rho_1)} R_n(\mathcal{F}) = (\sqrt{2 \log(2) L} + 1) \sqrt{n}$$

$$\implies R_n(\mathcal{F}) \leq \frac{(1 + \rho_1)}{\sqrt{n}} (\sqrt{2 \log(2) L} + 1).$$

Plugging this bound back in (49), along with (50) and (51), completes the proof. $\qquad \square$

