# OpenReview forum: "Optimization and Generalization Guarantees for Weight Normalization"
_TMLR — Accepted by TMLR_

### Review · Reviewer_z5TM · 2024-10-22

**Summary Of Contributions:**

This paper studies the following problem.
Given a dataset $(x_1,y_1),\dots (x_n,y_n)$, we define a loss function to be $L(\theta) := \frac{1}{n}  \sum_{i=1}^n \ell(y_i, f(\theta; x_i))$ where $f$ is a neural network parameterized by $\theta$ and $\ell$ is the loss function between the true value $y_i$ and the predicted value $f(\theta,x_i)$.
In particular, this paper considers the WeightNorm network, i.e. the norm of the weight vector in each layer is normalized to be $1$.
We train the neural network $f$ by using gradient descent and would like to minimize the generalization bound.
The authors provide the theoretical guarantees on the Hessian of WeightNorm networks and the generalization bounds.
Also, the authors evaluate their result on CIFAR-10 and MNIST datasets.

**Audience:**

Yes

**Broader Impact Concerns:**

.

**Claims And Evidence:**

Yes

**Requested Changes:**

- The authors may want to extend the result to non-smooth activation functions such as ReLU because it is more commonly used.

**Strengths And Weaknesses:**

Strengths:

- The paper provides the theoretical guarantees for WeightNorm networks.



Weaknesses:

- The result is limited to smooth activation functions which may further limit its practicality.
Also, the result mainly focuses on square loss which may not be helpful in classification tasks.

---

> ### Author Response · Authors · 2024-10-24
>
> We are grateful to the reviewer for the time and effort in reviewing our paper. We are also grateful for acknowledging the technical contribution of our paper.
>
> We agree that providing theoretical guarantees for WeightNorm under ReLUs is valuable. This is certainly an important future direction. We respectfully remark, however, that it is very common in the literature to focus on a single type of activation function: either smooth or non-smooth (in the latter, typically ReLUs). This stems from the fact that the choice of an activation function has a big impact on the type of mathematical tools used for establishing the theoretical guarantees. For example, using the references in Section 2 of our paper, (Zou and Gu, 2019; Nguyen, 2021) focus *only* on ReLUs, while (Du et al., 2019; Nguyen and Mondelli, 2020; Liu et al., 2022; Banerjee et al., 2023) focus *only* on smooth activations.
> Moreover, to the best of our knowledge, our work is the first one in providing both optimization and generalization guarantees for WeightNorm in deep networks. Thus, we had to make a choice regarding which type of activation function to use when tackling this largely unexplored problem. We believe that our work will further open the doors to more theoretical research on normalization methods for deep networks, and thus we expect a growing interest in ReLUs for WeightNorm. Finally, we remark that the setting of our work includes activations functions such as GELU (see Remark 6.1), a “smooth” version of ReLU widely used in practice; e.g., see the new references (Fang et al., 2023; Bouniot et al., 2024).
>
> We also agree with the reviewer that considering other losses beyond square loss is a valuable direction, e.g., cross-entropy used in classification tasks.  However, we remark that it is very common for theoretical works to only consider square loss in their studies: this can be seen in the analysis of both networks with ReLUs—e.g.,(Zou and Gu, 2019; Allen-Zhu et al., 2019)—and smooth activations—e.g., (Du et al., 2019; Nguyen and Mondelli, 2020; Banerjee et al., 2023). Finally, we point out that the recent work by Hui and Belkin (2021) (new reference) shows that in certain classification tasks, square loss has a comparable or even better performance than cross-entropy loss, thus showing that studying the square loss in our paper may have a wider relevance than commonly believed.
>
> **New references:**
>
> Quentin Bouniot, Ievgen Redko, Anton Mallasto, Charlotte Laclau, Oliver Struckmeier, Karol Arndt, Markus Heinonen, Ville Kyrki, and Samuel Kaski. “Understanding deep neural networks through the lens of their non-linearity”, International Conference on Learning Representations  (ICLR), 2024.
>
> Haishuo Fang, Ji-Ung Lee, Nafise Sadat Moosavi, and Iryna Gurevych. “Transformers with Learnable Activation Functions”, Findings of the Association for Computational Linguistics: EACL 2023, 2023.
>
> Like Hui and Mikhail Belkin. “Evaluation of Neural Architectures Trained with Square Loss vs Cross-entropy in Classification Tasks”, International Conference on Learning Representations  (ICLR), 2021.

---

### Review · Reviewer_83Qz · 2024-11-03

**Summary Of Contributions:**

This paper presents optimization and generalization guarantees for training neural networks with weight normalization. Under a proper parameterization of the networks (similar to NTK parameterization), the authors establish bounds on loss convergence and generalization for wide neural networks. Notably, these bounds have only a polynomial dependence on the depth of the network.

**Audience:**

Yes

**Broader Impact Concerns:**

/

**Claims And Evidence:**

No

**Requested Changes:**

1. Improve the readability of the proof in the appendix and add a high-level proof sketch in the main text.
2. More empirical validation of the theory to address Weaknesses 3 and 5.

**Strengths And Weaknesses:**

### Strengths

1. This paper presents optimization and generalization guarantees for neural networks with normalization, which is an important topic in deep learning theory.
2. The established bounds have only a polynomial dependence on the depth.

### Weaknesses

1. The proof is poorly written. The current version does not provide an outline for the proof, which makes it hard to see how the lemmas in the appendix are related to each other and contribute to the proof for the final theorem. Some equations and inequalities are too long and also the authors did not provide enough explanations. For example, the derivation on Pages 17, 20, 25, 30.
2. From the current version, it is hard to see how this paper may help researchers and practitioners to better understand the training dynamics of neural networks with weight normalization:
    1. Although this paper presents a lot of theoretical results, it merely states the difference to previous works on neural networks without normalization and doesn't provide any high-level explanation of why weight normalization is the key to deriving such results.
    2. The weight normalization studied in the paper is NOT the weight normalization used in practice because it has a trainable scaling factor but this paper just ignores it without further explanation.
3. A key proof insight is that the spectral norm of Hessian of the neural network output is small, but this also means the entire training is in the lazy training regime, where the neural network may be approximated by a kernel method. This regime usually leads to worse generalization in practice, no matter how good generalization bounds may be proved in theory. This issue could be addressed by running experiments on CIFAR-10 / ImageNet that closely follow the theoretical setup and show a comparable test error as a more standard training pipeline.
4. Assumptions A3.2 and A3.3 state that the weights are bounded in a local region near the initialization. Making such assumptions on the dynamics (rather than the loss landscape) significantly weakens the result because theoretically there might be no hyperparameter choices that can make the assumptions hold. This issue could be fixed if the authors could prove these assumptions for some learning rates.
5. Since a selling point of the paper is that the bounds do not strongly depend on the depth, can the authors actually show this in the experiments? In the current experiments, neural networks only have two hidden layers. It would be good to check if they still converge and generalize well as the number of layers increases.

---

> ### Author Response · Authors · 2024-11-11
> **Response, Part I**
>
> We are grateful to the reviewer for taking the time to provide us with a detailed review of our paper.
>
> --- We now address the points raised in the Weaknesses section:
>
> **Point 1.** We are sorry it was difficult to follow our proofs. In response to the reviewer’s comment, we will add an outline of the main proofs in the paper (in Sections 5 and 6). This will complement what we already have in the paper: in the first paragraph of Section 4, we have already mentioned that the lemmas in Section 4 are used in the main results of the following sections.
>
> We are grateful to the reviewer for pointing out derivations in specific pages of the Appendix where more explanations could be beneficial for our paper. In response, we will add more explanation to these derivations as requested by the reviewer. Nonetheless, we would like to point out that the apparent lack of “enough explanations” in these inequalities is either because we are using commonly found algebraic or norm inequalities, or because we are using derivations that have been pointed out in previous inequalities. For example, in Page 17 we only provide further detail to the last derivation of the inequality because it uses a specific lemma. As another example, in Page 21 we provide a great extent of detail to the derivations—e.g., the use of Cauchy-Schwarz inequalities, $L_2$ matrix norm to Frobenius norm inequality, and a previous lemma—and this is the reason why such details are omitted in the following inequalities such as the ones in Pages 25 and 30 pointed out by the reviewer. Finally, we would like to mention that, even though some inequalities may look “too long”, the reason why we decided to show them at such length is because we want to provide enough details for their derivation instead of shortening their derivations and be at risk of losing details that the reader may find useful.

---

> ### Author Response · Authors · 2024-11-11
> **Response, Part II**
>
> **Point 2.** We first would like to provide an example of how our work “may help researchers and practitioners to better understand the training dynamics of neural networks with weight normalization”: first, we note from equation (14) that the parameter $\\alpha$ could be larger when the minimum weight norm $\\|\\bar{W}\\|_2$ is larger, which, in that case, as can be seen in equation (18) from Theorem 5.1, will benefit the convergence rate, i.e., imply faster training. This result is what we supported with our simulations in Section 7. This beneficial dependency of the convergence rate on the minimum weight vector norm may provide an explanation to researchers and practitioners on why WeightNorm can accelerate training (Salimans and Kingma, 2016) since such beneficial dependency is absent in classic feedforward networks (i.e., in networks without WeightNorm).
>
> *Regarding the first subcomment*: We are grateful to the reviewer for providing us the opportunity to better explain why weight normalization is the key to deriving such different optimization results from feedforward networks. In response, we will add a remark in our paper with such explanation. Informally, WeightNorm allows for any gradient with respect to the weights to have extra factors of the form $1/\\| W^{(l)}\\|_2$. At the same time, factors of the form $\\| W^{(l)}\\|_2$ appear from bounds on the pre-activations, which are also present in neural networks without WeightNorm from prior works. Remarkably, WeightNorm results in the amount of $1/\\| W^{(l)}\\|_2$ factors being larger than $\\| W^{(l)}\\|_2$ factors, which implies the cancellation of the latter factors and the appearance of the former factors in the Hessian bound (see equation (1)). The appearance of such factors in the Hessian bound implies their appearance in the RSC and smoothness bounds, which finally implies their appearance in the convergence rate. We will also add a remark about how WeightNorm helped eliminating the exponential dependence on depth $L$ that appears in classic feedforward networks. Informally, what happens with neural networks without WeightNorm is that expressions that involved calculation of gradients *across* activations, due to chain rule, have factors $\\|W^{(l)}\\|_2$ floating around, which multiplied across hidden layers, become the product $\\prod^L\_{l=1}\\|W^{(l)}\\|_2$ for depth $L$. Prior works used random matrix theory and local assumptions on the weight matrices to obtain a bound of the form $\\|W^{(l)}\\|_2\\leq\\gamma$ for some constant $\\gamma>0$, which then translates to $\\prod^L\_{l=1}\\|W^{(l)}\\|_2\\leq \\gamma^L$. Note that this product of spectral norms leads to the exponential dependence on depth that appears in prior work (not considering WeightNorm) when $\\gamma>1$. The important insight about WeightNorm is that it introduces, again, in the gradient computations an appropriate amount of factors $\\frac{1}{\\|W^{(l)}\\|_2}$ which effectively cancel out these other $\\|W^{(l)}\\|_2$ factors that otherwise would be present should there be no WeightNorm (see the long inequality on top of Page 16 in the Appendix), thus effectively eliminating the exponential dependence on depth.
>
> *Regarding the second subcomment*: Indeed, we do not consider the more general setting of WeightNorm which adds a scalar *gain* $g$ (which can be made trainable) multiplying the normalized weight vectors: we have already mentioned this in the paragraph before the title of Section 4. However, since, to the best of our knowledge, our paper is the first one to formally study the theoretical underpinnings of optimization and generalization of WeightNorm in deep neural networks, we decided to simplify the problem in this first work by setting the gain $g$ to be a unit constant. What we have done should not come as a surprise: it is common in theoretical works to make certain assumptions in order to improve the tractability of the problem, especially when it is the first work studying the problem. As an example, many papers studying optimization and generalization of neural networks have made assumptions such as the networks having infinite width or being shallow—these assumptions are, if not impossible, hard to ensure in practice. Moreover, a practitioner is free to choose $g$ to be trainable or just be a constant—as an example, we do not consider a trainable gain in our experiments in Section 7. What is important to know is that our paper shows that when considering a constant gain $g=1$, we can provide theoretical guarantees to WeightNorm.

---

> ### Author Response · Authors · 2024-11-11
> **Response, Part III**
>
> **Point 3.** Regarding this comment, we would like to make a few points. Firstly, the Hessian bound we derived is valid for any *regime* where *all* the $L$ number of weights in the *hidden layers*, the width $m$, and the depth $L$, can take *any* value—notice from Theorem 4.1 that only the output layer is assumed to be bounded. It is true that the Hessian is small for larger values of weight norms and width, but this is not the result of an assumption on such parameters or a product of training: it is *intrinsic* to the architecture of the neural network. Secondly, we cannot conclude from our derivations that we are necessarily in the lazy training regime. As it has been previously argued in the literature, the lazy training regime is when the values of the weights do not change *too much* during the optimization and yet training goes through: it is known that when weights are able to change freely, even in a ball with a (large) constant radius of order $\\Theta(1)$ around the initialization point, we are no longer in the lazy training regime—we refer to the discussion in Section 1.1 from (Liu et al., 2020). Thus, if we look at Theorem 4.1 again, we find that the weights in the hidden layers are not restricted to take any value with respect to the initialization point, and the output weight can be stated to be in a ball with constant radius of order $\\Theta(1)$ around its initial value—in other words, we are not necessarily in the lazy training regime.
>
> **Point 4.** We are grateful to the reviewer for taking a closer look into our assumptions. Regarding assumption A3.2., we remark that we are **only** considering that the weight vector of the **last** layer is inside a neighborhood of its initialization value—the assumption does not include the weights from the hidden layers. Moreover, the radius $\\rho_1$ of this local region can be set to be any arbitrary constant of order $\\Theta(1)$ or even a polynomial of $L$; therefore, there is no assumption of $\\rho_1$ being arbitrarily close to zero, i.e., we are not restricting the local region to be arbitrarily *small* (this is a reason why we are not necessarily in the lazy regime, as pointed out earlier). We understand that such assumption on the output weight vector is a constraint on the dynamics, but it is a necessary one so that the linear output, which has no normalization, remains unbounded. Moreover, we point out that assumptions of closeness around initialization points are common in the theoretical literature on optimization of neural networks (e.g., see (Liu et al., 2020; Banerjee et al., 2023, Liu et al., 2023)) and so our assumption is not stronger than other existing works. Regarding assumption A3.3., we argue that this is not a restrictive assumption: what it simply says is that every time a new $\\theta_{t+1}$ is computed, we choose some constant $\\rho_2$ so that $\\theta_{t+1}$ is within a ball around $\\theta_{t}$ of radius $\\rho_2$. We argue that we should be able to find such a $\\rho_2$ because the empirical loss $\\mathcal{L}$ is **smooth**, and so, since we are using **gradient descent**, it is not possible for $\\theta_{t+1}$ to diverge and be “too far” from $\\theta_{t}$. Moreover, such $\rho_2$ is allowed to change across times, even though we did not explicitly state a time dependency on it. In conclusion, although assumptions A3.2 and A3.3 can be hard to characterize through hyperparameters, as we just argued, they are reasonable assumptions that are not foreign to theoretical works.
>
> **Point 5.** We appreciate the suggestion given by the reviewer and agree with the reviewer that experiments with deeper networks would be ideal to complement our theoretical contributions and expand our experiments already presented in Section 7. However, unfortunately, it is difficult for us to carry out multiple experiments with networks of increasing depth and provide useful plots due to computational constraints. We believe, however, that such experiments are valuable for follow-up works and we hope our paper will open the path to further research which we hope our paper will elicit. Since the contribution of our paper is theoretical, we anticipate more experimental papers to follow-up on our conclusions.

---

> > ### Author Response · Authors · 2024-11-11
> > **Response, Part IV (final part)**
> >
> > --- We now address the Requested Changes:
> >
> > We are grateful to the reviewer for the two points expressed in the Requested Changes section of the review, and we appreciate the reviewer’s interest in our paper. Regarding 1., we refer to our responses above in **Point 1.**: we will implement such changes to improve the readability of the proof and include a high-level proof sketch in the main text. Regarding 2., as mentioned in point **Point 3.** above, it cannot be inferred from our theoretical results that they are describing the lazy regime and, as mentioned in **Point 5.** above, we would defer further simulations to a follow-up work. Again, we would like to kindly remark that our paper’s main technical contributions are theoretical and that our simulations already show the effect that the values of the minimum weight vector norm have on the optimization of WeightNorm networks, which is a crucial observation derived from our results and which, as argued in **Point 2.** above, is also useful for those who want to “better understand the training dynamics of neural networks with weight normalization”.

---

### Review · Reviewer_GJGR · 2024-11-04

**Summary Of Contributions:**

Normalization strategies are a form of regularization that seeks to enhance learning algorithms by improving (1) the convergence dynamics during the training phase and (2) the generalization properties of trained predictors. This article investigates the impact of WeightNorm (Salimans and Kingma, 2016), a normalization strategy where weight vectors are constrained to be unit Euclidean norm. The analysis in this article is focused on a regression task where inputs are unit-norm (Assumption 2).  The hypothesis class under consideration is multi-layer feedforward networks with weight normalizationa and smooth 1-Lipschitz activations (Assumption 1).

The core contributions of this article are,
1. A training convergence guarantee for WeightNorm predictors under a variant of restricted strong convexity assumption (Assumption 3).
2. A uniform convergence generalization bound with favorable dependence on architectural hyperparameters such as width and depth.

To establish these results, the authors undertake careful calculations where the norms of the predictor gradient and Hessian (on the training data) are bounded. These calculations benefit from the unit-norm constraint on the weight vectors enforced by design in WeightNorm predictors. To see why, if $w_1$ and $w_2$ are two arbitrary vectors consider the impact on their distance $\|w1-w2\|_2$ before and after projection to the unit-surface. If $w_1$ and $w_2$ are initially far away from unit surface, then the distance between their projections are drastically smaller compared to the original distance (i.e better sensitivity w.r.t $w$’s). But if the vectors $w_1$ and $w_2$ are initially closer to the origin, then the distance between them is enlarged because each unit angular change corresponds to larger norm of perturbation in the unit surface. This intuition I believe conveys the authors' central observation that inverse of the minimum weight vector norms impacts the sensitivity.

**Audience:**

Yes

**Broader Impact Concerns:**

I don't anticipate any concern on the broader impact of this scientific article.

**Claims And Evidence:**

Yes

**Requested Changes:**

I suggest the following minor changes (and questions to address) to improve the exposition.
### Minor Technical Questions
- The inequalities in the final lines of equations after (49) differ the last inequality in (49) with a $\sqrt{m}$. Does this impact the result?
- Why is the output layer not directly assumed to be unit-norm like the feedforward layers? There is a consistent emphasis that the output layer vector $v$ is within distance $\rho$ from a reference unit vector. As a result all bounds additionally carry the factor $(1+\rho)$. It appears that the final results would be greatly simplified if $v$ is instead directly assumed to be on the unit-surface (i.e. weight normalization is also applied to $v$).


### Minor Editorial Comments
- Fix the spacing in the appendix proofs if possible.
- Don’t use $v$ as a free parameter when it also has a specific meaning as the last linear layer (for eg. in Proof of Lemma A.2).

**Strengths And Weaknesses:**

# Strengths
1. Theoretical guarantees for both optimization and generalization that indicate the favorable impact of weight normalization.
2. This article is well-written, and clear (including the several proofs in the appendix). To the best of my knowledge, the calculations in this appendix are sound and intuitive.

# Weaknesses
I think the technical results are potentially overstated in the writeup. At several points, the authors claim that their generalization bounds avoid the exponential dependence in depth. I contend that the contribution in this aspect is more nuanced. I am interested in hearing the authors’ perspective on my comments below.

1. Suppose the activation functions are $M_a$-Lipschitz and the weight normalization ensures $M_w$ Lipschitz constant in each layer (by an additional scaling factor of $M_w$ instead of unit norm), then the resulting bounds on the norms of gradients, Hessians, and the Lipschitz constant all exhibit the dependence $\prod_{1\leq l \leq L} M_a M_w$, i.e. the bounds are exponential in depth. The authors are *correct* in their observation that setting $M_a, M_w$ to be 1 presents a opportunity to instantiate a O(1)-Lipschitz predictor. However, I am not certain that weight normalization is critical for this effect.

When the learning task is classification (rather than regression as considered here), and the activations are ReLU, the bounds on the norm of the margin scale with the norms of the weight vectors. This is the primary motivation of the spectrally-normalized margin bounds developed in (Bartlett et.al, 2017). In particular, for any multi-layer feedforward network f with no bias vectors, if each of the feedforward weight vectors are scaled by a constant $M_{scaling}$, then the function output is correspondingly scaled by $M_{scaling}^L$. Further one is always free to choose $M_{scaling} = (M_a M_w)^{-1}$ which results in an equivalent 1-Lipschitz feedforward network with the same prediction properties and error as the original unnormalized predictor. Such an observation is also noted in Neyshabur et. al. (2015)'s work on path-normalization of ReLU networks. In this article, unless I'm mistaken the assumption of smooth activations can be removed for the generalization bound (correct me if I'm wrong!).

With this context, for a classification task if one fixes the margin threshold $\gamma$ in (Bartlett et.al, 2017) arbitrarily then the bound on the generalization error is indeed exponential in depth as it depends on the Lipschitz constant $(M_a M_w)^L$ (see spectral complexity in Bartlett, et.al  (2017)). However, if the the margin threshold is set to be $\gamma_{scaled} = \gamma *  (M_a M_w)^{L}$ then the bounds on the generalization error are not exponential in depth! Thus, generalization bounds in the classification setting are readily devoid of exponential dependence when viewed through the spectrally normalized margin perspective. This behaviour is possible because the absence of bias vectors lends the networks amenable to such a simple scaling. (Such a scaling is less clear if the networks have non-zero bias but this article restricts to discussion to zero-bias networks.)

As a result of this behaviour, generalization bounds are exponential in depth if the margin threshold or loss function are oblivious to the scaling and the bounds are not exponential in depth if the threshold or loss function is appropriately scaled (as spectrally normalized margin suggests). While the above comment is explicitly for classifications tasks, the analogous phenomena for regression tasks is the scaling of the observed outputs. If the observed output $y$ is assumed to have unit-norm then appropriately scaling the predictors ensures a good generalization bound and if instead each of the weight norm layers are scaled to an alternate threshold $M_w$ then the resulting bounds exhibit exponential dependence in depth again. Such an appropriate scaling does not require WeightNorm for classification as it can be overlayed on any learnt predictor due to the scaling invariance in classification prediction. Similarly for regression, this scaling has to be adapted to the norms of the observed outputs and any regression function can be scaled this way, if the regression task is itself normalized such that the observed outputs have unit norm. Thus, I think the absence of explicit exponential dependence in depth in the main generalization reflects the appropriate scaling for the task under consideration (indeed Theorem 6.1 assumes that $|y|<=1$ w.p.1).

2. I think there might be a ready-made bound for the Rademacher complexity of the output of WeightNorm predictors on the training data. To see this, I believe one can instantiate the bound on the Rademacher complexity just like Lemma A.8, Bartlett et. al (2017). (Assuming Bartlett et. al’s notation), For WeightNorm predictors, the layer-wise spectral norms $s_j$ are bounded by 1 and the l1 norm bounds $b_j$ are bounded by the layer widths ($m$ in this case) and the activation function Lipshcitz constants $rho_j$ are simply 1. In this case, the spectral complexity does not exhibit exponential dependence in depth in the same way the sensitivity calculations in this article do not. The resulting bound on the Rademacher complexity via the spectral complexity scales as $\sqrt{\text{depth}}$ just like the bound on the Rademacher complexity in this article in Page 42. The straightforward on the Rademacher complexity via Bartlett's result automatically translates to a bound on the generalization error by the standard analysis and already avoids the explicit exponential dependence in depth when the network is weight normalized. There are still subtle differences between this straightforward bound and the bounds developed in this article, (perhaps with width dependence) and I think this point of comparison needs to be clearly communicated in the draft to appropriately contextualize the exponential dependence on depth.

3. As a final technical point, even when the predictors are appropriately normalized to be 1-Lipschitz, there is still a version of exponential dependence in depth that matters. This stems from a conservative estimation of the true sensitivity in each layer, for eg. if at each input, a predictor is $\frac{1}{2}$-locally Lipshcitz at an input x, the function output exhibits a local Lipschitz constant of ${\frac{1}{2}}^L$ but the global Lipschitz constant is still 1! This results in a poor generalization bound since the spectral norm scaling of the margin thresholds is inadequate. Here, the exponential dependence on depth of the local Lipschitz senstivity is beneficial if one is able to appropriately account for it in generalization theory. This observation has motivated a few articles that I think deserve a citation and comparison amidst the discussion on exponential dependence in depth. In particular refer -  Nagarajan and Kolter (2019), Wei and Ma (2019), Muthukumar and Sulam (2023). While these article often discuss bounds for supervised classification, they each make a case for avoiding exponential dependence in depth and their results have reasonable extensions to the regression setting. Additionally, Wei and Ma (2019) consider smooth networks and compute bounds that depend on the local linear Lipschitz constant (i.e norms of Jacobians) and are closely related to the authors’ work.

If the authors agree with the above comments, then the claim of exponential dependence needs to be significantly revised to account for the above nuances. I want to emphasize that I do not intend to downplay the authors’ contribution but rather I would like a more appropriate contextualization of this well written article in certain key aspects.

### References
- Behnam Neyshabur, Ryota Tomioka, Nathan Srebro. Norm-Based Capacity Control in Neural Networks. (COLT 2015)
- Peter L Bartlett, Dylan J Foster, and Matus J Telgarsky. Spectrally-normalized margin bounds for
neural networks.  (NeurIPS, 2017)
- Vaishnavh Nagarajan and Zico Kolter. Deterministic PAC-bayesian generalization bounds for deep
networks via generalizing noise-resilience. (ICLR 2019)
- Colin Wei and Tengyu Ma. Data-dependent sample complexity of deep neural networks via lipschitz
augmentation. (NeurIPS, 2019)
- Ramchandran Muthukumar, and Jeremias Sulam.. Sparsity-aware generalization theory for deep neural networks. (COLT 2023).

---

> ### Author Response · Authors · 2024-11-11
> **Response, Part I**
>
> We are grateful to the reviewer for taking the time to provide us with a detailed review of our paper. We appreciate all the positive comments regarding our contributions. We are glad the reviewer found our paper and its proofs well-written and sound.
>
> -->**Regarding the Weaknesses section:**
>
> We appreciate the in-depth discussion of our generalization results provided by the reviewer. We now proceed to address the raised comments.
>
> **Regarding point 1.**, we make the following points:
>
> - Before stating why we believe weight normalization is critical for avoiding exponential dependence in our paper, we make a couple of remarks. Firstly, we start by agreeing that the Lipschitz constant being one helps avoiding constants that otherwise one would need to eliminate with some rescaling in order to avoid an (additional) exponential dependence in the generalization bound. The Lipschitz property is used, for example, in the uppermost unnumbered equation in page 39, in the inequality labelled as “(c)”—a similar inequality is also used by Golowich et al., 2018. Secondly, we also agree with the reviewer that our generalization results can work with activations that are not necessarily smooth, because we only use Lipschitz properties of the activation function and the fact that it takes zero value at zero. However, the reason why we kept the smooth activation assumption in our generalization results was to maintain harmony with the optimization part of the paper that only considers smooth activations. In response to the reviewer’s comment, we can add a small remark in our paper about how our generalization guarantees does not need the assumption of smooth activations.
> - Now we proceed to address how weight normalization is critical for eliminating exponential dependence on the depth in our paper (having a unit Lipschitz activation function is not enough). We are thankful to the reviewer for bringing-up the fact that in a classification setting, one may consider margin thresholds to avoid exponential dependence. As the reviewer also pointed out, things are different in our paper because we instead consider a regression setting. The reviewer seems to suggest that a correct scaling in the output should be enough for avoiding the exponential dependence (assuming we understood the reviewer correctly; otherwise, we would be happy to be corrected). However, we respectfully argue that this may not be true because additional assumptions may be needed. For example, Golowich et al., 2018 also consider regression tasks (see the first equation in Section 2 of (Golowich et al., 2018)) and they have to make *assumptions* on the *bounds* of the weight norms. Thus, Lemma 1 and Theorem 1 of (Golowich et al., 2018) make assumptions on the Frobenius norm of the weights. While the results by Golowitch et al., 2018 allow the elimination of an exponential dependence on the depth (which is the focus of their paper and the reason why we adapted their results to our setting), they still keep a product of the assumed bounds on the weight norms across the network’s depth, which, in itself, can lead to another exponential dependence if such norm bounds are larger than one. This is where WeightNorm comes into play. WeightNorm, remarkably, removes the necessity to assume any upper bounds on the weight norms because WeightNorm *conveniently* and *intrinsically* already constraints the ($L_2$-)norm of each row of the weight matrices to have unit value. This unit value avoids any product of constants that could lead to an exponential dependence across the depth. Indeed, if we look at the last equality of the unnumbered equation in page 39, we observe that we get an $L_2$-norm without any constant multiplying it (inside the function $g$); on the other hand, Lemma 1 from (Golowich et al., 2018) obtains instead an $L_2$-norm with an additional constant factor (inside the function $g$) which comes from constraining the Frobenius norm of the weight matrices (see also Theorem 1 by the same authors). A similar story of introducing constants due to norm constraints happens when Golowich et al., 2018 consider another type of norm in their Lemma 2 and Theorem 2. Thus, if we didn’t have WeightNorm, we would have to impose additional constraints on the weight norms, with the risk of exponential dependence if those constraints become larger than $1$. Therefore, we respectfully maintain the assertion that WeightNorm allows the avoidance of exponential depth dependency in the networks studied in our paper. We can add a remark based on the above to our manuscript, if the reviewer feels this will better inform the readers.

---

> > ### Author Response · Authors · 2024-11-11
> > **Response, Part II**
> >
> > **Regarding point 2.**, we are grateful to the reviewer for bringing up the analysis of (Barlett et al., 2017) to our attention. We start by pointing out that the neural network used by Bartlett et al., 2017 is a different one than ours: in particular, if we look at their equation (1.1), unlike our paper, they consider an activation function at the output—we notice that Bartlett et al., 2017 focus on classification tasks. This is besides the fact that they do not consider weight normalization. Thus, it is uncertain whether we can consider their Rademacher complexity bound as “ready-made” for the WeightNorm regression-oriented neural networks studied in our paper. For example, the Rademacher complexity bound in (Bartlett et al., 2017) seems to depend on a *margin* parameter $\\gamma$ which does not make sense in our regression setting. Moreover, we observe that the numerator of the Rademacher complexity bound in (Bartlett et al., 2017) (see Lemma A.8 in (Bartlett et al., 2017)) has a term equal to a bound on the quantity $\\sqrt{\\sum_i\\|x_i\\|_2^2}$. This quantity, according to our scaling used throughout our paper, is equal to $\\sqrt{n}$ with $n$ being the sample size. This is perhaps not a big issue for Bartlett et al., 2017 because they have a factor $\\gamma^n$ in the denominator—however, since we are in a regression setting, it is possible that such $\sqrt{n}$ could be harmful and loosen our bound in terms of $n$. Other than these potentially negative issues, it is true, as the reviewer remarked, that WeightNorm can prevent an exponential dependence resulting from the product of bounds on the layer-wise spectral norms. Therefore, given all these arguments, we respectfully argue that the “ready-made” bound for the Rademacher complexity suggested by the reviewer may not work for our case.  We will be happy to hear back from the reviewer in case we have accidentally missed any of the reviewer’s arguments.
> >
> > **Regarding point 3.**, we are grateful to the reviewer for pointing out these references related to the literature of generalization in neural networks. We will include them in our paper.
> >
> > We want to conclude by remarking that we value the reviewer’s appreciation of our work and that we appreciate the provided detailed discussion about our generalization results. Given our responses above, we respectfully believe that WeightNorm plays a critical role in avoiding an exponential dependence on the depth, as stated in our paper. If we, by any chance, have missed any argument raised by the reviewer, we would be grateful if the reviewer could let us know—we are happy to continue our discussion. As mentioned earlier, we will add a remark about the activation functions for our generalization results and the references suggested by the reviewer.

---

> ### Author Response · Authors · 2024-11-11
> **Response, Part III (final part)**
>
> -->**Regarding the Requested Changes section:**
>
> We are grateful to the reviewer for the suggested minor changes and questions to address.
> Regarding the two Minor Technical Questions (in the order presented by the reviewer):
> - We point out that the last inequality in equation (49) is the same as the first inequality in the unnumbered equation that follows equation (49), i.e., the latter is a continuation of bounds on the former. This continuation of bounds is needed to eliminate the $\\sqrt{m}$ scaling that appears in equation (49); otherwise, an inconvenient width dependence would appear on the sample complexity for generalization.
> - We respectfully argue that our analysis and final results would not be greatly simplified if $\\mathbf{v}$ is normalized. For example, introducing a normalized $\\mathbf{v}$ would alter the Hessian and make its analysis more complex because now the neural network’s output would have a non-zero double derivative with respect to the output weight vector (which would add an extra bound term on equation (21)). Moreover, a derivative with respect to $\\mathbf{v}$—e.g., the cross-derivatives in the Hessian (the term $W^{(L+1)}:=\\mathbf{v}$ in equations (21) and (37)) and the predictor gradient (equation (6))—would introduce additional normalization terms that would have to be carefully analyzed across the bounds and that would appear on our final results. We also emphasize that normalizing $\\mathbf{v}$ may still introduce a factor to be carried around the bounds (similar to how $1+\rho_1$ is carried around) if we take into account the regression tasks one would use the neural network for. If we normalize the output vector, then the absolute value of the neural network’s output is upper bounded by $1$ *whenever* $\\phi(0)=0$ (i.e., when the activation is zero at the origin), as can be seen in equation (40). This means that we may need to add an additional constant gain factor to the output if we want the output to span a different range of real numbers in the regression task—a similar effect which is currently carried out by $\\rho_1$. In conclusion, normalizing $\\mathbf{v}$ would introduce more complexity to our theoretical results—therefore, since the neural network, as it currently is in the paper, already shows the benefits of weight normalization in optimization and generalization, we do not see the need to normalize $\\mathbf{v}$.
>
> Regarding the two Minor Editorial Comments (in the order presented by the reviewer):
> - We will go over the appendix proofs and fix any spacing issues that we find.
> - We are thankful to the reviewer for spotting this confusing use of notation. We will change the free parameter $\\mathbf{v}$ from the proof of Lemma A.2 for a different variable notation that has not been used in the paper such as “$\\bar{\\mathbf{v}}$”.

---

> ### Comment · Action_Editor_WCJ6 · 2024-11-24
> **On the comparison to the existing literature**
>
> Dear Authors,
>
> Thank you for your patience with the reviewing process.
>
>
>
> **On the generalization bounds**:
>
> I have to agree with the reviewer regarding the generalization bounds (proofs on pages 38 to 42): to the best of my understanding, they really do follow almost as particular cases of Theorems 2 and 5 in [1] (the only very slight technicality to handle is the unboundedness of the square loss, which can be handled with a one line argument using your weight normalization strategy.  Certainly, the proofs are almost identical as well. Furthermore, as reviewer GJGR pertinently pointed out, the results are much weaker than those of Bartlet et al. [3].
>
> You mention that the bounds in [3] only apply to classification, but this is really a minute technicality as the exact same proof works for regression settings. It is fair to assume that the regression equivalent (the case of a truncated square loss) of [3]  is "known".  Similarly to the comparison with [1],  I admit that in your case, you can use the non truncated version because of the normalisation constraints on the weights, but an equivalent result to [3] for an unbounded loss still follows nearly immediately.
>
> You also mention that there is a factor of $\sqrt{\sum_{i=1}^n \|x_i\|^2}$. However, this is in fact a technicality that makes their result slightly stronger, not weaker, than yours. Indeed, ignoring function class capacity factors, the scaling in [3] is $\frac{\sqrt{\sum_{i=1}^n \|x_i\|^2}}{\gamma n}$. If we assume (as you do, assumption 2 on page 4 of your manuscript) that $\|x_i\|\leq 1$, then it is certainly the case that $\frac{\sqrt{\sum_{i=1}^n \|x_i\|^2}}{\gamma n}\leq \frac{1}{\gamma \sqrt{n}}$.
>
>
>
> Therefore, unless you can provide very strong counter arguments, I think it is fair to assume that the generalization bounds are nearly trivial. Certainly, I agree with all reviewers that the next revision/camera ready version of the paper **should not claim** to provide the first generalization bounds independent of depth.
>
>
> **On the optimization guarantees**
>
> However, to the best of my understanding, the generalization bounds only constitute a small proportion of the contribution. Therefore, the key factor which should determine acceptance or rejection is whether the optimization guarantees are truly different from the existing work of [2], or if their relationship with [2] is as close as that of the generalization bounds with those of [1]. I am much less familiar with this part of the literature, so I would have to rely on you (and the reviewers) to better explain the difference between your results and the corresponding theorems in [2]. For instance, if I look at corollary 4.1 in your paper and compare to Theorem 4.1 in [2] with $\gamma=1$, $\rho_1=\rho$, it appears the numerator of your result scales the same way as the whole bound in [2]. The denominator is the minimum norm of the weight vectors. Do we expect this to be greater than 1 or less than 1? It appears a possible area of differentiation between the works here is that you are applying normalization but still differentiating w.r.t. the original unnormalized vector.
>
>
>
> =========================
>
>
> Could I ask you to upload a revision which removes the claims of being the first bounds with no dependence on depth, and address (in your rebuttal) the distinction between your work and [2]?
>
>
> Many thanks in advance,
>
> AE
>
>
>
> **References**
>
>
> [1] Noah Golowich, Alexander Rakhlin, Ohad Shamir, "Size-Independent Sample Complexity of Neural Networks", COLT 2018
>
> [2] Arindam Banerjee, Pedro Cisneros-Velarde, Libin Zhu, Mikhail Belkin, "Restricted Strong Convexity of Deep Learning Models with Smooth Activations", ICLR 2023.
>
>
> [3] Peter Bartlett, Dylan J. Foster, Matus Telgarsky, "Spectrally-normalized margin bounds for neural networks", NeurIPS 2017

---

> ### Author Response · Authors · 2024-11-26
> **Response to AE, Part I**
>
> Dear Action Editor,
>
> We are grateful to you for reaching out to us and providing us with more information about the literature on generalization. In this response, we will use the labelled references “[1], [2], [3]” as used in your message.
>
> We are also grateful to you for pointing out the possibility that an adaptation of [3] could be done for WeightNorm networks. This would require a formal proof where, to the best of our understanding, we would need to prove that a WeightNorm network with a *standard* square loss has the **same** (or **very similar**) generalization effect as a network without WeightNorm with a *truncated* squared loss—which would be a new result in the literature. Considering this, we respectfully avoid affirming whether or not this would be a nearly “trivial” or “immediate” result. Nevertheless, even if one is able to obtain a tighter generalization bound for WeightNorm networks in our standard (non-truncated) squared loss setting, our paper’s original purpose and contribution will still hold and be the same as what we already showed with our current results: that WeightNorm naturally leads to a non-exponential dependence on the depth. If we have unintentionally misunderstood any technicality from your arguments, please, let us know.
>
> We continue by clarifying that our **current version** of the paper **does not claim** to provide the first generalization bounds *independent* of depth—in fact, it does not even claim to provide the first *non-exponential* dependence on the depth either. What we claim, again, is that WeightNorm allows for a dependence which is non-exponential on the depth. Our paper does mention that exponential dependence on the depth has been found in the previous literature of networks without WeightNorm, which we argue is true because, to the best of our knowledge, exponential terms, however they were derived, have appeared in some way or another in the literature. We believe this is particularly **true** in **our same setting** of standard (non-truncated) squared losses, such as in work [1]. Indeed, in work [1], the generalization bound contains a *product* of (assumed) bounds on the weight norms *across* the depth of the network, potentially resulting in another exponential term on the depth. Moreover, even if we consider the work [3], whose adaptation to a regression setting was discussed, we notice that the term $R_{\\mathcal{A}}$ in equation (1.1) from [3] has a product of weight norms across the depth, and thus, a *potential* exponential dependence on the depth (which is independent from the convenient dependence on the sample size $n$). Of course, it may be possible to show that $R_{\\mathcal{A}}$ has non-exponential dependence under WeightNorm, but this, again, would simply support our original paper’s claim that WeightNorm naturally allows for the removal of exponential dependence. Again, please, let us know if we have unintentionally misunderstood any technicality from your arguments.
>
> To further support the claim that we do not mention our paper to be the *first* in providing depth independent generalization bounds, we present excerpts from our paper where our generalization claims are compared to previous works. We say, for example that *“Thus, we avoid an exponential dependence on the depth found in the literature of networks without WeightNorm (Bartlett et al., 2017; Neyshabur et al., 2018; Golowich et al., 2018).”* in page 2—which we just argued it is true in the previous paragraph. A similar passage is found in page 8: *“This is a benign dependence [a non-exponential dependence], compared to the exponential dependence found in networks without WeightNorm (Bartlett et al., 2017; Neyshabur et al., 2018; Golowich et al., 2018).”*. Finally, another part of our paper says *“Unlike our work, all the cited works do not study WeightNorm networks–indeed, we show that WeightNorm allows the generalization bound to depend on $\sqrt{L}$, where $L$ is the depth.”* in Page 3—which is true, since WeighNorm was not studied in the cited papers. Again, in all these texts, our intention was to claim that WeightNorm allows for a generalization bound that is naturally non-exponential on the depth, unlike networks without WeightNorm (particularly in our setting of standard squared losses). We also remark that we gave due credit to the work [1] at the beginning of Section 6 and in the appendix when deriving our results. Finally, we suspect that the three quotations just presented from our paper were the ones that Reviewer GJGR was referring to during our discussion—and we, again, remark that we did not use the strong word “first” in any of them. In any case, we would like to know if you and/or Reviewer GJGR have further suggestions for the presentation of our generalization claims.

---

> ### Author Response · Authors · 2024-11-26
> **Response to AE, Part II (final part)**
>
> Regarding our optimization results, we have a series of remarks in the paper (Remarks 4.1, 4.2, 4.3, 4.4, and 5.5) pointing out the differences between our results and the ones from [2]. A first important difference is that we had to carefully deal with normalization terms across all our derivations, which introduced differences in our tensor/matrix analysis compared to [2]. As a result, as you noticed, normalization terms (depending on the minimum weight vector norm) showed up in our bounds. Another important difference is that our results, unlike [2], are “deterministic” (i.e., they do not need to hold with high probability) and are not constrained to a neighborhood around the initialization point for the weights corresponding to the hidden layers.
>
> Regarding your comment about our Corollary 4.1 and the Theorem 4.1 from [2]: we respectfully remark that our *general* bound for WeightNorm networks is presented in our Theorem 4.1: Corollary 4.1 is just the particular case of Theorem 4.1 when the activation function is zero at the origin, i.e., $\\phi(0)=0$. Our bound in Theorem 4.1 certainly has a different numerator than the one in Theorem 4.1 from [2].
> Now, regarding your follow-up question on whether the minimum norm of the weight vectors are expected to be greater than $1$ or less than $1$, we remark that this will depend on the initialization: it is known that if the weights have a certain norm at initialization, then one can expect that such norm will not decrease across training for WeightNorm networks; see (Jacot et al., 2018). This behavior was also observed in our experiments, where we chose initializations with minimum weight vector norms that are both over and under $1$, thus showing empirical evidence for our results under both regimes.
>
> Finally, addressing your last sentence within your comment about the optimization part of our work, we remark that we are differentiating with respect to the normalized vector. This is what elicited all the differences with respect to the work [2] that we mentioned two paragraphs above. Indeed, we can even add yet another difference with respect to [2]: it is possible for the bounds in [2] to have exponential dependence on the depth if the initialization variance is not carefully chosen (see Theorem 4.1 in [2]), while this is not the case for our paper because our non-exponential dependence on the depth is independent from the initialization variance.
>
> We hope our response addresses your concerns. We will be happy to continue our discussion and clarify any further points.
>
> Thanks again for reaching out to us and suggesting ways to improve the presentation of our paper.
>
> Best regards,
>
> The authors

---

### Decision · Action_Editor_WCJ6 · 2024-12-28

**Recommendation:** Accept with minor revision

**Comment:**

The paper studies neural networks with batch normalisation and proves results which consist of two components: optimisation guarantees and generalisation guarantees.

The optimisation guarantees constitute the bulk of the work and culminate in Theorem 5.1, which implies convergence of the empirical loss under suitable conditions on the iterates. This is an analogue of Theorem 5.3 in [1]. The generalisation guarantees instantiate the result of [3] (which generalises [2]) to the regression setting studied here, where the distance from initialisation is assumed to be bounded as a result of an assumption on the iterates.


Two out of three reviewers weakly recommend acceptance, whilst one competent reviewer recommends rejection.
The two most salient points during the discussion were:

1. The lack of novelty of the generalisation results (pointed our by GJGR and which I discussed with the authors during the discussion phase).

2. The limitation introduced by Assumptions 3.2 and 3.3.



Regarding point 1, it is true that the original description of the results was misleading in presenting the results as original and as a solution to the exponential width dependence present in typical norm based generalisation bounds for neural networks [2,3,4]. Indeed, the bounds are indeed a **nearly direct consequence of** existing work [3] when incorporating the assumptions made in this work. However, after the discussion with the authors, the authors have acceptably downplayed their claims. Perhaps going slightly further in this contextualisation step would further strengthen the camera-ready revision.

On a tangent note related to point 1, it is worth mentioning also that parameter counting bounds such as those of [4] naturally do not explicit any exponential depth dependence, even without batch norm (the results from [4] apply to DNNs as a particular case of CNNs). More generally, there are plenty of bounds for neural networks with subtle differences in depth scaling compared to [3]. See, e.g. [5] and the references therein for details and comparisons.


Regarding point 2, I agree that **assumptions 3.2 and 3.3** introduce a significant limitation. I understand the authors' rebuttal which states that the conditions on $\rho$ can be incredibly mild and that (quoted from the rebuttal) **"we should be able to find such a $\rho_2$ because the empirical loss  is smooth, and so, since we are using gradient descent, it is not possible for $\theta_{t+1}$  to diverge and be “too far” from $\theta_{t}$"**. However, I still feel like a full resolution would have to express this vague statement in more quantitative terms, and achieve a bound which only depends on constants related to the loss function rather than relying on assumptions on the iterates. I am not sure if that is fully possible, but I would like the authors to try in the camera-ready (minor) revision (even for a specific loss function). At an absolute minimum, if a solution is genuinely not possible, the limitations and the reason for the difficulty should be discussed.

The accepted ICLR work of [1] uses a similar *iterates assumption*, so I don't think this issue is strictly cause for rejection, but I would like to see either a solution or a candid description of the limitations in the camera ready revision.

It might also be a good idea to mention (near the discussion of Theorem 5.1) that remark 5.4 (which controls the ratio $\alpha/\beta$ by showing it is less than 1) holds regardless of the value of $\rho$ and incorporate reformulations of some of the rebuttal to reviewer 83Qz  into the revision to make Theorem 5.1 more approachable to readers (such as myself and reviewer 83Qx) who might be initially shocked by assumption 3.


In conclusion, the paper makes a valid contribution to the field and **deserves to be accepted** after some further polishing according to the changes requested above.

***References***


[1] Arindam Banerjee, Pedro Cisneros-Velarde, Libin Zhu, and Misha Belkin. Restricted strong convexity of deep learning models with smooth activations. ICLR 2023

[2] Behnam Neyshabur, Ryota Tomioka, and Nathan Srebro. Norm-Based Capacity Control in Neural Networks. COLT 2015

[3] Noah Golowich, Alexander Rakhlin, and Ohad Shamir. Size-Independent Sample Complexity of Neural Networks. COLT 2018.

[4] Peter L. Bartlett, Dylan J. Foster, and Matus Telgarsky. Spectrally-normalized margin bounds for neural networks. NeurIPS 2017.

[5] Philip M. Long and Hanie Sedghi. Generalization bounds for deep convolutional neural networks. ICLR 2020.

[6] Florian Graf, Sebastian Zeng, Bastian Rieck, Marc Niethammer, and Roland Kwitt. On Measuring Excess Capacity in Neural Networks. NeurIPS 2022.

**Audience:**

Optimisation and generalisation guarantees for neural networks are certainly mainstream for TMLR's audience. The results are arguably new (though somewhat incremental) and will certainly be of interest to many members of TMLR's audience.

**Claims And Evidence:**

Although there were many complaints from the reviewers regarding novelty/clarity aspects, there were no complaints regarding correctness. The parts of the paper I have read are also correct. The paper clearly includes some sound results.